# TRACER: Persistent Regularization for Robust Multimodal Finetuning

**Hesam Asadollahzadeh** [1]  **Feng Liu** [1]  **Christopher Leckie** [1]  **Sarah M. Erfani** [1]

## Abstract

Mainstream strategies for finetuning pretrained multimodal models often degrade out-of-distribution (OOD) robustness, a phenomenon known as catastrophic forgetting. In this paper, we develop a theoretical framework for multimodal contrastive finetuning, yielding closed-form solutions and a geometric decomposition for these strategies. This framework shows that self-distillation is more effective than other regularization approaches to retain the knowledge of the pretrained model. Our analysis reveals a largely overlooked limitation: standard Exponential Moving Average (EMA) teachers, widely used in robust finetuning, suffer from collapse. To solve this, we prove that a Weighted Moving Average (WMA) teacher maintains a persistent regularizing force over finite horizons and yields bias-free convergence in the task subspace while preserving orthogonal knowledge. These insights motivate **TRACER** (**T**rajectory-**R**obust **A**nchoring for **C**ontrastive **E**ncoder **R**egularization), which combines contrastive learning with WMA-guided multi-perspective distillation. Extensive experiments on CLIP finetuning demonstrate consistent OOD accuracy and calibration gains across three backbone architectures, and comprehensive ablations confirm that TRACER is both principled and robust to hyperparameter choices. Code is available at https://github.com/HesamAsad/TRACER.

## 1. Introduction

Pretrained models such as CLIP (Radford et al., 2021) have revolutionized machine learning through their remarkable zero-shot transfer and adaptive capabilities. These models derive their robustness from large-scale multimodal pretraining (Fang et al., 2022; Xu et al., 2024b), enabling diverse applications from visual recognition (Shen et al., 2022b; Zhang et al., 2022b) to generative modeling (Betker et al., 2023; Pi et al., 2024) and serving as backbones for large multimodal models (Alayrac et al., 2022; Liu et al., 2023; Zhu et al., 2024).

Despite these successes, adapting these pretrained models to downstream tasks via finetuning presents a fundamental challenge: while finetuning improves in-distribution (ID) performance, it often degrades out-of-distribution (OOD) robustness (Radford et al., 2021). This trade-off manifests as catastrophic forgetting of pretrained knowledge (Wortsman et al., 2022b), where models sacrifice their general-purpose representations to optimize for task-specific patterns, potentially overfitting to spurious correlations in the finetuning data.

Several empirical strategies have emerged to mitigate this trade-off. For example, LP-FT (Kumar et al., 2022) addresses the problem of randomly initialized heads distorting pretrained features by first learning a linear probe on frozen features before full finetuning. FLYP (Goyal et al., 2023) extends this idea by reusing CLIP's pretrained text encoder as the classification head, maintaining consistency with the pretraining objective. Post-hoc methods like WiSE-FT (Wortsman et al., 2022b) and Model Stock (Jang et al., 2024) perform weight averaging between pretrained and finetuned models to recover lost robustness. Regularization-based approaches, including $L_2$–SP (Li et al., 2018) and self-distillation with dynamic teachers (Oh et al., 2024), introduce constraints to preserve pretrained knowledge. However, most dynamic-teacher methods rely on an Exponential Moving Average (EMA), whose regularizing influence provably weakens as the teacher approaches the student. This limitation is often overlooked in the robust-finetuning literature on catastrophic forgetting, yet it is precisely the reason why the OOD robustness is most fragile. Crucially, robust finetuning depends on maintaining sufficient regularization strength throughout training; when that strength decays, OOD robustness erodes even as ID accuracy improves. This motivates *trajectory regularization*: keeping the teacher anchored to the optimization path so that it continues to exert a meaningful restoring force.

Moreover, despite the proliferation of these methods, a the-

[1]School of Computing and Information Systems (CIS), Faculty of Engineering and IT (FEIT), University of Melbourne, Australia. Correspondence to: Hesam Asadollahzadeh <h.asadollahzadeh@unimelb.edu.au>.

*Proceedings of the 43rd International Conference on Machine Learning*, Seoul, South Korea. PMLR 306, 2026. Copyright 2026 by the author(s).

oretical understanding of *what* changes during contrastive finetuning and *where* forgetting occurs remains elusive. We address this gap by developing a theoretical framework that reveals the geometric structure of how different finetuning strategies modify pretrained representations. We find that the linearized contrastive finetuning objective can be reformulated as a matrix least-squares problem through what we call the *contrastive target matrix*. This reformulation enables closed-form solutions for common finetuning strategies, exposing their fundamentally different geometric behaviors.

Our theoretical insights lead to the design of **TRACER** (**T**rajectory-**R**obust **A**nchoring for **C**ontrastive **E**ncoder **R**egularization), a practical finetuning method that implements our geometric principles. As shown in Figure 1, TRACER combines contrastive learning with dynamic self-distillation, yielding strong results on ImageNet and its distribution shifts. Across multiple CLIP architectures, TRACER consistently improves the ID-OOD trade-off. We validate these findings through extensive ablation studies spanning distillation components, regularization strength, teacher update schedules, and kernel shape, demonstrating both the robustness of our method to hyperparameter choices and contributions of each design element.

In summary, our work makes the following main contributions: **(i)** We introduce the *contrastive target matrix* reformulation of linearized contrastive loss, turning the objective into a least-squares problem and enabling closed-form solutions for standard finetuning and regularization strategies; **(ii)** We derive a geometric decomposition that separates task–subspace mixing from orthogonal preservation, explaining when and where forgetting occurs and providing a principled basis for dynamic teachers; **(iii)** We identify a largely overlooked limitation of standard teachers in robust finetuning, the inherent collapse of the EMA teacher–student learning signal, and show how *trajectory regularization* with a weighted moving-average (WMA) teacher preserves a meaningful regularization signal over finite horizons, enabling bias-free task-subspace convergence; we instantiate these principles in **TRACER** with consistent OOD gains on CLIP finetuning, supported by comprehensive ablations across four axes (§B).

## 2. Related Work

Contrastive language–image pretraining (Radford et al., 2021; Jia et al., 2021; Ilharco et al., 2021; Zhai et al., 2023) enables strong zero-shot transfer but naive finetuning can harm OOD robustness (Taori et al., 2020; Wortsman et al., 2022b). Robust finetuning explores weight interpolation/averaging (Wortsman et al., 2022b;a; Jang et al., 2024), weight- or output-space regularization (Li et al., 2018; Li & Hoiem, 2018), and contrastive variants aligned to text

prompts or energies (Goyal et al., 2023; Mao et al., 2024; Nam et al., 2024; Shu et al., 2023). CaRot (Oh et al., 2024) couples contrastive training with new regularizers to jointly improve OOD accuracy and calibration.

Self-distillation and dynamic teachers stabilize learning and preserve knowledge (Hinton et al., 2015; Zhang et al., 2019; Mobahi et al., 2020; Laine & Aila, 2017; Tarvainen & Valpola, 2017). Momentum/EMA teachers are effective yet can introduce persistent bias toward initialization, and their teacher–student gap collapses as training converges, reducing regularization exactly when OOD robustness is most vulnerable. This critical flaw is rarely made explicit in the robust finetuning literature. Our *WMA* teacher generalizes EMA by weighting the entire trajectory on normalized time, enabling endpoint-aware curricula (e.g., arcsine/Beta kernels) and *trajectory regularization* that preserves a meaningful teacher gap over finite horizons. As we prove, this yields bias-free task-subspace convergence. Our theory complements linearized analyses of supervised and contrastive learning (Ji et al., 2023; Tian, 2022; Nakada et al., 2023; Xue et al., 2024; Hao et al., 2025) and explains forgetting via an explicit geometric decomposition. An extended literature review appears in §A.

## 3. Theoretical Analysis

To address the ID-OOD trade-off, where finetuning improves in-distribution accuracy at the cost of out-of-distribution robustness, we develop a theoretical framework that reveals the underlying dynamics of this phenomenon.

### 3.1. Problem Setting and Preliminaries

**Finetuning task.** We consider robust finetuning of a pretrained vision–language model on paired image–text data $\{(\mathbf{x}_I^i, \mathbf{x}_T^i)\}_{i=1}^n$ drawn from a downstream task. The goal is to adapt the model so that in-distribution accuracy improves on this task while pretrained, broadly transferable representations are preserved for out-of-distribution generalization.

**Linearized image/text encoders.** Following linearized analyses widely used in the theory of contrastive learning (Ji et al., 2023; Tian, 2022; Nakada et al., 2023; Xue et al., 2024), we model the image and text encoders as linear projections, $g_I(\mathbf{x}) = \mathbf{W}_I \mathbf{x}$ and $g_T(\mathbf{x}) = \mathbf{W}_T \mathbf{x}$. The image encoder $\mathbf{W}_I$ is adapted from a pretrained state $\mathbf{W}_I^0$, while the text encoder $\mathbf{W}_T$ is frozen at its pretrained state $\mathbf{W}_T^0$. We collect the $n$ image and text features of a batch into matrices $\mathbf{X}_I \in \mathbb{R}^{d_I \times n}$ and $\mathbf{X}_T \in \mathbb{R}^{d_T \times n}$, where $d_I, d_T$ are the input feature dimensions and $p$ is the shared embedding dimension.

**Original MMCL objective.** The linearized multimodal contrastive learning (MMCL) loss is a standard analytic surrogate of the symmetric InfoNCE objective (the full deriva-

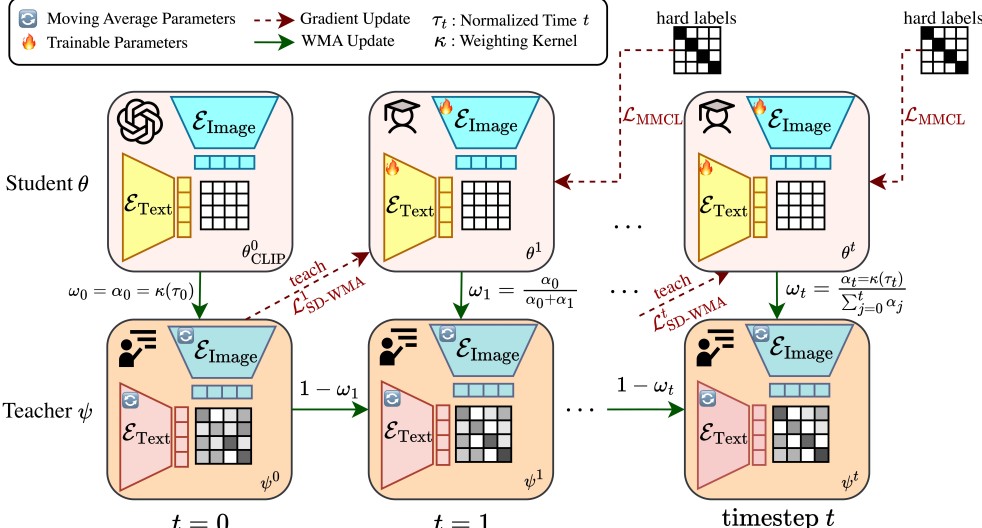

*Figure 1.* **Overview of TRACER.** The base contrastive objective is combined with a dynamic self-distillation loss from a Weighted Moving Average (WMA) teacher to preserve orthogonal pretrained knowledge while adaptively mixing within the task subspace. $\theta_{\text{CLIP}}^0$ represents the initial pretrained CLIP model. $\theta^t$ denotes the student model at time $t$, with its image and text encoder ($\mathcal{E}_{\text{Image}}$ and $\mathcal{E}_{\text{Text}}$) being trained. The student receives gradient updates from $\mathcal{L}_{\text{MMCL}}$. The WMA teacher model $\psi^t$ is updated from the student's parameters. The teacher then provides a teaching signal $\mathcal{L}_{\text{SD-WMA}}$ to regularize the student. This interplay allows TRACER to adapt to new tasks while preserving pretrained knowledge. The complete training procedure is detailed in Algorithm 1.

tion appears in §C.1):

$$
\begin{aligned}
\mathcal{L}_{\text{MMCL}}(\mathbf{W}_I, \mathbf{W}_T) = & \frac{1}{n(n-1)}\Big[\sum_{i \neq j} s_{ij} - (n-1)\sum_i s_{ii}\Big] \\
& + R(\mathbf{W}_I, \mathbf{W}_T),
\end{aligned} \tag{1}
$$

where $s_{ij} = (\mathbf{W}_I \mathbf{x}_I^i)^\top (\mathbf{W}_T \mathbf{x}_T^j)$ is the image–text similarity, and $R(\cdot)$ is a cross-Frobenius term. This loss balances pulling matched pairs together against pushing unmatched pairs apart. As shown in §C.2, when $\mathbf{W}_T = \mathbf{W}_T^0$ is frozen, optimizing equation 1 over $\mathbf{W}_I$ is equivalent (up to constants and a data-dependent quadratic term that arises naturally in the optimization) to a matrix least-squares problem driven by the contrastive target matrix introduced below.

**Notation.** We use $\mathbf{I}_n \in \mathbb{R}^{n \times n}$ for the identity matrix and $\mathbf{J}_n = \mathbf{1}_n \mathbf{1}_n^\top \in \mathbb{R}^{n \times n}$ for the all-ones matrix, where $\mathbf{1}_n$ denotes the $n$-dimensional all-ones vector. The matrix $n\mathbf{I}_n - \mathbf{J}_n$ acts as a centered contrastive operator on text features: it preserves the matched (diagonal) directions while subtracting the batch-mean direction, yielding a per-column "attract-paired/repel-others" signal. We let $\mathcal{P}_I := \mathbf{X}_I(\mathbf{X}_I^\top \mathbf{X}_I)^+ \mathbf{X}_I^\top$ denote the orthogonal projector onto $\text{range}(\mathbf{X}_I)$, the *task subspace* spanned by the finetuning image features, and $\mathbf{I} - \mathcal{P}_I$ the projector onto its orthogonal complement. Throughout, $(\cdot)^+$ is the Moore–Penrose pseudoinverse and $\|\cdot\|_{\text{F}}$ denotes the Frobenius norm.

### 3.2. Loss Reformulation via the Contrastive Target Matrix

We now rewrite the MMCL objective equation 1 in a form that exposes closed-form solutions via a single algebraic construct.

**Definition 3.1** (Contrastive Target Matrix). Given the frozen text encoder $\mathbf{W}_T^0$ and finetuning texts $\mathbf{X}_T$, we define the *contrastive target matrix* as

$$
\mathbf{Y}_{\text{FT}} := \mathbf{W}_T^0 \mathbf{X}_T (n\mathbf{I}_n - \mathbf{J}_n) \in \mathbb{R}^{p \times n}.
$$

Each column $\mathbf{y}_i$ is constructed to attract the image embedding $\mathbf{x}_I^i$ towards its paired text $\mathbf{x}_T^i$ and repel it from the remaining texts in the batch (detailed in §C.1).

**What is $\mathbf{Y}_{\text{FT}}$, and why is it "fixed"?** The matrix $\mathbf{Y}_{\text{FT}}$ depends only on the frozen text encoder $\mathbf{W}_T^0$ and the finetuning text features $\mathbf{X}_T$; it does *not* depend on the trainable image weights $\mathbf{W}_I$. Consequently, $\mathbf{Y}_{\text{FT}}$ stays *fixed throughout finetuning* and plays exactly the role of a target in a supervised regression problem: it is the centered contrastive signal that $\mathbf{W}_I \mathbf{X}_I$ should match. In analogy with linear regression, $(\mathbf{X}_I, \mathbf{Y}_{\text{FT}})$ form (inputs, targets) and the linearized MMCL loss reduces to a matrix least-squares problem with $\mathbf{Y}_{\text{FT}}$ as the regression target. The recurring factor $(n\mathbf{I}_n - \mathbf{J}_n)$ in $\mathbf{Y}_{\text{FT}}$ implements the contrastive centering with $\mathbf{I}_n$ and $\mathbf{J}_n$ as defined above.

Using $\mathbf{Y}_{\text{FT}}$, the linearized MMCL objective equation 1 reduces (up to constants and a data-dependent quadratic term) to:

$$\min_{\mathbf{W}_I} \frac{1}{2} \|\mathbf{W}_I \mathbf{X}_I - \mathbf{Y}_{\text{FT}}\|_F^2. \tag{2}$$

This formulation is crucial because it enables closed-form solutions for various finetuning strategies under gradient descent, offering insights into their behavior (see §C.2 for the formal trace-to-least-squares equivalence and §C.3 for the full derivations and proofs). Using this reformulation, we analyze how different finetuning strategies mitigate forgetting by preserving or adapting pretrained knowledge, and reveal the geometric structure of updates.

### 3.3. Closed-Form Solutions

This subsection follows the setting of Yang et al. (2024b) and provides closed-form solutions for various finetuning strategies following our reformulation (Equation 2), revealing a *geometric decomposition*: finetuning involves (i) preserving pretrained knowledge in directions *orthogonal* to the finetuning data, and (ii) adapting or mixing knowledge *within* the task-relevant subspace. We first present the closed-form solutions below.

**Theorem 3.2** (Unified Framework for Contrastive Finetuning Solutions). *Let* $\mathcal{P}_I := \mathbf{X}_I(\mathbf{X}_I^\top \mathbf{X}_I)^+ \mathbf{X}_I^\top$ *be the orthogonal projector onto* $\text{range}(\mathbf{X}_I)$. *Gradient descent initialized at* $\mathbf{W}_I^0$ *on the objective* $\mathcal{L}(\mathbf{W}_I) = \frac{1}{2} \|\mathbf{W}_I \mathbf{X}_I - \mathbf{Y}_{FT}\|_F^2 + \mathcal{R}(\mathbf{W}_I)$ *converges to the following solutions:*

1. ***Direct Finetuning*** *(*$\mathcal{R}(\mathbf{W}_I) = 0$*):*

$$\mathbf{W}_{FT} = \mathbf{W}_I^0(\mathbf{I} - \mathcal{P}_I) + \mathbf{Y}_{FT}\mathbf{X}_I^\top(\mathbf{X}_I \mathbf{X}_I^\top)^+$$

2. $L_2$ ***Regularization*** *(L2-SP (Li et al., 2018)), with* $\mathcal{R}(\mathbf{W}_I) = \frac{\lambda}{2} \|\mathbf{W}_I - \mathbf{W}_I^0\|_F^2$:

$$\mathbf{W}_{L_2} = (\mathbf{Y}_{FT}\mathbf{X}_I^\top + \lambda \mathbf{W}_I^0)(\mathbf{X}_I \mathbf{X}_I^\top + \lambda \mathbf{I})^{-1}$$

3. ***Static Self-Distillation*** *(SD (Furlanello et al., 2018)), with* $\mathcal{R}(\mathbf{W}_I) = \frac{\lambda}{2} \|\mathbf{W}_I \mathbf{X}_I - \mathbf{W}_I^0 \mathbf{X}_I\|_F^2$:

$$\mathbf{W}_{SD} = \mathbf{W}_I^0\left(\mathbf{I} - \frac{1}{1+\lambda}\mathcal{P}_I\right) + \frac{1}{1+\lambda}\mathbf{Y}_{FT}\mathbf{X}_I^\top(\mathbf{X}_I \mathbf{X}_I^\top)^+$$

*Here,* $^+$ *denotes the Moore-Penrose pseudoinverse and* $\lambda > 0$ *is the regularization parameter.*

**Geometric Interpretation:** As visualized in Figure 2, Direct Finetuning discards pretrained knowledge within the finetuning data subspace, replacing it with the new task solution, while preserving orthogonal components. $L_2$ regularization shrinks the entire solution towards the pretrained weights, leading to a complex, non-surgical blend.

**Self-Distillation achieves a nuanced trade-off**: it preserves pretrained knowledge in the subspace orthogonal to the finetuning data ($\mathbf{W}_I^0(\mathbf{I} - \mathcal{P}_I)$), and within the task-relevant subspace, it computes a convex combination of the projected

pretrained weights and the optimal solution for the new task. This enables control over knowledge retention and adaptation (further details in §C.4).

### 3.4. Dynamic Self-Distillation with a WMA Teacher

**Why static SD is biased.** Restricted to the task subspace, the static-SD solution in Theorem 3.2 is the convex combination $\mathbf{W}_{SD}\mathcal{P}_I = \frac{\lambda}{1+\lambda}\mathbf{W}_I^0\mathcal{P}_I + \frac{1}{1+\lambda}\mathbf{W}_{FT}^\star$, where $\mathbf{W}_{FT}^\star = \mathbf{Y}_{FT}\mathbf{X}_I^\top(\mathbf{X}_I \mathbf{X}_I^\top)^+$ is the minimum-norm task solution. Because the anchor is *fixed* at $\mathbf{W}_I^0$, for any finite $\lambda > 0$ the solution stays *offset* from $\mathbf{W}_{FT}^\star$ in the task subspace by exactly $\frac{\lambda}{1+\lambda}(\mathbf{W}_I^0\mathcal{P}_I - \mathbf{W}_{FT}^\star)$, an anchor bias that cannot be removed by tuning $\lambda$ alone (smaller $\lambda$ reduces the bias but weakens orthogonal preservation; larger $\lambda$ does the reverse). Geometrically (Figure 2), static SD lies on the segment between $\mathbf{W}_I^0\mathcal{P}_I$ and $\mathbf{W}_{FT}^\star$ rather than reaching the task optimum.

**Intuition: static SD vs. EMA vs. WMA.** The teacher choice controls *where* the regularizer is anchored along the trajectory: *Static SD* fixes the anchor at the start ($\mathbf{W}_I^0$) and is biased toward initialization; an *EMA teacher* stays near the current student state, so its regularizing signal vanishes as training converges (precisely when OOD robustness is most fragile); a *WMA teacher* stays near a *trajectory-weighted consensus* of the optimization path: it remembers the start but adapts over time, and with a suitable kernel retains meaningful mass at *both* ends of training. WMA thus addresses two limitations at once: (i) the static-SD anchor bias and (ii) the EMA collapse of the teacher–student gap. We instantiate it as a *dynamic* teacher that evolves as a WMA of the student's trajectory, illustrated at the system level in Figure 1, with the formal definition below.

**Definition 3.3** (WMA Teacher). The WMA teacher $\mathbf{W}_{\text{Teacher}}^t$ averages student states $\mathbf{W}_I^k$ over time $k = 0, \ldots, t$, weighted by a kernel $\kappa(\tau_k)$ on normalized time $\tau_k = \frac{k+c_1}{T+c_2} \in (0,1)$. The offsets $c_1, c_2 > 0$ keep $\tau_k$ strictly inside $(0,1)$ (avoiding $\tau_0 = 0$ and $\tau_T = 1$), which is required for kernels such as $\text{Beta}(0.5, 0.5)$ that diverge at the endpoints; we use $c_1 = 0.5$, $c_2 = 1$ in all experiments. The online recursion is:

$$\omega_t = \frac{\kappa(\tau_t)}{\sum_{j=0}^t \kappa(\tau_j)},$$
$$\mathbf{W}_{\text{Teacher}}^t = (1 - \omega_t)\mathbf{W}_{\text{Teacher}}^{t-1} + \omega_t \mathbf{W}_I^t,$$
$$\mathbf{W}_{\text{Teacher}}^0 = \mathbf{W}_I^0.$$

**Persistent Regularization and Bias-Free Convergence:** Unlike an Exponential Moving Average (EMA) teacher, whose regularizing influence vanishes as it converges to the student, the WMA teacher (especially with a U-shaped kernel like Beta(0.5,0.5)) maintains a persistent regularizing force over finite training horizons (see §C.5.2). This force continuously pulls the student towards its robust pretrained initialization. We prove that this adaptive anchoring

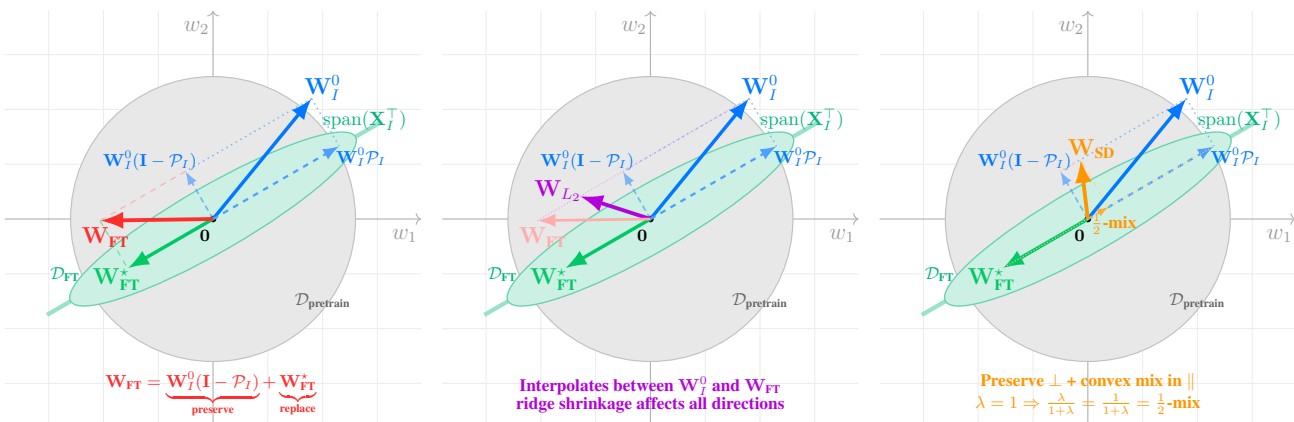

*(a)* **Direct FT:** Preserves orthogonal, replaces parallel component.

*(b)* **L2-SP:** Blends all directions, no structure preservation.

*(c)* **SD:** Preserves orthogonal, mixes parallel components.

*Figure 2.* **Geometric interpretation of finetuning strategies in 2D weight space.** The green line represents $\mathrm{span}(\mathbf{X}_I^\top)$, the subspace where finetuning data concentrates. Starting from pretrained weights $\mathbf{W}_I^0$ (blue), each method combines the orthogonal component $\mathbf{W}_I^0(\mathbf{I} - \mathcal{P}_I)$ and the new task solution $\mathbf{W}_{\mathrm{FT}}^\star = \mathbf{Y}_{\mathrm{FT}}\mathbf{X}_I^\top(\mathbf{X}_I\mathbf{X}_I^\top)^+$ (green) differently: **(a)** Direct FT preserves the orthogonal component and replaces the parallel component entirely; **(b)** L2-SP creates a global blend without clean structural decomposition; **(c)** Static Self-Distillation preserves the orthogonal component and forms a convex combination of the parallel components (shown with $\lambda = 1$ giving equal weighting).

achieves bias-free convergence to the task-optimal solution within the finetuning subspace:

**Theorem 3.4** (Bias-Free Convergence in the Task Subspace). *Let* $\mathbf{W}_{FT}^\star = \mathbf{Y}_{FT}\mathbf{X}_I^\top(\mathbf{X}_I\mathbf{X}_I^\top)^+$ *denote the* minimum-norm task solution *(i.e., the minimum-Frobenius-norm minimizer of equation 2). The WMA teacher's projection onto the task subspace converges to* $\mathbf{W}_{FT}^\star\mathcal{P}_I$*, and consequently, the student's projection also converges to* $\mathbf{W}_{FT}^\star\mathcal{P}_I$.

We use the name *minimum-norm task solution* for $\mathbf{W}_{\mathrm{FT}}^\star$ to distinguish it from the Direct FT solution in Theorem 3.2, which additionally retains the orthogonal component $\mathbf{W}_I^0(\mathbf{I} - \mathcal{P}_I)$: $\mathbf{W}_{\mathrm{FT}}^\star$ is the task-subspace target a preservation-aware method should reach *within* $\mathrm{range}(\mathbf{X}_I)$. The theorem shows that dynamic teachers eliminate the static anchor bias while preserving orthogonal knowledge (formal proof in §C.5).

## 4. Methodology: TRACER

Guided by these geometric insights and the proven benefits of a WMA teacher in the above section, we propose **TRACER**, a novel finetuning method for multimodal models. TRACER combines the standard symmetric InfoNCE loss with dynamic self-distillation guided by a WMA teacher, as illustrated in Figure 1.

The total training objective for TRACER is:

$$\mathcal{L}_{\mathrm{TRACER}} = \mathcal{L}_{\mathrm{MMCL}} + \lambda_{\mathrm{SD}}\,\mathcal{L}_{\mathrm{SD\text{-}WMA}}. \tag{3}$$

**Multi-Modal Contrastive Loss ($\mathcal{L}_{\mathrm{MMCL}}$):** This is the primary finetuning loss, typically a symmetric InfoNCE

objective. In our implementation, we also include a cross-Frobenius regularizer to prevent embedding collapse (standard CLIP finetuning recipe). This component drives the student model to learn new task-specific alignments.

**Dynamic Self-Distillation Loss ($\mathcal{L}_{\mathrm{SD\text{-}WMA}}$):** This is the core mechanism for robust knowledge preservation and adaptive mixing. It ensures the student retains generalizable features by learning from an evolving teacher model. As detailed in §C.6, $\mathcal{L}_{\mathrm{SD\text{-}WMA}}$ is a composite distillation loss that includes several perspectives: *(i) Feature Distillation (FD):* Directly aligns student and teacher embeddings. *(ii) Contrastive Relational Distillation (CRD):* Matches batch-wise similarity distributions between student and teacher. *(iii) Interactive Contrastive Learning (ICL):* Encourages student-teacher cross-modal alignment. *(iv) Cross Knowledge Distillation (Cross-KD):* Aligns cross-modal logits to transfer relational structure. This multi-perspective approach operationalizes the theoretical insight of preserving distinct aspects of pretrained knowledge.

**Weighted Moving Average (WMA) Teacher:** The teacher model is a central component of TRACER. Unlike an EMA teacher, which gradually collapses onto the student, our WMA teacher is a weighted average of the *entire* student trajectory up to time $t$, using a carefully chosen weighting kernel (e.g., a Beta kernel with $\beta_1 = \beta_2 = 0.5$ as shown in Figure 1 and detailed in §C.5.1). This ensures that early pretrained states retain a non-trivial contribution to the teacher over finite horizons, aligning with our trajectory-regularization motivation. This persistent regularization provides a continuous restoring force, preventing the student from over-specializing on spurious correlations in the finetuning data.

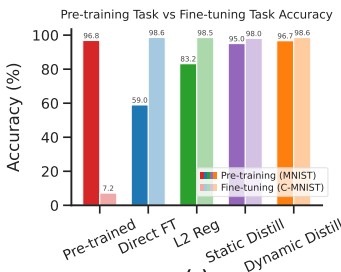 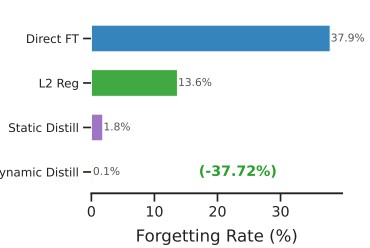 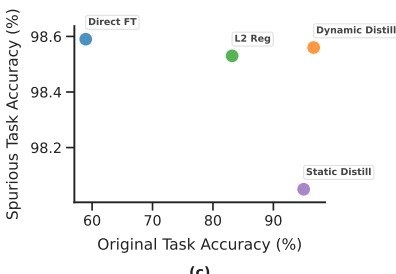

(a)             (b)             (c)

*Figure 3.* **Toy Experiment.** We compare a pretrained model against four finetuning methods on a finetuning task. **(a)** Performance on the original MNIST and new Colored MNIST task. All finetuning methods successfully learn the new task. Direct FT and L2 Reg suffer severe performance degradation (catastrophic forgetting). **(b)** Catastrophic forgetting rate, quantified as the percentage drop in accuracy on the original task. Self-distillation methods are more effective at preserving knowledge. **(c)** The performance trade-off between the original task ($x$-axis) and the spurious task ($y$-axis). Distillation methods achieve a much better trade-off, retaining high original task accuracy while mastering the new task.

# 5. Experiments

This section evaluates TRACER on ImageNet and natural distribution shifts, including a controlled toy study to validate theoretical predictions. We conduct comprehensive ablations across four axes (distillation components, strength, teacher update frequency, and Beta-kernel shape); extended protocols are in §B, loss definitions in §C.6, and teacher details in §C.5.1.

## 5.1. Synthetic Experiment

We design a controlled toy experiment with spurious correlations (Arjovsky et al., 2019) to validate our theory. The behaviors of Direct Finetuning, L2 Regularization, and Self-Distillation in a non-linear architecture align with our closed-form predictions.

### 5.1.1. EXPERIMENTAL SETUP

**Datasets.** We use two variants of the MNIST dataset (LeCun et al., 1998; Deng, 2012). (i) Original Pretraining Task: We create a multimodal version of MNIST, where each grayscale digit image is paired with a simple text description (e.g., an image of a '7' is paired with the text "the digit 7"). The model is pretrained on this dataset to learn robust, general-purpose representations for digit recognition. (ii) Finetuning Task: We create a dataset to introduce a spurious correlation. Images of digits 0-4 are colored red with 95% probability, while digits 5-9 are colored blue with 95% probability. This setup forces the model during finetuning to learn an easy-to-exploit but non-causal feature (color) to solve the new task, creating a direct conflict with the original digit recognition knowledge.

**Model Architecture.** We employ lightweight, non-linear models to show that our theory extends beyond the linear case. The architecture consists of a 'LightViT' (Dosovitskiy et al., 2021) image encoder and a 'LightTextTransformer' (Vaswani et al., 2017) text encoder. Both models project their inputs into a shared 128-dimensional embedding space,

where a standard InfoNCE contrastive loss is applied during pretraining.

**Pretraining.** The model is pretrained on MNIST using a contrastive objective for 10 epochs, achieving high accuracy on digit recognition but poor performance on the color-based task.

**Finetuning Strategies.** We finetune the pretrained image encoder on the ColoredMNIST (Arjovsky et al., 2019; Zhang et al., 2022a) task for 10 epochs while keeping the text encoder frozen, mirroring our theoretical setup. We compare the following methods: (i) *Pretrained:* The baseline model without any finetuning. (ii) *Direct Finetuning:* The image encoder is finetuned on the new task, as analyzed in §C.3. (iii) $L_2$ *Regularization:* We add a penalty term $\frac{\lambda}{2} \left\| \mathbf{W}_I - \mathbf{W}_I^0 \right\|_{\mathrm{F}}^2$ to the finetuning loss, corresponding to our analysis of $L_2$ regularization. (iv) *Static Distillation:* We use the initial pretrained model $\mathbf{W}_I^0$ as a fixed teacher and add a distillation loss term to the finetuning objective, as analyzed for $\mathbf{W}_{SD}$. (v) *Dynamic Distillation:* We use a teacher model whose weights are a moving average of the student's weights, corresponding to our analysis of the WMA teacher.

### 5.1.2. RESULTS AND DISCUSSION

**Analysis of Forgetting.** As predicted by our theory, Direct Finetuning exhibits severe catastrophic forgetting. It achieves near-perfect accuracy (98.5%) on the new color-based task by overwriting its original knowledge, causing its performance on the original MNIST test set to degrade from 96.8% to 59.0%, a forgetting rate of 37.9%. $L_2$ Regularization offers an improvement, but still forgets 13.6% of the original task's performance. In contrast, both Static and Dynamic Distillation demonstrate resilience to forgetting. They also master the new task but retain a larger portion of the original knowledge, with forgetting rates of only 1.8% and 0.1%, respectively. This result empirically supports our geometric interpretation: by interpolating between old and new knowledge within the task-relevant subspace while preserv-

ing knowledge in the orthogonal subspace, self-distillation methods achieve a better balance.

**The Performance Trade-off.** The scatter plot in Figure 3(c) visualizes this trade-off: distillation methods achieve a better Pareto frontier, with Dynamic Distillation finding a slightly better solution than its static counterpart, aligning with our theoretical analysis.

## 5.2. Main ImageNet Results and Ablations

We report the main ImageNet results and ablations below; detailed per-backbone tables follow.

*Table 1.* **ImageNet accuracy.** We report accuracy ($\uparrow$) on ImageNet and its distribution shift variants by finetuning CLIP ViT-B/16 with six methods. All values are averaged over three seeds with standard deviations shown as subscripts. In each column, the best value is bold and the second-best is underlined.

| Method | IN | IN-V2 | IN-R | IN-A | IN-S | ObjNet | Avg. |
|---|---|---|---|---|---|---|---|
| ZS | 68.33 | 61.93 | 77.71 | 49.95 | 48.26 | 54.17 | 58.39 |
| LP-FT | 82.44$_{\pm0.08}$ | 72.74$_{\pm0.18}$ | 72.81$_{\pm0.22}$ | 49.28$_{\pm0.31}$ | 50.31$_{\pm0.15}$ | 54.42$_{\pm0.14}$ | 59.91$_{\pm0.18}$ |
| FLYP | 82.72$_{\pm0.09}$ | 72.76$_{\pm0.21}$ | 71.32$_{\pm0.25}$ | 48.49$_{\pm0.35}$ | 49.87$_{\pm0.18}$ | 54.83$_{\pm0.16}$ | 59.45$_{\pm0.20}$ |
| Lipsum-FT | **83.32**$_{\pm0.05}$ | 73.57$_{\pm0.12}$ | 75.93$_{\pm0.14}$ | 49.87$_{\pm0.28}$ | 51.43$_{\pm0.12}$ | 54.35$_{\pm0.11}$ | 61.03$_{\pm0.14}$ |
| CaRot | 83.15$_{\pm0.06}$ | **74.08**$_{\pm0.14}$ | 77.74$_{\pm0.16}$ | 51.57$_{\pm0.24}$ | 52.68$_{\pm0.13}$ | 56.63$_{\pm0.12}$ | 62.54$_{\pm0.14}$ |
| TRACER | 82.76$_{\pm0.07}$ | **74.14**$_{\pm0.15}$ | **79.33**$_{\pm0.18}$ | **54.92**$_{\pm0.26}$ | **53.69**$_{\pm0.14}$ | **58.26**$_{\pm0.13}$ | **64.07**$_{\pm0.15}$ |

### 5.2.1. EXPERIMENTAL SETUP

**Objective.** Our experiments are designed to validate our theoretical claims and assess TRACER, as a practical implementation of our framework, on robust finetuning. We focus on evaluating both accuracy and calibration under distribution shifts.

**Datasets and Evaluation.** We use ImageNet-1K (IN) (Deng et al., 2009; Russakovsky et al., 2015) as our in-distribution (ID) downstream task. To measure OOD robustness, we evaluate all finetuned models on a standard suite of five distribution shift datasets: ImageNet-V2 (IN-V2) (Recht et al., 2019), ImageNet-Rendition (IN-R) (Hendrycks et al., 2021a), ImageNet-Adversarial (IN-A) (Hendrycks et al., 2021b), ImageNet-Sketch (IN-S) (Wang et al., 2019), and ObjectNet (Barbu et al., 2019). We report the average performance across these five datasets as "Avg. shifts" or "OOD".

**Baselines.** We compare TRACER against a comprehensive set of baselines, including zero-shot (ZS (Radford et al., 2021)), linear probing then finetuning (LP-FT (Kumar et al., 2022)), finetune-like-you-pretrain (FLYP (Goyal et al., 2023)), Lipsum-FT (Nam et al., 2024), and the recent robust finetuning method CaRot (Oh et al., 2024).

**Metrics.** We report top-1 accuracy and Expected Calibration Error (ECE), which measures the gap between predicted confidence and empirical accuracy (lower is better). We average across the five OOD datasets to summarize robustness.

**Implementation Details and Experimental Setup.** We

*Table 2.* **ImageNet Accuracy.** (except ObjectNet) with additional baselines. All values are averaged over three seeds.

| Method | IN$\uparrow$ | IN-V2$\uparrow$ | IN-R$\uparrow$ | IN-A$\uparrow$ | IN-S$\uparrow$ | Avg. shifts$\uparrow$ |
|---|---|---|---|---|---|---|
| ZS | 68.33 | 61.93 | 77.71 | 49.95 | 48.26 | 59.46 |
| Direct FT | 82.83$_{\pm0.10}$ | 72.57$_{\pm0.28}$ | 68.53$_{\pm0.32}$ | 39.23$_{\pm0.35}$ | 47.97$_{\pm0.22}$ | 57.08$_{\pm0.24}$ |
| L2-SP (Li et al., 2018) | 82.87$_{\pm0.09}$ | 72.63$_{\pm0.22}$ | 68.77$_{\pm0.24}$ | 39.73$_{\pm0.28}$ | 48.23$_{\pm0.15}$ | 57.34$_{\pm0.18}$ |
| Static SD (Hinton et al., 2015) | 82.07$_{\pm0.08}$ | 73.13$_{\pm0.26}$ | 72.87$_{\pm0.18}$ | 42.33$_{\pm0.38}$ | 49.87$_{\pm0.21}$ | 59.55$_{\pm0.22}$ |
| LP-FT (Kumar et al., 2022) | 82.14$_{\pm0.08}$ | 72.09$_{\pm0.20}$ | 70.44$_{\pm0.22}$ | 46.32$_{\pm0.30}$ | 48.65$_{\pm0.16}$ | 59.38$_{\pm0.18}$ |
| FLYP (Goyal et al., 2023) | 82.72$_{\pm0.09}$ | 72.76$_{\pm0.24}$ | 71.32$_{\pm0.26}$ | 48.49$_{\pm0.34}$ | 49.87$_{\pm0.19}$ | 60.61$_{\pm0.21}$ |
| CAR-FT (Mao et al., 2024) | 83.27$_{\pm0.06}$ | 74.03$_{\pm0.18}$ | 75.37$_{\pm0.28}$ | 49.53$_{\pm0.24}$ | 52.97$_{\pm0.20}$ | 62.98$_{\pm0.18}$ |
| Lipsum-FT (Nam et al., 2024) | 83.33$_{\pm0.05}$ | 73.57$_{\pm0.12}$ | 75.93$_{\pm0.14}$ | 49.87$_{\pm0.28}$ | 51.43$_{\pm0.12}$ | 62.70$_{\pm0.14}$ |
| Model Stock (Jang et al., 2024) | **84.07**$_{\pm0.07}$ | **74.83**$_{\pm0.16}$ | 71.77$_{\pm0.20}$ | 51.23$_{\pm0.30}$ | 51.77$_{\pm0.17}$ | 62.40$_{\pm0.18}$ |
| ARF (Han et al., 2024) | 82.73$_{\pm0.08}$ | 72.77$_{\pm0.19}$ | 75.63$_{\pm0.22}$ | 50.27$_{\pm0.28}$ | 51.83$_{\pm0.16}$ | 62.63$_{\pm0.17}$ |
| CaRot (Oh et al., 2024) | 83.16$_{\pm0.06}$ | 74.08$_{\pm0.14}$ | 77.74$_{\pm0.16}$ | 51.57$_{\pm0.22}$ | 52.74$_{\pm0.13}$ | 64.03$_{\pm0.14}$ |
| TRACER | 82.76$_{\pm0.07}$ | 74.12$_{\pm0.15}$ | **79.30**$_{\pm0.18}$ | **54.72**$_{\pm0.24}$ | **53.69**$_{\pm0.14}$ | **65.46**$_{\pm0.15}$ |

finetune CLIP variants on ImageNet-1K (IN) and evaluate on five OOD datasets: IN-V2, IN-R, IN-A, IN-S, and ObjectNet, following Taori et al. (2020). For all methods, we finetune for 10 epochs using the AdamW optimizer with a learning rate of $1 \times 10^{-5}$ and a weight decay of 0.01. The batch size is set to 224 for ViT-L/14 and 512 for ViT-B/16 and ResNet50. All experiments are run over three random seeds and we report mean and standard deviation. For TRACER, the WMA teacher uses a Beta$(0.5, 0.5)$ weighting kernel and combines symmetric InfoNCE with the composite SD loss (§C.6).

### 5.2.2. RESULTS AND ANALYSIS

**OOD accuracy and calibration on ViT-B/16.** As shown in Table 1, TRACER achieves strong OOD accuracy on ViT-B/16, particularly on the most challenging shifts (ObjectNet and IN-A). In Appendix Table 5, TRACER achieves the lowest average OOD ECE, indicating probabilistic reliability under shift. With additional baselines (Table 2), TRACER remains among the top OOD performers while staying competitive on IN. Additionally, the cross-backbone experiments (Table 3) show similar trends for RN50 and ViT-L/14.

**OOD degradation as the empirical signature of catastrophic forgetting.** We treat *OOD accuracy degradation* as the primary measurable symptom of catastrophic forgetting under finetuning: when a model retains less pretrained, broadly transferable structure, the loss shows up most clearly on inputs that differ from the finetuning distribution. Three results in our experiments support this view: (i) Direct FT drops *below the zero-shot baseline* on average OOD accuracy (Table 2: 57.08% vs. 59.46% ZS); (ii) the toy experiment (§5.1) measures explicit forgetting rates of 37.9% for Direct FT vs. 0.1% for dynamic SD; and (iii) the CKA/SVCCA analysis (Figure 4) shows that the deeper layers of Direct FT drift away from the pretrained representation, while TRACER keeps similarity $> 0.97$ across all layers. Together, these justify reading the OOD column of Tables 1–3 as a quantitative measure of how much pretrained knowledge each method preserves.

*Table 3*. **ImageNet accuracy and ECE across backbones**. We provide summarized results on CLIP RN50, ViT-B/16, and ViT-L/14, averaged over three seeds. The best and the second-best in each column are bold and underlined, respectively. OOD columns are highlighted to emphasize robustness. (See Table 1 and Appendix Table 5 for ViT-B/16, and Table 6 and 7 for details.)

| Method | RN50 | | | | ViT-B/16 | | | | ViT-L/14 | | | |
|---|---|---|---|---|---|---|---|---|---|---|---|---|
| | ID Acc.↑ | ID ECE↓ | OOD Acc.↑ | OOD ECE↓ | ID Acc.↑ | ID ECE↓ | OOD Acc.↑ | OOD ECE↓ | ID Acc.↑ | ID ECE↓ | OOD Acc.↑ | OOD ECE↓ |
| LP-FT | 76.25 | 0.1042 | 41.62 | 0.3274 | 82.44 | 0.051 | 59.91 | 0.147 | 84.74 | 0.1056 | 64.11 | 0.2521 |
| FLYP | 76.16 | 0.0516 | 42.70 | 0.2127 | 82.72 | 0.064 | 59.45 | 0.184 | 86.19 | 0.0729 | 71.44 | 0.1470 |
| CaRot | 76.12 | 0.0471 | 42.71 | 0.2109 | **83.15** | 0.047 | 62.54 | 0.079 | **86.95** | **0.0349** | 74.13 | 0.0737 |
| TRACER | **76.48** | **0.0470** | **42.73** | **0.1807** | 82.76 | **0.045** | **64.07** | **0.073** | 86.27 | 0.0507 | **75.32** | **0.0732** |

**Static SD vs. TRACER.** Table 2 isolates the contribution of the trajectory-regularized teacher on ViT-B/16. A static self-distillation anchor at $W_I^0$ already recovers a large portion of OOD accuracy over Direct FT (Avg. shifts $57.08\% \rightarrow 59.55\%$), confirming that distillation-based preservation is necessary. Static SD still leaves a substantial gap to TRACER ($59.55\% \rightarrow 65.46\%$, **+5.9**), largest where the pretrained representation is most informative and the static-SD anchor bias is most punishing: IN-R $72.87\% \rightarrow$ **79.30%** (+6.4), IN-A $42.33\% \rightarrow$ **54.72%** (+12.4). These are exactly the renditions/adversarial shifts where Theorem 3.4 predicts the WMA teacher's benefit: it removes the static-SD task-subspace bias while preserving the orthogonal directions that matter for unseen styles and natural adversarial examples.

**Ablation Studies.** Beyond the primary results, we conducted comprehensive ablation studies across *four axes* (detailed in §B) to thoroughly validate TRACER's design choices. **(1) Multi-perspective distillation** (Table 8): (a) *FD and CRD are the strongest single components*. (b) *The four components are complementary*: every pair including FD or CRD beats its singletons. (c) *All four together is best overall*: the full TRACER setting achieves the highest Avg. All; FD stabilizes features, CRD preserves relational structure, ICL enriches mutual information, and Cross-KD blends relational and interactive cues. **(2) Distillation strength** (Table 9): Sweeping $\lambda_{SD}$ from 0.1 to 10.0 reveals that moderate values ($\approx 1.0$–$2.0$) achieve the best ID-OOD trade-off, while higher values improve calibration at the cost of ID accuracy. **(3) Teacher update frequency** (Table 10): TRACER maintains stable OOD accuracy ($\sim 64.0$–$64.2\%$) across update frequencies from every step to every 2500 steps, eliminating the need for brittle scheduling required by CaRot (Oh et al., 2024). **(4) Beta-kernel shape** (Table 11): Arcsine-like weighting (Beta$(0.5, 0.5)$) proves most effective. These extensive ablations confirm that TRACER's design is both principled and robust.

**Empirical Validation of Geometric Preservation.** To verify our theoretical claim that TRACER preserves knowledge in the orthogonal subspace, we conduct a layer-wise representational similarity analysis using Centered Kernel Alignment (CKA) (Kornblith et al., 2019) on the CLIP ViT-B/16 image encoder (Figure 4). We extract feature maps from every layer (Patch Embeddings, Transformer Blocks 0–11,

and the Final Projection) on the ImageNet validation set and compare finetuned models against the pretrained model using CKA and SVCCA (Raghu et al., 2017). As shown in Figure 4, Direct FT exhibits a drop in similarity in the deeper layers (Blocks 6–11) relative to the pretrained model. This confirms that catastrophic forgetting manifests as a **Feature Distortion** of high-level semantic representations. In contrast, TRACER maintains near-perfect similarity ($> 0.97$) across all layers. This provides empirical evidence for our geometric interpretation: TRACER successfully anchors the optimization to the pretrained geometry, performing surgical updates that adapt to the task without overwriting robust feature extractors.

**Computational Efficiency and Complexity.** TRACER also offers efficiency advantages over CaRot (Oh et al., 2024), whose spectral regularization scales as $\mathcal{O}(d^3)$. TRACER's distillation operates on batch similarity matrices ($\mathcal{O}(B^2)$, $B \ll d$), and the WMA update matches standard EMA cost ($\mathcal{O}(P)$). As shown in Table 4, this reduces training time per epoch compared to CaRot.

*Table 4*. **Computational Efficiency.** Computational efficiency comparison per epoch on ImageNet-1K using CLIP ViT-B/16 on an NVIDIA H100 GPU.

| Method | Cost | Time / Epoch | Overhead | Avg. OOD Acc. |
|---|---|---|---|---|
| Direct FT | $\mathcal{O}(P)$ | $\sim 16$ min | $1.00\times$ | 57.08% |
| CaRot (Oh et al., 2024) | $\mathcal{O}(B^2 + d^3)$ | $\sim 29$ min | $1.81\times$ | 62.54% |
| TRACER (Ours) | $\mathcal{O}(B^2)$ | $\sim$ **22 min** | $1.38\times$ | **64.07%** |

**Teacher Dynamics and Regularization Strength.** Our theory posits that the WMA teacher in TRACER provides a more persistent regularizing signal than the EMA teacher used in methods like CaRot. To validate this, we track the KL divergence between teacher and student throughout training on ImageNet. As shown in Figure 5, for the EMA teacher, the KL decays steadily, indicating that the teacher is rapidly collapsing onto the student and its regularizing influence is diminishing. In contrast, the WMA teacher maintains a higher and more stable KL throughout the entire training process. This sustained divergence supports the view that the WMA teacher provides a persistent "restoring force," as predicted by our analysis in §C.5.2. This helps prevent the student from converging to a narrow task-specific minimum and reflects the persistent regularization strength required for robust generalization.

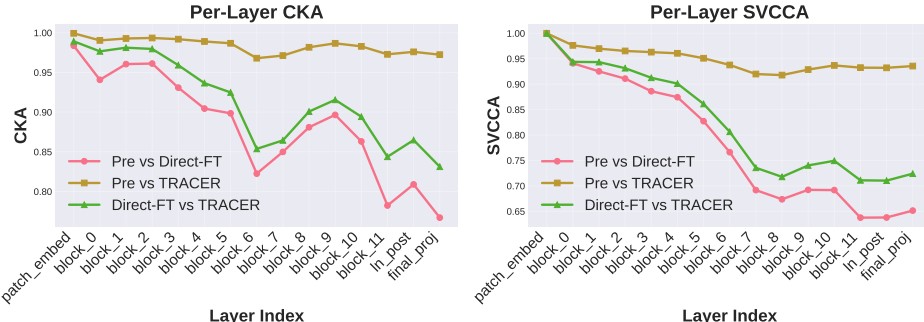

*Figure 4.* **Layer-wise Representational Similarity.** We compare the internal representations of the Pretrained model against `Direct FT` and `TRACER` using CKA (left) and SVCCA (right) across all layers of the CLIP ViT-B/16 image encoder. `TRACER` (gold) preserves the geometric structure of the pretrained knowledge significantly better than `Direct FT` (pink), particularly in deeper layers.

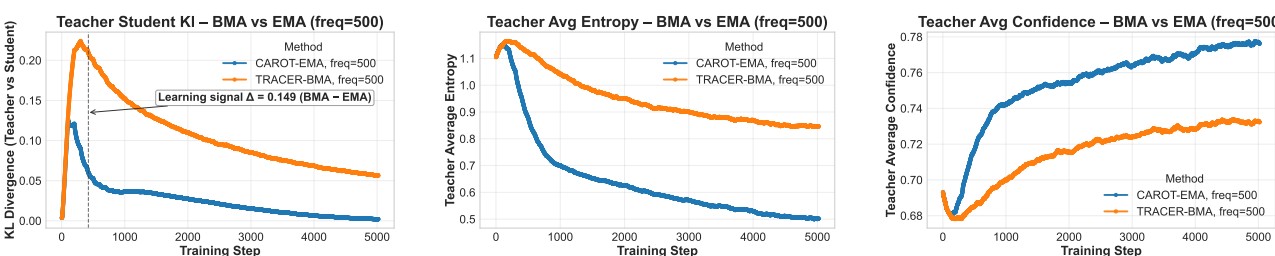

*Figure 5.* **Teacher–Student Knowledge Gap During Training.** Compared to the EMA teacher (blue), which shows rapidly vanishing KL divergence and thus a weakening regularization signal (left), the WMA teacher (orange) sustains a higher and more stable KL gap. This stability is supported by higher teacher entropy (middle) and moderated confidence (right), preventing overfitting. Together, these trends confirm that WMA provides a stronger and more persistent self-distillation signal than EMA.

**Algorithmic Simplicity.** While EMA-based methods often require complex, sparse update schedules (e.g., updating only every 500 steps with linear warmup and careful momentum tuning) to prevent collapse, TRACER is robust to update frequency. As shown in Table 10, TRACER maintains consistent performance ($\sim 64.0 - 64.2\%$ OOD accuracy) whether the teacher is updated every step or every 2500 steps, reducing the need for brittle hyperparameter tuning. Figure 6 illustrates this failure mode in prior methods: without careful tuning, the EMA teacher in CaRot collapses immediately *when updated at every step*, whereas TRACER's WMA teacher remains stable even under a dense update schedule.

## 6. Conclusion

We proved that TRACER's trajectory-averaging WMA teacher, unlike its EMA counterpart, maintains a persistent regularizing force over finite training horizons. This force continuously anchors the model to its robust pretrained initialization, helping prevent overfitting and improving out-of-distribution performance. Our extensive ablation studies confirm that TRACER's design is both principled and robust to hyperparameter choices across distillation strength, update frequency, and kernel shape. Our work bridges the geometry of finetuning with the practical design of robust methods, and these principles motivate future extensions

to parameter-efficient methods, continual learning, random-feature analyses of the same trajectory-regularization idea, and larger vision–language backbones.

**Limitations and Scope.** Our empirical evaluation focuses on CLIP-style vision–language backbones and standard vision robustness benchmarks. We do not yet provide experiments on broader modalities or multimodal LLM settings, and our theoretical analysis is developed in the linearized image/text-encoder regime; we discuss concrete extensions to random-feature settings, to larger VLMs, and the relationship to prompt-based adaptation in §F.

## Impact Statement

This work develops methods for improving the robustness and calibration of finetuned multimodal models under distribution shift. Our primary motivation is enhancing the reliability of foundation models as they are deployed in real-world applications where inputs may differ from training data. Robust and calibrated models are particularly valuable in safety-critical domains such as medical imaging and autonomous systems, where overconfident predictions under shift can lead to harmful outcomes. Our method is purely defensive in nature: it mitigates catastrophic forgetting and improves generalization. We see no obvious pathways by which this work accelerates harmful capabilities.

## Acknowledgements

This research was conducted by the ARC Centre of Excellence for Automated Decision-Making and Society (CE200100005), and funded by the Australian Government through the Australian Research Council. This research was supported by The University of Melbourne's Research Computing Services and the Petascale Campus Initiative. We would like to thank Navid Akhavan Attar, Aryan Yazdan Parast, and Hugo Lyons Keenan for valuable discussions and feedback.

## Conflict of Interest Disclosure

The authors declare no financial conflicts of interest.

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

# Appendix

# A. Additional Related Work

### A.1. Contrastive Language-Image Pretraining

Initial advancements in contrastive learning between vision and language modalities were made by Virtex (Desai & Johnson, 2021), ICMLM (Sariyildiz et al., 2020), and ConVIRT (Zhang et al., 2022c). These early approaches laid the groundwork for later models like CLIP (Radford et al., 2021; Ilharco et al., 2021) and ALIGN (Jia et al., 2021), which scaled contrastive techniques to larger datasets and model architectures. Subsequent work explores improved cross-modal interaction and training recipes (Yuan et al., 2021; Yu et al., 2022; Fang et al., 2023). Following these, several open-weight contrastive models have been introduced to improve CLIP's performance and robustness (Sun et al., 2023; Zhai et al., 2023; Li et al., 2023b; Fang et al., 2024; Xu et al., 2024a; Schuhmann et al., 2022). For example, SigLIP (Zhai et al., 2023; Tschannen et al., 2025) modifies the contrastive loss by using a sigmoid function instead of softmax, and FLIP (Li et al., 2023c) integrates masking strategies to accelerate training.

### A.2. Theory of Contrastive Learning

A rich theoretical literature analyzes contrastive learning from first principles, characterizing when and why contrastive objectives recover useful features and class structure (Saunshi et al., 2019). The alignment–uniformity lens formalizes how pulling positives together while spreading embeddings uniformly on the sphere drives representation quality (Wang & Isola, 2020). For tractability, many works study *linearized* or simplified contrastive losses that replace log-exp with linear functions and show that their gradients align with those of standard objectives up to reweighting, enabling closed-form analysis and geometric insight (Ji et al., 2023; Tian, 2022; Nakada et al., 2023; Xue et al., 2024). This linearized viewpoint has proven effective in theoretical analyses across self-supervised contrastive learning (CL) (Ji et al., 2023; HaoChen et al., 2022; HaoChen & Ma, 2023; Shen et al., 2022a), multimodal contrastive learning (MMCL) (Nakada et al., 2023), non-contrastive methods (Liu et al., 2022), and supervised CL (Xue et al., 2023). Complementing these results, large-scale empirical studies suggest that many design choices of popular losses (e.g., log-exp, cosine similarity) are not essential for effective representation learning (Garrido et al., 2023).

### A.3. Finetuning, Forgetting, and Regularization

Catastrophic forgetting, adapting to new data at the expense of prior knowledge, has long been recognized as a central challenge in sequential and transfer learning (McCloskey & Cohen, 1989; French, 1999). Mitigation strategies include: (i) regularization, which constrains parameter updates via importance penalties or output consistency (Kirkpatrick et al., 2017; Zenke et al., 2017; Li & Hoiem, 2018); (ii) replay, which mixes current data with stored or synthesized memories (Robins, 1995; Rebuffi et al., 2017; Aljundi et al., 2019); and (iii) architectural growth, which expands capacity and distills across modules (Rusu et al., 2016; Yan et al., 2021; Wang et al., 2022a). L2-SP (Li et al., 2018) tethers the solution to the pretrained initialization via weight-space regularization, while output-space regularizers distill prior behaviors during adaptation (Li & Hoiem, 2018). Additionally, parameter-efficient finetuning methods such as adapters (Houlsby et al., 2019) and prefix tuning (Li & Liang, 2021) enable task adaptation without full model updates, thus mitigating forgetting. Among these, Low-Rank Adaptation (LoRA) (Hu et al., 2022) has gained prominence for finetuning large language models by injecting trainable low-rank matrices into existing weights, achieving competitive performance with reduced parameter updates and minimal forgetting. Further work explores functional regularization (Titsias et al., 2020) and knowledge-preserving contrastive losses (Jung et al., 2020) to encourage feature stability. As model sizes grow, scalable and minimally invasive adaptation techniques, balancing plasticity and stability, remain critical to continual and transfer learning paradigms.

### A.4. Robust Finetuning of CLIP

Robustness evaluates how well models maintain performance under distribution shifts, which can include synthetic corruptions (Hendrycks & Dietterich, 2019) as well as real-world variations in viewpoint, style, and time (Barbu et al., 2019; Hendrycks et al., 2021a; Wang et al., 2019; Recht et al., 2019). A standard protocol for evaluating CLIP-like models, proposed by (Taori et al., 2020), involves finetuning on ImageNet and measuring transfer performance on a suite of realistic OOD sets (ImageNet-V2, -A, -R, -Sketch, and ObjectNet), which is now standard practice. This evaluation highlights a central challenge: naive finetuning methods like Linear Probing (LP), which only trains a classification head, or **Direct Full finetuning**, which updates all parameters, often create a trade-off between in-distribution (ID) performance and OOD

robustness. To address this, a diverse array of robust finetuning techniques has been developed. A prominent line of work involves post-hoc averaging or interpolating model weights. For instance, **WiSE-FT** (Wortsman et al., 2022b) averages the weights of the zero-shot and a fully finetuned model, while **Model Soup** (Wortsman et al., 2022a) averages the weights of multiple models found through a hyperparameter search. This concept is extended by **Model Stock** (Jang et al., 2024), which efficiently builds and averages a diverse set of minimally adapted models. Other post-hoc methods include **TPGM** (Tian et al., 2023a) and its efficient successor **Fast TPGM** (Tian et al., 2023b), which project finetuned weights back towards the initial weights, and **DaWin** (Oh et al., 2025), which introduces a training-free, dynamic interpolation where the mixing coefficient is decided on a per-sample basis using predictive entropy.

Beyond post-hoc modifications, many methods introduce regularization during the finetuning process itself. These can constrain the model in weight-space, such as **L2-SP** (Li et al., 2018) which penalizes weight deviation, or by maintaining an **EMA** of model parameters to find smoother, more robust solutions. Others operate in the output-space, where **Knowledge Distillation (KD)** (Hinton et al., 2015) aligns the student's predictions with the robust zero-shot teacher. A particularly relevant strategy for Vision-Language Models is using the text modality for guidance. This includes continuing contrastive learning with supervised image-text pairs as in Finetune-Like-You-Pretrain (**FLYP**) (Goyal et al., 2023), aligning with fixed context-specific prompts in **CAR-FT** (Mao et al., 2024), regularizing the model's energy function using random texts to preserve broad semantic alignment in **Lipsum-FT** (Nam et al., 2024), or improving discrimination with both positive and negative prompts as in **CLIPood** (Shu et al., 2023). Alternative strategies modify the training pipeline, such as the two-stage **LP-FT** approach (Kumar et al., 2022) which first finds a good head via linear probing before full finetuning. More advanced methods like **CaRot** (Oh et al., 2024) aim to simultaneously improve OOD accuracy and confidence calibration through a principled combination of contrastive learning and novel regularization terms.

### A.5. Knowledge Distillation and Self-Distillation

Knowledge Distillation (KD) was initially introduced for compression, where a smaller student learns from a larger teacher's outputs (Hinton et al., 2015). The same principle underpins continual and transfer learning, where a pretrained model guides finetuning to preserve capabilities, often termed Learning without Forgetting (LwF) (Li & Hoiem, 2018). Self-distillation (SD) is a special case where the model learns from its own initial state (Zhang et al., 2019; Mobahi et al., 2020). Beyond single-modality SD, multimodal KD aligns internal signals and outputs to preserve cross-modal structure (Fang et al., 2021; Wang et al., 2022b; Li et al., 2023a; Liang et al., 2023; Li et al., 2024), with recent work demonstrating effective CLIP distillation via affinity matching and weight inheritance (Wu et al., 2023; Yang et al., 2024a).

### A.6. Dynamic Teachers, Weight Averaging, and Mode Connectivity

Temporal ensembling and EMA teachers stabilize training and improve targets (Laine & Aila, 2017; Tarvainen & Valpola, 2017), and they underpin momentum-encoder methods in self-supervised learning (He et al., 2020; Grill et al., 2020; Caron et al., 2021). Separately, model averaging and linear mode connectivity suggest that interpolations and averages often lie in flat, low-loss regions and improve robustness (Izmailov et al., 2018; Frankle & Carbin, 2019). Wise-FT leverages interpolation between pretrained and finetuned weights to strengthen OOD performance (Wortsman et al., 2022b;a).

*Table 5.* **ImageNet calibration (ECE) on ViT-B/16.** We report ECE (↓) on ImageNet and its distribution shift variants. In each column, the best value is bold and the second-best is underlined.

| Method | IN | IN-V2 | IN-R | IN-A | IN-S | ObjNet | Avg. |
|---|---|---|---|---|---|---|---|
| ZS | 0.057 | 0.055 | 0.054 | **0.097** | 0.085 | **0.078** | 0.074 |
| LP-FT | 0.051 | 0.089 | 0.061 | 0.205 | 0.166 | 0.212 | 0.147 |
| FLYP | 0.064 | 0.117 | 0.097 | 0.244 | 0.220 | 0.238 | 0.184 |
| Lipsum-FT | **0.038** | 0.052 | 0.043 | 0.129 | 0.102 | 0.132 | 0.091 |
| CaRot | 0.047 | **0.037** | 0.058 | 0.124 | **0.070** | 0.108 | 0.079 |
| TRACER (Ours) | 0.045 | 0.039 | **0.041** | 0.104 | 0.078 | 0.103 | **0.073** |

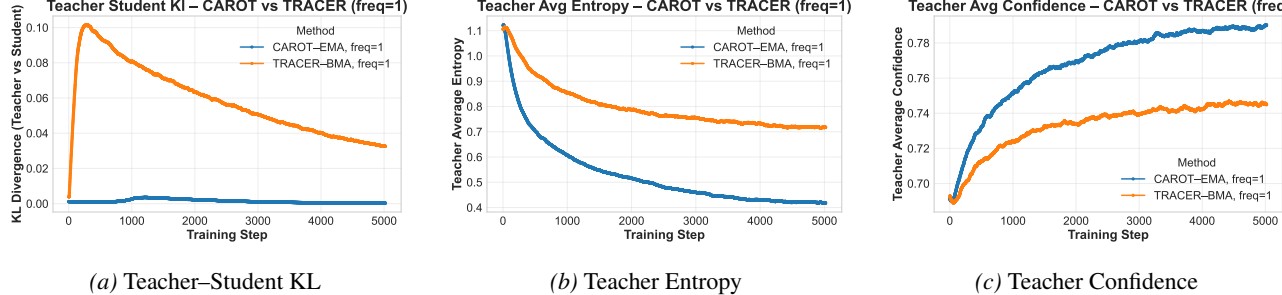

*(a)* Teacher–Student KL          *(b)* Teacher Entropy          *(c)* Teacher Confidence

*Figure 6.* **Comparison of Teacher Dynamics (Update Frequency = 1).** We track the evolution of the teacher model for CaRot (EMA) and TRACER (WMA) when updated at every step. The EMA teacher (blue) rapidly collapses onto the student (KL → 0), losing its regularizing capability. The WMA teacher (orange) maintains a persistent, stable gap, providing continuous regularization without needing brittle update schedules.

# B. Additional Experiments and Ablations

**Experimental Protocol.** We use the same seeds, and hyperparameter configurations as in the main experiments, varying only the stated factor per ablation.

**Additional Experimental Results.** To further demonstrate the generalizability of our method, we present results using the CLIP RN50 and ViT-L/14 backbones. A summary of these experiments, including ViT-B/16 for comparison, is provided in Table 3, with detailed results reported below.

*Table 6.* **ImageNet results on CLIP ResNet50**

| Method | IN | IN-V2 | IN-R | IN-A | IN-S | ObjectNet | Avg. shifts |
|---|---|---|---|---|---|---|---|
| | | Acc.↑ | | | | | |
| ZS | 59.83 | 52.90 | 60.72 | 23.25 | 35.45 | 40.27 | 42.52 |
| FT | 76.21 | 64.87 | 50.66 | 18.11 | 33.90 | 42.32 | 41.97 |
| LP-FT | 76.25 | 64.48 | 49.55 | 18.60 | 33.33 | 42.13 | 41.62 |
| FLYP | 76.16 | 65.10 | 51.55 | 20.08 | 34.24 | 42.53 | 42.70 |
| CaRot | 76.12 | 65.36 | 52.16 | 19.32 | 34.05 | 42.67 | 42.71 |
| TRACER (Ours) | 76.48 | 65.58 | 51.54 | 19.52 | 34.34 | 42.66 | 42.73 |
| | | ECE↓ | | | | | |
| ZS | 0.0624 | 0.0559 | 0.0530 | 0.2048 | 0.0740 | 0.0899 | 0.0955 |
| FT | 0.0983 | 0.1623 | 0.1860 | 0.4692 | 0.2824 | 0.3023 | 0.2804 |
| LP-FT | 0.1042 | 0.1759 | 0.2709 | 0.5184 | 0.3520 | 0.3197 | 0.3274 |
| FLYP | 0.0516 | 0.0872 | 0.1439 | 0.3872 | 0.2021 | 0.2432 | 0.2127 |
| CaRot | 0.0471 | 0.0601 | 0.0948 | 0.3435 | 0.3435 | 0.2127 | 0.2109 |
| TRACER (Ours) | 0.0470 | 0.0564 | 0.1176 | 0.3456 | 0.1741 | 0.2097 | 0.1807 |

*Table 7.* **ImageNet results on CLIP ViT-L/14**

| Method | IN | | IN-V2 | IN-R | IN-A | IN-S | ObjectNet | Avg. shifts |
|---|---|---|---|---|---|---|---|---|
| | | | | | Acc.↑ | | | |
| ZS | 75.55 | | 69.85 | 87.85 | 70.76 | 59.61 | 66.59 | 70.93 |
| FT | 84.74 | | 75.32 | 75.36 | 55.65 | 54.44 | 59.76 | 64.11 |
| LP-FT | 85.26 | | 76.76 | 80.21 | 55.95 | 56.84 | 60.12 | 65.98 |
| FLYP | 86.19 | | 78.21 | 83.81 | 68.85 | 60.20 | 66.15 | 71.44 |
| CaRot | 86.95 | | 79.28 | 87.96 | 72.68 | 62.66 | 68.05 | 74.13 |
| TRACER (Ours) | 86.27 | | 78.54 | 89.70 | 74.87 | 63.71 | 69.76 | 75.32 |
| | | | | | ECE↓ | | | |
| ZS | 0.0590 | | 0.0686 | 0.0339 | 0.0640 | 0.1037 | 0.0852 | 0.0711 |
| FT | 0.1056 | | 0.1741 | 0.1613 | 0.3151 | 0.3234 | 0.2865 | 0.2521 |
| LP-FT | 0.0993 | | 0.1531 | 0.0872 | 0.2593 | 0.2613 | 0.2572 | 0.2036 |
| FLYP | 0.0729 | | 0.1219 | 0.0621 | 0.1443 | 0.2164 | 0.1903 | 0.1470 |
| CaRot | 0.0349 | | 0.0634 | 0.0353 | 0.0732 | 0.0914 | 0.1051 | 0.0737 |
| TRACER (Ours) | 0.0507 | | 0.0581 | 0.0442 | 0.0665 | 0.1052 | 0.0918 | 0.0732 |

**Ablation 1: Multi-perspective distillation.** The ablation study on multi-perspective distillation (Table 8) quantifies the contribution of each perspective to out-of-distribution (OOD) accuracy and calibration. Results show that CRD and FD emerge as the strongest individual components for OOD accuracy and ECE, respectively, while combining all four perspectives yields the best overall performance and remains among the top performers on OOD metrics. These findings highlight the complementary nature of the terms: FD stabilizes features, CRD preserves batch-level relational structure, ICL enriches mutual information in the teacher's space, and CrossKD blends relational and interactive cues.

*Table 8.* Ablation of TRACER components across ImageNet (IN) and distribution shifts. For each setting (row), accuracy (Acc.↑) is reported in %, and expected calibration error (ECE↓) in $[0, 1]$. OOD Avg. is the mean over {IN-V2, IN-R, IN-A, IN-S, ObjectNet}. Method names encode the presence of losses ($\mathcal{L}_{FD}, \mathcal{L}_{CrossKD}, \mathcal{L}_{ICL}, \mathcal{L}_{CRD}$) as ✓ or −. Rows with only one loss term active are in gray.

| $\mathcal{L}_{FD}$ | $\mathcal{L}_{CrossKD}$ | $\mathcal{L}_{ICL}$ | $\mathcal{L}_{CRD}$ | IN | IN-V2 | IN-R | IN-A | IN-S | ObjectNet | Avg. shifts | Avg. All |
|---|---|---|---|---|---|---|---|---|---|---|---|
| | | | | | | | Acc.↑ | | | | |
| − | − | − | − | 82.69 | 72.73 | 71.35 | 48.52 | 49.84 | 54.86 | 59.40 | 63.33 |
| − | − | − | ✓ | 83.17 | 74.29 | 77.75 | 53.09 | 53.03 | 57.46 | 63.12 | 66.47 |
| − | − | ✓ | − | 82.50 | 73.13 | 72.12 | 49.23 | 50.03 | 55.26 | 59.95 | 63.71 |
| − | − | ✓ | ✓ | 83.23 | 74.31 | 76.50 | 52.39 | 52.53 | 57.00 | 62.55 | 65.99 |
| − | ✓ | − | − | 83.19 | 74.04 | 74.67 | 50.65 | 51.39 | 56.40 | 61.43 | 65.06 |
| − | ✓ | − | ✓ | 83.08 | 74.40 | 78.68 | 53.53 | 53.37 | 57.45 | 63.49 | 66.75 |
| − | ✓ | ✓ | − | 83.03 | 73.95 | 74.11 | 50.60 | 51.28 | 55.77 | 61.14 | 64.79 |
| − | ✓ | ✓ | ✓ | **83.27** | 74.48 | 77.60 | 52.76 | 52.94 | 57.28 | 63.01 | 66.39 |
| ✓ | − | − | − | 83.06 | 74.16 | 78.14 | 54.39 | 53.14 | 57.79 | 63.52 | 66.78 |
| ✓ | − | − | ✓ | 82.45 | 73.91 | 79.67 | 54.88 | 53.93 | 58.02 | **64.08** | 67.14 |
| ✓ | − | ✓ | − | 83.08 | 74.39 | 77.59 | 53.84 | 53.10 | 57.59 | 63.30 | 66.60 |
| ✓ | − | ✓ | ✓ | 82.92 | 74.40 | 79.21 | 54.61 | 53.75 | 57.99 | 63.99 | 67.15 |
| ✓ | ✓ | − | − | 83.01 | 74.21 | 78.91 | 54.09 | 53.28 | 58.04 | 63.71 | 66.92 |
| ✓ | ✓ | − | ✓ | 82.27 | 73.71 | 79.81 | 54.65 | 53.82 | 58.14 | 64.03 | 67.07 |
| ✓ | ✓ | ✓ | − | 83.06 | 74.38 | 78.38 | 54.07 | 53.25 | 57.86 | 63.59 | 66.83 |
| ✓ | ✓ | ✓ | ✓ | 82.81 | 73.94 | 79.55 | 54.83 | 53.96 | 58.02 | 64.06 | **67.19** |

| $\mathcal{L}_{FD}$ | $\mathcal{L}_{CrossKD}$ | $\mathcal{L}_{ICL}$ | $\mathcal{L}_{CRD}$ | IN | IN-V2 | IN-R | IN-A | IN-S | ObjectNet | Avg. shifts | Avg. All |
|---|---|---|---|---|---|---|---|---|---|---|---|
| | | | | | | | ECE.↓ | | | | |
| − | − | − | − | 0.0635 | 0.1171 | 0.0967 | 0.2435 | 0.2200 | 0.2383 | 0.1836 | 0.1632 |
| − | − | − | ✓ | 0.0415 | 0.0412 | 0.0413 | 0.1328 | 0.0860 | 0.1211 | 0.0845 | 0.0773 |
| − | − | ✓ | − | 0.0585 | 0.1000 | 0.0817 | 0.2117 | 0.1974 | 0.2168 | 0.1615 | 0.1444 |
| − | − | ✓ | ✓ | **0.0393** | 0.0523 | 0.0429 | 0.1534 | 0.1111 | 0.1441 | 0.1008 | 0.0905 |
| − | ✓ | − | − | 0.0483 | 0.0830 | 0.0662 | 0.2007 | 0.1660 | 0.1918 | 0.1415 | 0.1260 |
| − | ✓ | − | ✓ | 0.0453 | 0.0374 | 0.0434 | 0.1141 | 0.0753 | 0.1096 | 0.0760 | 0.0709 |
| − | ✓ | ✓ | − | 0.0507 | 0.0824 | 0.0691 | 0.2007 | 0.1684 | 0.1968 | 0.1435 | 0.1280 |
| − | ✓ | ✓ | ✓ | 0.0401 | 0.0442 | 0.0392 | 0.1345 | 0.0897 | 0.1260 | 0.0867 | 0.0790 |
| ✓ | − | − | − | 0.0430 | 0.0674 | 0.0479 | 0.1592 | 0.1383 | 0.1661 | 0.1158 | 0.1037 |
| ✓ | − | − | ✓ | 0.0474 | 0.0380 | 0.0455 | 0.1034 | 0.0720 | 0.0975 | 0.0713 | **0.0673** |
| ✓ | − | ✓ | − | 0.0436 | 0.0684 | 0.0482 | 0.1632 | 0.1384 | 0.1691 | 0.1175 | 0.1052 |
| ✓ | − | ✓ | ✓ | 0.0419 | 0.0454 | 0.0397 | 0.1157 | 0.0853 | 0.1162 | 0.0805 | 0.0740 |
| ✓ | ✓ | − | − | 0.0399 | 0.0537 | 0.0398 | 0.1340 | 0.1082 | 0.1374 | 0.0946 | 0.0855 |
| ✓ | ✓ | − | ✓ | 0.0531 | 0.0404 | 0.0500 | 0.0936 | 0.0703 | 0.0888 | **0.0686** | 0.0660 |
| ✓ | ✓ | ✓ | − | 0.0402 | 0.0572 | 0.0418 | 0.1420 | 0.1176 | 0.1499 | 0.1017 | 0.0915 |
| ✓ | ✓ | ✓ | ✓ | 0.0446 | 0.0416 | 0.0430 | 0.1027 | 0.0757 | 0.1054 | 0.0737 | 0.0688 |

**Ablation 2: Distillation strength $\lambda_{SD}$.** The ablation on distillation strength $\lambda_{SD}$ (Table 9) examines the balance between teacher influence and task adaptation. We sweep $\lambda_{SD} \in \{0.1, 0.2, 0.3, 0.4, 0.5, 0.7, 1.0, 1.2, 1.5, 2.0, 3.0, 4.0, 5.0, 10.0\}$. Results indicate that moderate values of $\lambda_{SD}$ ($\approx$ 1.0–2.0) achieve the best OOD accuracy, while larger values improve calibration by lowering ECE but slightly reduce in-distribution (ID) accuracy. This aligns with our theory that stronger distillation enhances calibration through teacher anchoring, whereas moderate strength provides the optimal trade-off between adaptation and preservation for OOD performance.

*Table 9.* Ablation of distillation coefficient $\lambda_{SD}$ across ImageNet (IN) and distribution shifts. For each setting (row), accuracy (Acc.↑) is reported in %, and expected calibration error (ECE↓) in [0, 1]. OOD Avg. is the mean over {IN-V2, IN-R, IN-A, IN-S, ObjectNet}.

| $\lambda_{SD}$ | IN | IN-V2 | IN-R | IN-A | IN-S | ObjectNet | OOD Avg. |
|---|---|---|---|---|---|---|---|
| | | | | Acc.↑ (%) | | | |
| 10.0 | 80.52 | 72.08 | 79.50 | 54.07 | 53.27 | 57.45 | 63.27 |
| 5.0 | 81.08 | 72.54 | 79.79 | 54.37 | 53.64 | 57.65 | 63.60 |
| 4.0 | 81.25 | 72.67 | 79.78 | 54.75 | 53.74 | 57.66 | 63.72 |
| 3.0 | 81.58 | 73.12 | 79.90 | 54.83 | 53.84 | 57.75 | 63.89 |
| 2.0 | 81.94 | 73.49 | **80.03** | 54.64 | **54.02** | 57.88 | 64.01 |
| 1.5 | 82.27 | 73.52 | 79.72 | **55.28** | 53.99 | 58.08 | **64.12** |
| 1.2 | 82.50 | 73.74 | 79.55 | 55.20 | 53.94 | **58.14** | 64.11 |
| 1.0 | 82.70 | 74.07 | 79.64 | 54.87 | 53.85 | 58.08 | 64.10 |
| 0.7 | 82.90 | 74.41 | 79.21 | 54.72 | 53.73 | 58.03 | 64.02 |
| 0.5 | 83.16 | 74.31 | 78.76 | 54.13 | 53.52 | 57.76 | 63.70 |
| 0.4 | 83.26 | 74.48 | 78.19 | 53.64 | 53.27 | 57.82 | 63.48 |
| 0.3 | 83.28 | **74.52** | 77.53 | 53.55 | 52.91 | 57.44 | 63.19 |
| 0.2 | **83.29** | 74.30 | 76.80 | 52.61 | 52.58 | 57.04 | 62.67 |
| 0.1 | 83.25 | 74.08 | 75.34 | 51.52 | 51.89 | 56.49 | 61.86 |

| $\lambda_{SD}$ | IN | IN-V2 | IN-R | IN-A | IN-S | ObjectNet | OOD Avg. |
|---|---|---|---|---|---|---|---|
| | | | | ECE↓ | | | |
| 10.0 | 0.0637 | 0.0475 | 0.0621 | 0.0839 | 0.0725 | 0.0795 | 0.0691 |
| 5.0 | 0.0631 | 0.0467 | 0.0600 | **0.0812** | 0.0700 | **0.0772** | **0.0670** |
| 4.0 | 0.0606 | 0.0457 | 0.0583 | 0.0817 | 0.0705 | 0.0801 | 0.0673 |
| 3.0 | 0.0590 | 0.0466 | 0.0566 | 0.0866 | 0.0701 | 0.0814 | 0.0683 |
| 2.0 | 0.0547 | 0.0422 | 0.0528 | 0.0885 | **0.0682** | 0.0863 | 0.0676 |
| 1.5 | 0.0511 | **0.0396** | 0.0490 | 0.0911 | 0.0707 | 0.0897 | 0.0680 |
| 1.2 | 0.0484 | 0.0408 | 0.0455 | 0.0963 | 0.0713 | 0.0957 | 0.0699 |
| 1.0 | 0.0465 | 0.0410 | 0.0445 | 0.1001 | 0.0742 | 0.1010 | 0.0722 |
| 0.7 | 0.0419 | 0.0445 | 0.0416 | 0.1120 | 0.0841 | 0.1144 | 0.0793 |
| 0.5 | 0.0409 | 0.0466 | **0.0390** | 0.1259 | 0.0953 | 0.1295 | 0.0873 |
| 0.4 | **0.0394** | 0.0502 | 0.0415 | 0.1353 | 0.1030 | 0.1363 | 0.0933 |
| 0.3 | 0.0397 | 0.0554 | 0.0424 | 0.1459 | 0.1161 | 0.1502 | 0.1020 |
| 0.2 | 0.0421 | 0.0626 | 0.0463 | 0.1628 | 0.1341 | 0.1660 | 0.1144 |
| 0.1 | 0.0470 | 0.0798 | 0.0601 | 0.1890 | 0.1594 | 0.1895 | 0.1356 |

**Ablation 3: Teacher update frequency.** The ablation on teacher update frequency (Table 10) investigates the trade-off between stability and plasticity in the dynamic teacher. We vary the frequency from 1 to 2500 steps ($\approx$ 1 epoch). The results show that updating every 50–100 steps yields the highest OOD accuracy, whereas slower update schedules lead to lower OOD ECE. These findings align with the dynamic-teacher analysis: slower updates preserve early robustness and calibration, while faster updates allow the teacher to better track the evolving task solution and enhance accuracy.

*Table 10.* Ablation of teacher update frequency in TRACER distillation across ImageNet (IN) and distribution shifts. For each setting (row), accuracy (Acc.↑) is reported in %, and expected calibration error (ECE↓) in [0, 1]. OOD Avg. is the mean over {IN-V2, IN-R, IN-A, IN-S, ObjectNet}. The update frequency denotes the number of training steps between each teacher model update from the student; lower frequencies (e.g., 2-10 steps) result in a teacher that closely follows the student's trajectory providing fine-grained regularization, while higher frequencies (e.g., 500-2500 steps) maintain a more stable teacher that changes less frequently, providing stronger regularization from earlier checkpoints and the initial pretrained model.

| Update Freq. | Acc.↑ (%) | | | | | | |
| --- | --- | --- | --- | --- | --- | --- | --- |
| | IN | IN-V2 | IN-R | IN-A | IN-S | ObjectNet | OOD Avg. |
| 2500 | 81.38 | 72.97 | **79.83** | 55.05 | 53.88 | 58.17 | 63.98 |
| 1000 | 81.90 | 73.27 | 79.71 | 54.93 | 54.21 | 58.26 | 64.08 |
| 500 | 82.13 | 73.53 | 79.80 | 54.72 | **54.23** | 58.30 | 64.12 |
| 100 | 82.54 | 73.92 | 79.76 | **55.09** | 54.00 | **58.31** | **64.22** |
| 50 | 82.62 | 73.84 | 79.69 | 54.96 | 53.96 | 58.12 | 64.11 |
| 10 | 82.57 | 74.01 | 79.58 | 54.96 | 53.88 | 58.00 | 64.09 |
| 5 | 82.70 | 73.98 | 79.57 | 54.81 | 53.93 | 58.07 | 64.07 |
| 2 | 82.73 | **74.12** | 79.57 | 54.51 | 53.99 | 58.09 | 64.06 |
| 1 | **82.76** | 74.14 | 79.33 | 54.92 | 53.69 | 58.26 | 64.07 |

| Update Freq. | ECE↓ | | | | | | |
| --- | --- | --- | --- | --- | --- | --- | --- |
| | IN | IN-V2 | IN-R | IN-A | IN-S | ObjectNet | OOD Avg. |
| 2500 | 0.0614 | 0.0426 | 0.0632 | **0.0839** | 0.0722 | **0.0764** | **0.0677** |
| 1000 | 0.0562 | 0.0434 | 0.0581 | 0.0908 | 0.0698 | 0.0825 | 0.0689 |
| 500 | 0.0524 | 0.0419 | 0.0548 | 0.0935 | **0.0677** | 0.0868 | 0.0689 |
| 100 | 0.0475 | 0.0410 | 0.0487 | 0.0985 | 0.0694 | 0.0920 | 0.0699 |
| 50 | 0.0481 | 0.0413 | 0.0471 | 0.0992 | 0.0713 | 0.0949 | 0.0708 |
| 10 | 0.0473 | 0.0408 | 0.0468 | 0.0983 | 0.0714 | 0.0978 | 0.0710 |
| 5 | 0.0480 | **0.0400** | 0.0451 | 0.1001 | 0.0738 | 0.0975 | 0.0713 |
| 2 | 0.0471 | 0.0409 | 0.0440 | 0.1034 | 0.0733 | 0.0990 | 0.0721 |
| 1 | **0.0446** | 0.0394 | **0.0412** | 0.1041 | 0.0784 | 0.1030 | 0.0732 |

**Ablation 4: Beta kernel shape.** The ablation on the Beta kernel shape (Table 11) evaluates the role of endpoint-aware curricula. We vary $\beta \in \{0.2, 0.5, 0.7, 0.9, 1.0, 1.5\}$. Results show that smaller $\beta$ values (0.2–0.5), which emphasize endpoints, enhance both OOD accuracy and ECE by reinforcing strong early anchoring and late solution emphasis. In contrast, larger $\beta$ values favor mid-trajectory weighting, yielding marginal ID improvements at the cost of reduced OOD gains. These findings suggest that arcsine-like weighting is particularly effective for robust finetuning. Additional kernel families are discussed in §C.5.1.

*Table 11*. Ablation of $\beta$ value in Beta($\beta$, $\beta$) distribution for teacher weighting in TRACER distillation across ImageNet (IN) and distribution shifts. For each setting (row), accuracy (Acc.↑) is reported in %, and expected calibration error (ECE↓) in [0, 1]. OOD Avg. is the mean over {IN-V2, IN-R, IN-A, IN-S, ObjectNet}. The $\beta$ value controls the shape of the distribution used for sampling teacher ensemble weights. Lower $\beta$ values ($< 1$) assign higher weights to the pretrained model and early training steps, $\beta = 1$ corresponds to uniform weighting, while higher $\beta$ values ($> 1$) emphasize intermediate training steps and down-weight both the initial pretrained model and final training steps.

| $\beta$ | IN | IN-V2 | IN-R | IN-A | IN-S | ObjectNet | OOD Avg. |
|---|---|---|---|---|---|---|---|
| | | | Acc.↑ (%) | | | | |
| 1.5 | **83.19** | 74.38 | 77.15 | 52.43 | 52.95 | 57.06 | 62.79 |
| 1.0 | 83.09 | **74.39** | 78.10 | 52.93 | 53.25 | 57.41 | 63.22 |
| 0.9 | 83.14 | 74.26 | 78.28 | 53.59 | 53.35 | 57.41 | 63.38 |
| 0.7 | 82.97 | 74.32 | 78.89 | 54.28 | 53.58 | 57.80 | 63.77 |
| 0.5 | 82.76 | 74.14 | 79.33 | 54.92 | 53.69 | 58.26 | 64.07 |
| 0.2 | 81.91 | 73.46 | **79.96** | **55.27** | **54.08** | **58.34** | **64.22** |

| $\beta$ | IN | IN-V2 | IN-R | IN-A | IN-S | ObjectNet | OOD Avg. |
|---|---|---|---|---|---|---|---|
| | | | ECE↓ | | | | |
| 1.5 | 0.0403 | 0.0485 | 0.0418 | 0.1433 | 0.1069 | 0.1418 | 0.0965 |
| 1.0 | 0.0407 | 0.0478 | 0.0393 | 0.1317 | 0.0957 | 0.1286 | 0.0886 |
| 0.9 | **0.0401** | 0.0477 | 0.0402 | 0.1280 | 0.0926 | 0.1262 | 0.0869 |
| 0.7 | 0.0416 | 0.0456 | **0.0388** | 0.1148 | 0.0856 | 0.1161 | 0.0802 |
| 0.5 | 0.0446 | 0.0394 | 0.0412 | 0.1041 | 0.0784 | 0.1030 | 0.0732 |
| 0.2 | 0.0548 | **0.0424** | 0.0542 | **0.0917** | **0.0670** | **0.0843** | **0.0679** |

# C. Additional Theoretical Details

This section provides the full derivations, proofs, and detailed geometric interpretations for the theoretical analysis presented in the main paper.

## C.1. Derivation of $\mathcal{L}_{\text{CL}}$

We start with the linearized multimodal contrastive learning (MMCL) loss function, which balances positive and negative pairs across a batch, as commonly used in theoretical analyses (Ji et al., 2023; Tian, 2022; Nakada et al., 2023; Xue et al., 2024). The original formulation is given by:

$$\mathcal{L}_{\text{MMCL}}(\mathbf{W}_I, \mathbf{W}_T) = \frac{1}{2n(n-1)} \sum_i \sum_{j \neq i} (s_{ij} - s_{ii}) + \frac{1}{2n(n-1)} \sum_i \sum_{j \neq i} (s_{ji} - s_{ii}) + \frac{\rho}{2} \|\mathbf{W}_I^\top \mathbf{W}_T\|_F^2$$

where $s_{ij} = (\mathbf{W}_I \mathbf{x}_I^i)^\top (\mathbf{W}_T \mathbf{x}_T^j)$ represents the similarity score between image $i$ and text $j$.

Expanding the first term:

$$\frac{1}{2n(n-1)} \sum_i \sum_{j \neq i} (s_{ij} - s_{ii}) = \frac{1}{2n(n-1)} \left[ \sum_i \sum_{j \neq i} s_{ij} - \sum_i \sum_{j \neq i} s_{ii} \right]$$

Since for each $i$, there are $(n-1)$ values of $j \neq i$, the second sub-sum simplifies:

$$= \frac{1}{2n(n-1)} \left[ \sum_i \sum_{j \neq i} s_{ij} - (n-1) \sum_i s_{ii} \right]$$

Expanding the second term:

$$\frac{1}{2n(n-1)} \sum_i \sum_{j \neq i} (s_{ji} - s_{ii}) = \frac{1}{2n(n-1)} \left[ \sum_i \sum_{j \neq i} s_{ji} - \sum_i \sum_{j \neq i} s_{ii} \right]$$

Similarly, this becomes:

$$= \frac{1}{2n(n-1)} \left[ \sum_i \sum_{j \neq i} s_{ji} - (n-1) \sum_i s_{ii} \right]$$

Combining both terms: Adding the first and second terms yields:

$$\frac{1}{2n(n-1)} \left[ \sum_i \sum_{j \neq i} s_{ij} + \sum_i \sum_{j \neq i} s_{ji} - 2(n-1) \sum_i s_{ii} \right]$$

Note that $\sum_i \sum_{j \neq i} s_{ji}$ is simply a re-indexing of $\sum_j \sum_{i \neq j} s_{ij}$, which is equivalent to $\sum_i \sum_{j \neq i} s_{ij}$. Therefore:

$$\sum_i \sum_{j \neq i} s_{ij} + \sum_i \sum_{j \neq i} s_{ji} = 2 \sum_i \sum_{j \neq i} s_{ij}$$

Substituting back into $\mathcal{L}_{\text{MMCL}}$:

$$\mathcal{L}_{\text{MMCL}} = \frac{1}{2n(n-1)} \left[ 2 \sum_i \sum_{j \neq i} s_{ij} - 2(n-1) \sum_i s_{ii} \right] + \frac{\rho}{2} \|\mathbf{W}_I^\top \mathbf{W}_T\|_F^2$$

$$= \frac{1}{n(n-1)} \left[ \sum_i \sum_{j \neq i} s_{ij} - (n-1) \sum_i s_{ii} \right] + \frac{\rho}{2} \|\mathbf{W}_I^\top \mathbf{W}_T\|_F^2$$

We define the core contrastive alignment term as:

$$\mathcal{L}_{\text{CL}} = \sum_{i=1}^n \sum_{j \neq i} s_{ij} - (n-1) \sum_{i=1}^n s_{ii}. \tag{4}$$

And the regularization term as $R(\mathbf{W}_I, \mathbf{W}_T) = \frac{\rho}{2} \|\mathbf{W}_I^\top \mathbf{W}_T\|_F^2$. Thus, the total MMCL loss can be written as:

$$\mathcal{L}_{\text{MMCL}} = \frac{1}{n(n-1)} \mathcal{L}_{\text{CL}} + R(\mathbf{W}_I, \mathbf{W}_T)$$

## C.2. Reformulation of the Least-Squares Objective

We demonstrate how the contrastive alignment term $\mathcal{L}_{\mathrm{CL}}$ can be re-expressed as a matrix least-squares problem, which is the foundation of our theoretical analysis. Recall $\mathcal{L}_{\mathrm{CL}} = \sum_{i=1}^{n}\sum_{j\neq i} s_{ij} - (n-1)\sum_{i=1}^{n} s_{ii}$. Let $\mathbf{H}_I = \mathbf{W}_I \mathbf{X}_I$ and $\mathbf{H}_T = \mathbf{W}_T^0 \mathbf{X}_T$. Then $s_{ij} = (\mathbf{H}_I)_i^\top (\mathbf{H}_T)_j$. Let $\mathbf{S} = \mathbf{H}_I^\top \mathbf{H}_T$. The sum of all similarities is $\mathbf{1}^\top \mathbf{S} \mathbf{1} = \sum_{i,j} s_{ij}$. The sum of diagonal similarities is $\mathrm{Tr}(\mathbf{S}) = \sum_i s_{ii}$. Then, $\sum_{i=1}^{n}\sum_{j\neq i} s_{ij} = \sum_{i,j} s_{ij} - \sum_i s_{ii} = \mathbf{1}^\top \mathbf{S} \mathbf{1} - \mathrm{Tr}(\mathbf{S})$. Substituting this into $\mathcal{L}_{\mathrm{CL}}$:

$$
\begin{aligned}
\mathcal{L}_{\mathrm{CL}} &= (\mathbf{1}^\top \mathbf{S} \mathbf{1} - \mathrm{Tr}(\mathbf{S})) - (n-1)\,\mathrm{Tr}(\mathbf{S}) \\
&= \mathbf{1}^\top \mathbf{S} \mathbf{1} - n\,\mathrm{Tr}(\mathbf{S}) \\
&= \mathrm{Tr}(\mathbf{1}\mathbf{1}^\top \mathbf{S}) - n\,\mathrm{Tr}(\mathbf{S}) \\
&= \mathrm{Tr}((\mathbf{J}_n - n\mathbf{I}_n)^\top \mathbf{S}) \\
&= \mathrm{Tr}((\mathbf{J}_n - n\mathbf{I}_n)^\top \mathbf{H}_I^\top \mathbf{H}_T) \\
&= \mathrm{Tr}(\mathbf{H}_T(\mathbf{J}_n - n\mathbf{I}_n)\mathbf{H}_I^\top) \\
&= \mathrm{Tr}(\mathbf{W}_T^0 \mathbf{X}_T (\mathbf{J}_n - n\mathbf{I}_n)(\mathbf{W}_I \mathbf{X}_I)^\top) \\
&= -\mathrm{Tr}\!\left((\mathbf{W}_T^0 \mathbf{X}_T (n\mathbf{I}_n - \mathbf{J}_n))^\top \mathbf{W}_I \mathbf{X}_I\right).
\end{aligned}
$$

Let $\mathbf{Y}_{\mathrm{FT}} = \mathbf{W}_T^0 \mathbf{X}_T (n\mathbf{I}_n - \mathbf{J}_n)$, as defined in Definition 3.1. Then

$$
\mathcal{L}_{\mathrm{CL}} = -\mathrm{Tr}(\mathbf{Y}_{\mathrm{FT}}^\top \mathbf{W}_I \mathbf{X}_I).
$$

Expanding the Frobenius norm of the least-squares residual gives:

$$
\begin{aligned}
\frac{1}{2}\|\mathbf{W}_I \mathbf{X}_I - \mathbf{Y}_{\mathrm{FT}}\|_{\mathrm{F}}^2 &= \frac{1}{2}\mathrm{Tr}\!\left((\mathbf{W}_I \mathbf{X}_I - \mathbf{Y}_{\mathrm{FT}})^\top (\mathbf{W}_I \mathbf{X}_I - \mathbf{Y}_{\mathrm{FT}})\right) \\
&= \frac{1}{2}\mathrm{Tr}\big(\mathbf{X}_I^\top \mathbf{W}_I^\top \mathbf{W}_I \mathbf{X}_I - \mathbf{X}_I^\top \mathbf{W}_I^\top \mathbf{Y}_{\mathrm{FT}} \\
&\quad - \mathbf{Y}_{\mathrm{FT}}^\top \mathbf{W}_I \mathbf{X}_I + \mathbf{Y}_{\mathrm{FT}}^\top \mathbf{Y}_{\mathrm{FT}}\big) \\
&= \underbrace{\frac{1}{2}\|\mathbf{W}_I \mathbf{X}_I\|_{\mathrm{F}}^2}_{(\star)} \underbrace{-\,\mathrm{Tr}(\mathbf{Y}_{\mathrm{FT}}^\top \mathbf{W}_I \mathbf{X}_I)}_{=\,\mathcal{L}_{\mathrm{CL}}} + \underbrace{\frac{1}{2}\|\mathbf{Y}_{\mathrm{FT}}\|_{\mathrm{F}}^2}_{\text{constant in } \mathbf{W}_I}.
\end{aligned}
$$

**A clarification on the trace–least-squares relationship.** As the expansion above shows, $\frac{1}{2}\|\mathbf{W}_I \mathbf{X}_I - \mathbf{Y}_{\mathrm{FT}}\|_{\mathrm{F}}^2$ is *not* equal to $-\mathrm{Tr}(\mathbf{Y}_{\mathrm{FT}}^\top \mathbf{W}_I \mathbf{X}_I)$ up to a constant alone; the two differ by the data-dependent quadratic term $(\star) = \frac{1}{2}\|\mathbf{W}_I \mathbf{X}_I\|_{\mathrm{F}}^2$. Hence minimizing the pure trace functional $-\mathrm{Tr}(\mathbf{Y}_{\mathrm{FT}}^\top \mathbf{W}_I \mathbf{X}_I)$ in isolation is unbounded below and is *not* equivalent to minimizing the least-squares objective.

What we use throughout the rest of the analysis is the *least-squares* reformulation

$$
\min_{\mathbf{W}_I} \tfrac{1}{2}\|\mathbf{W}_I \mathbf{X}_I - \mathbf{Y}_{\mathrm{FT}}\|_{\mathrm{F}}^2 = \min_{\mathbf{W}_I} \tfrac{1}{2}\|\mathbf{W}_I \mathbf{X}_I\|_{\mathrm{F}}^2 - \mathrm{Tr}(\mathbf{Y}_{\mathrm{FT}}^\top \mathbf{W}_I \mathbf{X}_I) + \mathrm{const}, \tag{5}
$$

which differs from the original linearized MMCL term by the additive data-dependent quadratic $(\star)$. This extra term can be viewed as the natural *data-dependent quadratic regularization* that arises whenever a bilinear similarity is matched against a fixed target $\mathbf{Y}_{\mathrm{FT}}$ under a Frobenius surrogate, and it is precisely what makes the resulting problem a well-posed matrix least-squares program with closed-form solutions. All subsequent closed-form solutions (Theorem C.2) and dynamic-teacher analyses (§C.5) are derived from this least-squares objective; the original $-\mathrm{Tr}(\cdot)$ form is recovered by dropping $(\star)$, but our results are unaffected because they follow from the least-squares program.

## C.3. Unified Framework for Contrastive Finetuning: Proofs and Details

Our proofs rely on the following lemma for gradient descent on a matrix quadratic program.

*Lemma* C.1 (Gradient Descent for Matrix Quadratic Programs). Let $\mathcal{Q} : \mathbb{R}^{p\times d} \to \mathbb{R}^{p\times d}$ be a positive semi-definite (PSD) linear operator and $\mathbf{P} \in \mathbb{R}^{p\times d}$. Consider the quadratic objective

$$
f(\mathbf{W}) = \frac{1}{2}\langle \mathbf{W}, \mathcal{Q}(\mathbf{W})\rangle_F - \langle \mathbf{P}, \mathbf{W}\rangle_F, \tag{6}
$$

where $\langle \cdot, \cdot \rangle_F$ denotes the Frobenius inner product. Let $\|\mathcal{Q}\|_{\mathrm{op}}$ denote the operator norm of $\mathcal{Q}$ induced by the Frobenius norm. If $\mathbf{P} \in \mathrm{Range}(\mathcal{Q})$, then gradient descent initialized at $\mathbf{W}_0$ with step size $\gamma \in (0, 2/\|\mathcal{Q}\|_{\mathrm{op}})$ converges to

$$
\mathbf{W}_\infty = (\mathbf{I} - \Pi_{\mathcal{Q}})(\mathbf{W}_0) + \mathcal{Q}^+(\mathbf{P}), \tag{7}
$$

where $\Pi_{\mathcal{Q}}$ is the orthogonal projector onto $\mathrm{Range}(\mathcal{Q})$ and $\mathcal{Q}^+$ is the Moore-Penrose pseudoinverse of $\mathcal{Q}$.

*Proof.* The gradient of $f$ is given by $\nabla f(\mathbf{W}) = \mathcal{Q}(\mathbf{W}) - \mathbf{P}$, yielding the gradient descent update

$$
\mathbf{W}_{t+1} = \mathbf{W}_t - \gamma(\mathcal{Q}(\mathbf{W}_t) - \mathbf{P}). \tag{8}
$$

Since $\mathcal{Q}$ is PSD, we have the orthogonal decomposition

$$\mathbb{R}^{p \times d} = \text{Range}(\mathcal{Q}) \oplus \text{Null}(\mathcal{Q}). \tag{9}$$

Let $\Pi_{\mathcal{Q}}$ and $\Pi_{\mathcal{Q}^\perp} = \mathbf{I} - \Pi_{\mathcal{Q}}$ denote the orthogonal projectors onto the range and null space of $\mathcal{Q}$, respectively.

**Analysis of the null space component.** Projecting the gradient descent update onto $\text{Null}(\mathcal{Q})$ yields

$$\Pi_{\mathcal{Q}^\perp}(\mathbf{W}_{t+1}) = \Pi_{\mathcal{Q}^\perp}(\mathbf{W}_t) - \gamma \Pi_{\mathcal{Q}^\perp}(\mathcal{Q}(\mathbf{W}_t)) + \gamma \Pi_{\mathcal{Q}^\perp}(\mathbf{P})$$
$$= \Pi_{\mathcal{Q}^\perp}(\mathbf{W}_t),$$

where we used that $\mathcal{Q}(\mathbf{W}_t) \in \text{Range}(\mathcal{Q})$ implies $\Pi_{\mathcal{Q}^\perp}(\mathcal{Q}(\mathbf{W}_t)) = \mathbf{0}$, and our assumption $\mathbf{P} \in \text{Range}(\mathcal{Q})$ implies $\Pi_{\mathcal{Q}^\perp}(\mathbf{P}) = \mathbf{0}$. Thus, the null space component remains invariant throughout the optimization:

$$\Pi_{\mathcal{Q}^\perp}(\mathbf{W}_t) = \Pi_{\mathcal{Q}^\perp}(\mathbf{W}_0) \quad \forall t \geq 0. \tag{10}$$

**Analysis of the range component.** Let $\mathbf{W}'_t = \Pi_{\mathcal{Q}}(\mathbf{W}_t)$ denote the projection onto $\text{Range}(\mathcal{Q})$. The dynamics of this component follow

$$\mathbf{W}'_{t+1} = (\mathbf{I} - \gamma \mathcal{Q})\mathbf{W}'_t + \gamma \mathbf{P}. \tag{11}$$

The restriction of $\mathcal{Q}$ to its range, denoted $\mathcal{Q}_R : \text{Range}(\mathcal{Q}) \to \text{Range}(\mathcal{Q})$, is positive definite (since for any non-zero $x \in \text{Range}(\mathcal{Q})$, we must have $\mathcal{Q}(x) \neq 0$, otherwise $x$ would be in $\text{Null}(\mathcal{Q})$). For $\gamma \in (0, 2/\|\mathcal{Q}\|_{\text{op}})$, the operator $\mathbf{I} - \gamma \mathcal{Q}_R$ has spectral radius less than 1, making it a contraction mapping. By the Banach fixed-point theorem, the sequence $\{\mathbf{W}'_t\}$ converges to the unique fixed point $\mathbf{W}'_\infty \in \text{Range}(\mathcal{Q})$ satisfying

$$\mathbf{W}'_\infty = (\mathbf{I} - \gamma \mathcal{Q})\mathbf{W}'_\infty + \gamma \mathbf{P}. \tag{12}$$

Rearranging gives $\mathcal{Q}(\mathbf{W}'_\infty) = \mathbf{P}$, which has the unique solution $\mathbf{W}'_\infty = \mathcal{Q}^+(\mathbf{P})$ in $\text{Range}(\mathcal{Q})$.

**Synthesis.** Combining the analyses of both components, we obtain

$$\mathbf{W}_\infty = \lim_{t \to \infty} \left( \Pi_{\mathcal{Q}^\perp}(\mathbf{W}_t) + \mathbf{W}'_t \right)$$
$$= \Pi_{\mathcal{Q}^\perp}(\mathbf{W}_0) + \mathcal{Q}^+(\mathbf{P})$$
$$= (\mathbf{I} - \Pi_{\mathcal{Q}})(\mathbf{W}_0) + \mathcal{Q}^+(\mathbf{P}),$$

completing the proof. $\square$

*Theorem* C.2 (Unified Framework for Contrastive Finetuning Solutions (Full Proof)). Let $\mathcal{P}_I := \mathbf{X}_I(\mathbf{X}_I^\top \mathbf{X}_I)^+ \mathbf{X}_I^\top$ denote the orthogonal projection onto the subspace spanned by the finetuning data $\mathbf{X}_I$. Consider the general finetuning objective:

$$\mathcal{L}(\mathbf{W}_I) = \frac{1}{2} \|\mathbf{W}_I \mathbf{X}_I - \mathbf{Y}_{\text{FT}}\|_F^2 + \mathcal{R}(\mathbf{W}_I) \tag{13}$$

where $\mathcal{R}(\mathbf{W}_I)$ represents different regularization strategies. Gradient descent initialized at $\mathbf{W}_I^0$ with sufficiently small learning rate converges to the following solutions:

| Strategy | $\mathcal{R}(\mathbf{W}_I)$ | Solution |
|---|---|---|
| Direct Finetuning | $0$ | $\mathbf{W}_{\text{FT}} = \mathbf{W}_I^0(\mathbf{I} - \mathcal{P}_I) + \mathbf{Y}_{\text{FT}}\mathbf{X}_I^\top(\mathbf{X}_I\mathbf{X}_I^\top)^+$ |
| $L_2$ Regularization (L2-SP (Li et al., 2018)) | $\frac{\lambda}{2}\left\|\mathbf{W}_I - \mathbf{W}_I^0\right\|_F^2$ | $\mathbf{W}_{L_2} = (\mathbf{Y}_{\text{FT}}\mathbf{X}_I^\top + \lambda\mathbf{W}_I^0)(\mathbf{X}_I\mathbf{X}_I^\top + \lambda\mathbf{I})^{-1}$ |
| Self-Distillation (SD (Furlanello et al., 2018)) | $\frac{\lambda}{2}\left\|\mathbf{W}_I\mathbf{X}_I - \mathbf{W}_I^0\mathbf{X}_I\right\|_F^2$ | $\mathbf{W}_{SD} = \mathbf{W}_I^0(\mathbf{I} - \frac{1}{1+\lambda}\mathcal{P}_I) + \frac{1}{1+\lambda}\mathbf{Y}_{\text{FT}}\mathbf{X}_I^\top(\mathbf{X}_I\mathbf{X}_I^\top)^+$ |

Here, $^+$ denotes the Moore-Penrose pseudoinverse and $\lambda > 0$ is the regularization parameter.

*Proof.* Let $\mathbf{C}_I = \mathbf{X}_I \mathbf{X}_I^\top$.

**Direct Finetuning.** The objective is $\mathcal{L}(\mathbf{W}_I) = \frac{1}{2}\|\mathbf{W}_I\mathbf{X}_I - \mathbf{Y}_{\text{FT}}\|_F^2$. We rewrite this in the quadratic form of Lemma C.1:

$$\mathcal{L}(\mathbf{W}_I) = \frac{1}{2}\langle \mathbf{W}_I\mathbf{X}_I - \mathbf{Y}_{\text{FT}}, \mathbf{W}_I\mathbf{X}_I - \mathbf{Y}_{\text{FT}}\rangle_F$$
$$= \frac{1}{2}\langle \mathbf{W}_I, \mathbf{W}_I(\mathbf{X}_I\mathbf{X}_I^\top)\rangle_F - \langle \mathbf{W}_I, \mathbf{Y}_{\text{FT}}\mathbf{X}_I^\top\rangle_F + \frac{1}{2}\|\mathbf{Y}_{\text{FT}}\|_F^2$$
$$= \frac{1}{2}\langle \mathbf{W}_I, \mathbf{W}_I\mathbf{C}_I\rangle_F - \langle \mathbf{W}_I, \mathbf{Y}_{\text{FT}}\mathbf{X}_I^\top\rangle_F + \text{const.}$$

This matches the form $f(\mathbf{W}) = \frac{1}{2}\langle \mathbf{W}, \mathcal{Q}(\mathbf{W})\rangle_F - \langle \mathbf{P}, \mathbf{W}\rangle_F$ with $\mathcal{Q}(\mathbf{W}) = \mathbf{W}\mathbf{C}_I$ and $\mathbf{P} = \mathbf{Y}_{\text{FT}}\mathbf{X}_I^\top$.

The operator $\mathcal{Q}$ is linear and positive semi-definite, as $\langle \mathbf{W}_I, \mathcal{Q}(\mathbf{W}_I)\rangle_F = \|\mathbf{W}_I\mathbf{X}_I\|_F^2 \geq 0$. The condition $\mathbf{P} \in \text{Range}(\mathcal{Q})$ holds because the rows of $\mathbf{P} = \mathbf{Y}_{\text{FT}}\mathbf{X}_I^\top$ are linear combinations of the rows of $\mathbf{X}_I^\top$, which form the row space of $\mathbf{C}_I$.

By Lemma C.1, gradient descent converges to $\mathbf{W}_\infty = \Pi_{\mathcal{Q}^\perp}(\mathbf{W}_I^0) + \mathcal{Q}^+(\mathbf{P})$.

1. **Null Space Component:** The null space of $\mathcal{Q}$ consists of matrices $\mathbf{A}$ such that $\mathcal{Q}(\mathbf{A}) = \mathbf{A}\mathbf{C}_I = \mathbf{0}$. This holds if and only if the rows of $\mathbf{A}$ are in the null space of $\mathbf{C}_I$. The orthogonal projector onto this component of the initial matrix $\mathbf{W}_I^0$ is $\Pi_{\mathcal{Q}^\perp}(\mathbf{W}_I^0) = \mathbf{W}_I^0(\mathbf{I} - \mathcal{P}_I)$, where $\mathcal{P}_I = \mathbf{C}_I\mathbf{C}_I^+$ is the projector onto the row space of $\mathbf{X}_I$. This component is preserved.

2. **Range Component:** The pseudoinverse $\mathcal{Q}^+$ finds the minimum Frobenius norm solution to $\mathcal{Q}(\mathbf{W}) = \mathbf{P}$ that lies in $\text{Range}(\mathcal{Q})$. This is the solution to $\mathbf{W}\mathbf{C}_I = \mathbf{Y}_{\text{FT}}\mathbf{X}_I^\top$, which is $\mathcal{Q}^+(\mathbf{P}) = (\mathbf{Y}_{\text{FT}}\mathbf{X}_I^\top)\mathbf{C}_I^+$.

Combining the components gives the final solution:

$$\mathbf{W}_{\text{FT}} = \mathbf{W}_I^0(\mathbf{I} - \mathcal{P}_I) + \mathbf{Y}_{\text{FT}}\mathbf{X}_I^\top(\mathbf{X}_I\mathbf{X}_I^\top)^+.$$

$L_2$ **Regularization.** The objective $\mathcal{L}(\mathbf{W}_I) = \frac{1}{2}\|\mathbf{W}_I\mathbf{X}_I - \mathbf{Y}_{\text{FT}}\|_F^2 + \frac{\lambda}{2}\|\mathbf{W}_I - \mathbf{W}_I^0\|_F^2$. This objective is strongly convex for $\lambda > 0$. The unique minimizer is found by setting the gradient to zero:

$$\nabla_{\mathbf{W}_I}\mathcal{L} = (\mathbf{W}_I\mathbf{X}_I - \mathbf{Y}_{\text{FT}})\mathbf{X}_I^\top + \lambda(\mathbf{W}_I - \mathbf{W}_I^0) = 0$$
$$\rightarrow \mathbf{W}_I\mathbf{X}_I\mathbf{X}_I^\top + \lambda\mathbf{W}_I = \mathbf{Y}_{\text{FT}}\mathbf{X}_I^\top + \lambda\mathbf{W}_I^0$$
$$\rightarrow \mathbf{W}_I(\mathbf{X}_I\mathbf{X}_I^\top + \lambda\mathbf{I}) = \mathbf{Y}_{\text{FT}}\mathbf{X}_I^\top + \lambda\mathbf{W}_I^0$$

Since $\mathbf{X}_I\mathbf{X}_I^\top$ is PSD, the matrix $(\mathbf{X}_I\mathbf{X}_I^\top + \lambda\mathbf{I})$ is positive definite and thus invertible. The solution is:

$$\mathbf{W}_{L_2} = (\mathbf{Y}_{\text{FT}}\mathbf{X}_I^\top + \lambda\mathbf{W}_I^0)(\mathbf{X}_I\mathbf{X}_I^\top + \lambda\mathbf{I})^{-1}.$$

A more detailed analysis of the limit behavior of this solution as $\lambda \to 0$ and $\lambda \to \infty$ is provided in §C.4.

**Self-Distillation.** The objective is $\mathcal{L}(\mathbf{W}_I) = \frac{1}{2}\|\mathbf{W}_I\mathbf{X}_I - \mathbf{Y}_{\text{FT}}\|_F^2 + \frac{\lambda}{2}\|\mathbf{W}_I\mathbf{X}_I - \mathbf{W}_I^0\mathbf{X}_I\|_F^2$. Expanding and grouping terms reveals the quadratic structure:

$$\mathcal{L}(\mathbf{W}_I) = \frac{1}{2}\|\mathbf{W}_I\mathbf{X}_I\|_F^2 - \text{Tr}(\mathbf{Y}_{\text{FT}}^\top\mathbf{W}_I\mathbf{X}_I) + \frac{1}{2}\|\mathbf{Y}_{\text{FT}}\|_F^2$$
$$+ \frac{\lambda}{2}\|\mathbf{W}_I\mathbf{X}_I\|_F^2 - \lambda\,\text{Tr}((\mathbf{W}_I^0\mathbf{X}_I)^\top\mathbf{W}_I\mathbf{X}_I) + \frac{\lambda}{2}\|\mathbf{W}_I^0\mathbf{X}_I\|_F^2$$
$$= \frac{1+\lambda}{2}\|\mathbf{W}_I\mathbf{X}_I\|_F^2 - \text{Tr}((\mathbf{Y}_{\text{FT}}^\top + \lambda(\mathbf{W}_I^0\mathbf{X}_I)^\top)\mathbf{W}_I\mathbf{X}_I) + \text{const.}$$
$$= \frac{1+\lambda}{2}\langle\mathbf{W}_I, \mathbf{W}_I\mathbf{C}_I\rangle_F - \langle\mathbf{W}_I, \mathbf{Y}_{\text{FT}}\mathbf{X}_I^\top + \lambda\mathbf{W}_I^0\mathbf{C}_I\rangle_F + \text{const.}$$

This matches the form of Lemma C.1 with $\mathcal{Q}_{SD}(\mathbf{W}_I) = (1+\lambda)\mathbf{W}_I\mathbf{C}_I$ and $\mathbf{P}_{SD} = \mathbf{Y}_{\text{FT}}\mathbf{X}_I^\top + \lambda\mathbf{W}_I^0\mathbf{C}_I$.

The operator $\mathcal{Q}_{SD}$ is PSD. Its range and null space are identical to those of $\mathcal{Q}$ from the Direct Finetuning case. The terms $\mathbf{Y}_{\text{FT}}\mathbf{X}_I^\top$ and $\lambda\mathbf{W}_I^0\mathbf{C}_I$ are both in $\text{Range}(\mathcal{Q}_{SD})$ (as shown before for $\mathbf{Y}_{\text{FT}}\mathbf{X}_I^\top$, and $\mathbf{W}_I^0\mathbf{C}_I$ by definition). Thus, their sum $\mathbf{P}_{SD}$ is also in the range.

We apply Lemma C.1 to find the limit $\mathbf{W}_\infty = \Pi_{\mathcal{Q}_{SD}^\perp}(\mathbf{W}_I^0) + \mathcal{Q}_{SD}^+(\mathbf{P}_{SD})$.

1. **Null Space Component:** $\text{Null}(\mathcal{Q}_{SD}) = \text{Null}(\mathcal{Q})$, so the invariant component is again $\Pi_{\mathcal{Q}_{SD}^\perp}(\mathbf{W}_I^0) = \mathbf{W}_I^0(\mathbf{I} - \mathcal{P}_I)$.

2. **Range Component:** The pseudoinverse is $\mathcal{Q}_{SD}^+ = \frac{1}{1+\lambda}\mathcal{Q}^+$, where $\mathcal{Q}^+$ corresponds to the direct finetuning case. Applying it to $\mathbf{P}_{SD}$:

$$\mathcal{Q}_{SD}^+(\mathbf{P}_{SD}) = \frac{1}{1+\lambda}\mathcal{Q}^+\left(\mathbf{Y}_{\text{FT}}\mathbf{X}_I^\top + \lambda\mathbf{W}_I^0\mathbf{C}_I\right)$$
$$= \frac{1}{1+\lambda}\left((\mathbf{Y}_{\text{FT}}\mathbf{X}_I^\top)\mathbf{C}_I^+ + \lambda\mathcal{Q}^+(\mathcal{Q}(\mathbf{W}_I^0))\right)$$
$$= \frac{1}{1+\lambda}\left((\mathbf{Y}_{\text{FT}}\mathbf{X}_I^\top)\mathbf{C}_I^+ + \lambda\Pi_{\mathcal{Q}}(\mathbf{W}_I^0)\right)$$
$$= \frac{1}{1+\lambda}\left(\mathbf{Y}_{\text{FT}}\mathbf{X}_I^\top(\mathbf{X}_I\mathbf{X}_I^\top)^+ + \lambda\mathbf{W}_I^0\mathcal{P}_I\right).$$

Combining the components for the final solution $\mathbf{W}_{SD}$:

$$\mathbf{W}_{SD} = \mathbf{W}_I^0(\mathbf{I} - \mathcal{P}_I) + \frac{\lambda}{1+\lambda}\mathbf{W}_I^0\mathcal{P}_I + \frac{1}{1+\lambda}\mathbf{Y}_{\text{FT}}\mathbf{X}_I^\top(\mathbf{X}_I\mathbf{X}_I^\top)^+$$

$$= \mathbf{W}_I^0\left(\mathbf{I} - \mathcal{P}_I + \frac{\lambda}{1+\lambda}\mathcal{P}_I\right) + \frac{1}{1+\lambda}\mathbf{Y}_{\text{FT}}\mathbf{X}_I^\top(\mathbf{X}_I\mathbf{X}_I^\top)^+$$

$$= \mathbf{W}_I^0\left(\mathbf{I} - \frac{1}{1+\lambda}\mathcal{P}_I\right) + \frac{1}{1+\lambda}\mathbf{Y}_{\text{FT}}\mathbf{X}_I^\top(\mathbf{X}_I\mathbf{X}_I^\top)^+.$$

This completes the proof. $\qquad\square$

## C.4. Geometric Interpretation of Solutions

The closed-form solutions presented in Theorem C.2 provide a geometric understanding of how different finetuning strategies modify pretrained representations. We decompose the solution for $\mathbf{W}_I$ into components acting on the subspace spanned by the finetuning data $\mathbf{X}_I$ (parallel component) and its orthogonal complement (orthogonal component).

**Direct Finetuning.** The solution $\mathbf{W}_{\text{FT}}$ is a sum of two orthogonal parts: **(1)** $\mathbf{W}_I^0(\mathbf{I} - \mathcal{P}_I)$: This is the projection of the pretrained weights onto the orthogonal complement of the finetuning data subspace ($\text{Null}(\mathbf{X}_I^\top)$). This component preserves the action of $\mathbf{W}_I^0$ on data vectors orthogonal to the finetuning examples. **(2)** $\mathbf{Y}_{\text{FT}}\mathbf{X}_I^\top(\mathbf{X}_I\mathbf{X}_I^\top)^+$: This is the minimum-norm solution that fits the new contrastive task within the finetuning data subspace. This component lies entirely within the range of $\mathbf{X}_I^\top$.

*Interpretation:* Direct finetuning completely replaces (forgets) any pretrained knowledge related to features present in the finetuning data, substituting it with the new task-specific solution. It only preserves knowledge in directions entirely unrelated to the finetuning examples.

$L_2$ **Regularization.** The solution $\mathbf{W}_{L_2}$ is the standard matrix ridge regression solution. It creates a complex blend of the new task solution and the initial weights. There is no clean separation of orthogonal and parallel components as in direct finetuning or self-distillation. The key insight is that $L_2$ regularization modifies the data covariance matrix $\mathbf{X}_I\mathbf{X}_I^\top$ by adding $\lambda\mathbf{I}$, which acts as a *ridge* that prevents overfitting by shrinking the solution along all eigendirections of the data. Unlike direct finetuning and self-distillation, which primarily modify weights in the subspace spanned by $\mathbf{X}_I$, $L_2$ regularization affects all directions in the weight space, blending the old and new across the entire parameter space.

**Detailed Analysis of the $L_2$ Regularization Solution.** The solution for $L_2$ regularization is given by:

$$\mathbf{W}_{L_2} = \left(\mathbf{Y}_{\text{FT}}\mathbf{X}_I^\top + \lambda\mathbf{W}_I^0\right)\left(\mathbf{X}_I\mathbf{X}_I^\top + \lambda\mathbf{I}\right)^{-1}.$$

To analyze its behavior, we consider the eigendecomposition of the data covariance matrix $\mathbf{C}_I := \mathbf{X}_I\mathbf{X}_I^\top$. Since $\mathbf{C}_I$ is a real, symmetric, positive semi-definite (PSD) matrix, it has an eigendecomposition $\mathbf{C}_I = \mathbf{U}\boldsymbol{\Lambda}\mathbf{U}^\top$, where $\mathbf{U}$ is an orthogonal matrix of eigenvectors and $\boldsymbol{\Lambda}$ is a diagonal matrix of non-negative eigenvalues. Using this decomposition, the inverse term in the solution becomes:

$$(\mathbf{C}_I + \lambda\mathbf{I})^{-1} = (\mathbf{U}\boldsymbol{\Lambda}\mathbf{U}^\top + \lambda\mathbf{U}\mathbf{U}^\top)^{-1} = (\mathbf{U}(\boldsymbol{\Lambda} + \lambda\mathbf{I})\mathbf{U}^\top)^{-1} = \mathbf{U}(\boldsymbol{\Lambda} + \lambda\mathbf{I})^{-1}\mathbf{U}^\top.$$

The matrix $(\boldsymbol{\Lambda} + \lambda\mathbf{I})$ is diagonal with entries $\lambda_k + \lambda$, so its inverse has entries $1/(\lambda_k + \lambda)$.

**Analysis of the Limit as $\lambda \to 0$.** Let $r = \text{rank}(\mathbf{C}_I)$. We partition the eigenvectors $\mathbf{U}$ and eigenvalues $\boldsymbol{\Lambda}$ into components corresponding to non-zero and zero eigenvalues. Let $\mathbf{U}_r \in \mathbb{R}^{d_I \times r}$ contain eigenvectors for $r$ positive eigenvalues ($\boldsymbol{\Lambda}_r$), and $\mathbf{U}_0 \in \mathbb{R}^{d_I \times (d_I - r)}$ for zero eigenvalues. The projectors onto the range and null space of $\mathbf{C}_I$ are $\mathcal{P}_{\text{range}} = \mathbf{U}_r\mathbf{U}_r^\top$ and $\mathcal{P}_{\text{null}} = \mathbf{U}_0\mathbf{U}_0^\top$, respectively. Note that $\mathcal{P}_{\text{range}} = \mathcal{P}_I$ and $\mathcal{P}_{\text{null}} = \mathbf{I} - \mathcal{P}_I$.

The inverse term can be split:

$$(\mathbf{C}_I + \lambda\mathbf{I})^{-1} = \mathbf{U}_r(\boldsymbol{\Lambda}_r + \lambda\mathbf{I}_r)^{-1}\mathbf{U}_r^\top + \frac{1}{\lambda}\mathbf{U}_0\mathbf{U}_0^\top.$$

Substituting this back into $\mathbf{W}_{L_2}$:

$$\mathbf{W}_{L_2} = \left(\mathbf{Y}_{\text{FT}}\mathbf{X}_I^\top + \lambda\mathbf{W}_I^0\right)\left[\mathbf{U}_r(\boldsymbol{\Lambda}_r + \lambda\mathbf{I}_r)^{-1}\mathbf{U}_r^\top + \frac{1}{\lambda}\mathbf{U}_0\mathbf{U}_0^\top\right]$$

$$= \underbrace{\left(\mathbf{Y}_{\text{FT}}\mathbf{X}_I^\top + \lambda\mathbf{W}_I^0\right)\mathbf{U}_r(\boldsymbol{\Lambda}_r + \lambda\mathbf{I}_r)^{-1}\mathbf{U}_r^\top}_{\text{Term 1}}$$

$$+ \underbrace{\left(\mathbf{Y}_{\text{FT}}\mathbf{X}_I^\top + \lambda\mathbf{W}_I^0\right)\frac{1}{\lambda}\mathbf{U}_0\mathbf{U}_0^\top}_{\text{Term 2}}.$$

For Term 2, since $\mathbf{X}_I^\top \mathbf{U}_0 = \mathbf{0}$ (columns of $\mathbf{U}_0$ are in the null space of $\mathbf{C}_I$), it simplifies to:

$$\text{Term 2} = \frac{1}{\lambda} \mathbf{Y}_{\text{FT}} \underbrace{\mathbf{X}_I^\top \mathbf{U}_0}_{\mathbf{0}} \mathbf{U}_0^\top + \mathbf{W}_I^0 \mathbf{U}_0 \mathbf{U}_0^\top = \mathbf{W}_I^0 \mathcal{P}_{\text{null}} = \mathbf{W}_I^0 (\mathbf{I} - \mathcal{P}_I).$$

As $\lambda \to 0$, Term 1 converges to:

$$\lim_{\lambda \to 0} \text{Term 1} = \left( \mathbf{Y}_{\text{FT}} \mathbf{X}_I^\top \right) \mathbf{U}_r \mathbf{\Lambda}_r^{-1} \mathbf{U}_r^\top = \mathbf{Y}_{\text{FT}} \mathbf{X}_I^\top \mathbf{C}_I^+,$$

where $\mathbf{C}_I^+ = \mathbf{U}_r \mathbf{\Lambda}_r^{-1} \mathbf{U}_r^\top$ is the Moore-Penrose pseudoinverse of $\mathbf{C}_I$. Combining the limits of both terms, we get:

$$\lim_{\lambda \to 0} \mathbf{W}_{L_2} = \mathbf{Y}_{\text{FT}} \mathbf{X}_I^\top (\mathbf{X}_I \mathbf{X}_I^\top)^+ + \mathbf{W}_I^0 (\mathbf{I} - \mathcal{P}_I).$$

This is precisely the direct finetuning solution, $\mathbf{W}_{\text{FT}}$.

**Analysis of the Limit as $\lambda \to \infty$.** For the limit as $\lambda \to \infty$, we factor out $\lambda$:

$$\mathbf{W}_{L_2} = \left( \mathbf{Y}_{\text{FT}} \mathbf{X}_I^\top + \lambda \mathbf{W}_I^0 \right) \frac{1}{\lambda} \left( \frac{1}{\lambda} \mathbf{C}_I + \mathbf{I} \right)^{-1}$$

$$= \left( \frac{1}{\lambda} \mathbf{Y}_{\text{FT}} \mathbf{X}_I^\top + \mathbf{W}_I^0 \right) \left( \frac{1}{\lambda} \mathbf{C}_I + \mathbf{I} \right)^{-1}.$$

As $\lambda \to \infty$, the term $\frac{1}{\lambda} \to 0$. Therefore, the expression converges to:

$$\lim_{\lambda \to \infty} \mathbf{W}_{L_2} = \left( \mathbf{0} + \mathbf{W}_I^0 \right) \left( \mathbf{0} + \mathbf{I} \right)^{-1} = \mathbf{W}_I^0.$$

Thus, the regularization parameter $\lambda$ smoothly interpolates the solution between two meaningful extremes: pure task adaptation and pure preservation of pretrained weights.

**Self-Distillation.** The solution $\mathbf{W}_{SD}$ provides the most sophisticated and effective compromise. We can rewrite it to reveal its structure:

$$\mathbf{W}_{SD} = \mathbf{W}_I^0 - \frac{1}{1+\lambda} \mathbf{W}_I^0 \mathcal{P}_I + \frac{1}{1+\lambda} \mathbf{Y}_{\text{FT}} \mathbf{X}_I^\top (\mathbf{X}_I \mathbf{X}_I^\top)^+$$

$$= \mathbf{W}_I^0 (\mathbf{I} - \mathcal{P}_I) + \mathbf{W}_I^0 \mathcal{P}_I - \frac{1}{1+\lambda} \mathbf{W}_I^0 \mathcal{P}_I + \frac{1}{1+\lambda} \left( \mathbf{Y}_{\text{FT}} \mathbf{X}_I^\top (\mathbf{X}_I \mathbf{X}_I^\top)^+ \right)$$

$$= \underbrace{\mathbf{W}_I^0 (\mathbf{I} - \mathcal{P}_I)}_{\substack{\text{Component orthogonal to finetuning data} \\ \textbf{(Preserved)}}} + \underbrace{\frac{\lambda}{1+\lambda} \left( \mathbf{W}_I^0 \mathcal{P}_I \right) + \frac{1}{1+\lambda} \left( \mathbf{Y}_{\text{FT}} \mathbf{X}_I^\top (\mathbf{X}_I \mathbf{X}_I^\top)^+ \right)}_{\substack{\text{Component within finetuning data subspace} \\ \textbf{(Convex Combination)}}}$$

*Interpretation*: Self-Distillation operates with surgical precision: 1. **Outside the finetuning subspace**, it acts as an identity function, preserving the components of the pretrained model that are irrelevant to the new task. 2. **Inside the finetuning subspace**, it does not discard the pretrained knowledge. Instead, it computes a convex combination of the projected pretrained weights and the optimal solution for the new contrastive task. The hyperparameter $\lambda$ smoothly controls this trade-off. This demonstrates that Self-Distillation achieves a "best of both worlds" scenario: preserving general capabilities while adapting to new information where necessary.

## C.5. Dynamic Self-Distillation: WMA Details and Convergence

We extend the analysis of static self-distillation to a dynamic teacher, specifically a Weighted Moving Average (WMA) teacher, which adapts its regularization throughout training. This section provides the detailed definitions, dynamics, and convergence proofs.

**Definition C.3** (SD–WMA Objective (Repeated from Main Text)). At step $t$, the student weights $\mathbf{W}_I^t$ solve

$$\mathcal{L}_{\text{SD-WMA}}(\mathbf{W}_I) = \frac{1}{2} \left\| \mathbf{W}_I \mathbf{X}_I - \mathbf{Y}_{\text{FT}} \right\|_F^2 + \frac{\lambda}{2} \left\| \mathbf{W}_I \mathbf{X}_I - \mathbf{W}_{\text{Teacher}}^{t-1} \mathbf{X}_I \right\|_F^2, \qquad \text{initialized from } \mathbf{W}_I^{t-1}. \tag{14}$$

**Definition C.4** (Weighted Moving Average (WMA) Teacher (Repeated from Main Text)). Let the normalized time grid be

$$\tau_k = \frac{k + c_1}{T + c_2} \in (0, 1), \qquad c_1, c_2 > 0.$$

Choose any nonnegative *weighting kernel* $\kappa : [0,1] \to \mathbb{R}_{\geq 0}$ and define unnormalized weights $\alpha_k = \kappa(\tau_k)$ The *online* normalization and teacher recursion are

$$\omega_t = \frac{\alpha_t}{\sum_{j=0}^t \alpha_j}, \qquad \mathbf{W}_{\text{Teacher}}^t = (1 - \omega_t) \mathbf{W}_{\text{Teacher}}^{t-1} + \omega_t \mathbf{W}_I^t, \qquad \mathbf{W}_{\text{Teacher}}^0 = \mathbf{W}_I^0. \tag{15}$$

*Remark* C.5 (Teacher as a normalized history average). Unrolling equation 15 yields a normalized convex average of the student's history:

$$\mathbf{W}_{\text{Teacher}}^t = \sum_{k=0}^{t} \underbrace{\frac{\alpha_k}{\sum_{j=0}^{t} \alpha_j}}_{\omega_{k|t}} \mathbf{W}_I^k, \qquad \omega_{k|t} \geq 0, \quad \sum_{k=0}^{t} \omega_{k|t} = 1.$$

Thus the teacher is an *expectation* with respect to the discrete distribution $\text{Categorical}(\omega_{0|t}, \ldots, \omega_{t|t})$: $\mathbf{W}_{\text{Teacher}}^t = \mathbb{E}_{K \sim \omega_{\cdot|t}}[\mathbf{W}_I^K]$.

### C.5.1. WMA VS. EMA TEACHERS

This section contrasts the proposed *Weighted Moving Average* (WMA) teacher with the standard *Exponential Moving Average* (EMA), which underlies mean-teacher approaches.

**EMA (mean-teacher).** EMA maintains an exponentially decaying average:

$$\mathbf{W}_{\text{EMA}}^t = \rho \, \mathbf{W}_{\text{EMA}}^{t-1} + (1 - \rho) \, \mathbf{W}_I^t, \qquad \rho \in (0, 1), \;\; \mathbf{W}_{\text{EMA}}^0 = \mathbf{W}_I^0. \tag{16}$$

Unrolling this recursion gives a geometric kernel over *lag*:

$$\mathbf{W}_{\text{EMA}}^t = \rho^t \mathbf{W}_I^0 + (1 - \rho) \sum_{k=1}^{t} \rho^{t-k} \mathbf{W}_I^k = \sum_{k=0}^{t} \underbrace{\omega_{k|t}^{\text{EMA}}}_{\text{depends on } t-k} \mathbf{W}_I^k,$$

with $\omega_{0|t}^{\text{EMA}} = \rho^t$, $\omega_{k|t}^{\text{EMA}} = (1-\rho)\rho^{t-k}$ for $k \geq 1$, and $\sum_{k=0}^{t} \omega_{k|t}^{\text{EMA}} = 1$. The kernel is *stationary in lag*: weights depend only on recency $t - k$.

**WMA (normalized-time kernel).** In contrast, WMA assigns weights via a *kernel over normalized time* $\tau_k = (k + c_1)/(T + c_2)$:

$$\mathbf{W}_{\text{WMA}}^t = \sum_{k=0}^{t} \underbrace{\omega_{k|t}^{\text{WMA}}}_{\propto \, \kappa(\tau_k)} \mathbf{W}_I^k, \qquad \omega_{k|t}^{\text{WMA}} = \frac{\alpha_k}{\sum_{j=0}^{t} \alpha_j}, \quad \alpha_k = \kappa(\tau_k).$$

Here the kernel is *position-aware* in absolute (normalized) time, not just lag. The symmetric Beta kernel ($\beta_1 = \beta_2$) permits simultaneous emphasis of *both* endpoints (early stability and late convergence), a pattern that is *not* attainable with any single-parameter EMA.

**Key differences.**

- **Shape control.** EMA imposes a monotone geometric decay from the present; WMA can be early-peaked, late-peaked, flat (uniform), bimodal (e.g., arcsine), etc.

- **Invariance to schedule granularity.** WMA weights are defined on normalized time: if the training is retimed or step granularity changes while preserving the path over $[0, 1]$, the kernel $\kappa$ need not be retuned. EMA depends on the absolute decay $\rho$ and typically requires retuning when $T$ or logging cadence changes.

- **Endpoint behavior.** With $\beta_1 = \beta_2 = \frac{1}{2}$ (arcsine), WMA places substantial weight near $k \approx 0$ and $k \approx t$, preserving early information *and* emphasizing late iterates; EMA cannot simultaneously upweight both ends.

- **Recovering classical averages.** Choosing $\kappa$ uniform (Beta$(1, 1)$) yields the simple running average (Polyak/Ruppert; SWA (Izmailov et al., 2018)). EMA cannot realize an exactly uniform window without time-varying $\rho_t$.

- **Online normalization.** Both EMA and WMA are online and convex at each step; WMA's $\omega_t = \alpha_t / \sum_{j \leq t} \alpha_j$ admits arbitrary nonnegative $\alpha_t$ induced by $\kappa$.

**Mean-teacher within the WMA recursion.** In SD–WMA (Definition C.4), the step weight is $\omega_t = \alpha_t / \sum_{j=0}^{t} \alpha_j$, which is generally time-varying. To *recover EMA exactly* with constant $\omega \equiv 1 - \rho$, choose any $\alpha_0 > 0$ and set, for $t \geq 1$,

$$\alpha_t = \frac{\omega}{(1 - \omega)^t} \alpha_0 \qquad \Longleftrightarrow \qquad \alpha_t = \frac{1 - \rho}{\rho^t} \alpha_0, \tag{17}$$

which yields $\omega_t \equiv \omega$ and makes the WMA recursion identical to equation 16. If one insists on $\alpha_t = \kappa(\tau_t)$ with $\tau_t = (t + c_1)/(T + c_2)$, EMA corresponds to an exponential kernel over normalized time, $\kappa(\tau) = C (1 - \omega)^{-(T+c_2)\tau + c_1'}$, for suitable constants $C, c_1'$ (fixed per run), which reproduces $\omega_t \equiv \omega$ via equation 17.

C.5.2. THE PERSISTENT REGULARIZER OF THE WMA TEACHER

A key advantage of the WMA teacher over the more common EMA teacher lies in the dynamics of the regularization it provides. The self-distillation loss, $\mathcal{L}_{\text{SD}}$, induces a **regularizing gradient field**, $\mathbf{g}_R(\mathbf{W}_I^t) := \nabla_{\mathbf{W}_I} \mathcal{L}_{\text{SD}}(\mathbf{W}_T^t, \mathbf{W}_I^t)$, that pulls the student towards the teacher. The persistence of this field is critical for preventing the student from over-specializing on the finetuning task.

**The Vanishing Regularizer of EMA.** An EMA teacher is a low-pass filter of the student's trajectory: $\mathbf{W}_{\text{EMA}}^t = \rho \mathbf{W}_{\text{EMA}}^{t-1} + (1 - \rho) \mathbf{W}_I^t$. As the student's updates converge ($\|\mathbf{W}_I^{t+1} - \mathbf{W}_I^t\|_F \to 0$), the teacher necessarily converges to the student's final parameters ($\lim_{t \to \infty} \|\mathbf{W}_{\text{EMA}}^t - \mathbf{W}_I^t\|_F = 0$). Consequently, any regularizer based on the teacher–student gap vanishes:

$$\lim_{t \to \infty} \|\mathbf{g}_R(\mathbf{W}_I^t; \mathbf{W}_{\text{EMA}}^t)\|_F = 0,$$

allowing the optimization to be dominated entirely by the task loss near the end of training.

**The Finite-Horizon Persistence of WMA.** The WMA teacher is a weighted average of the *entire* student history: $\mathbf{W}_{\text{WMA}}^t = \sum_{k=0}^t \omega_{k|t} \mathbf{W}_I^k$. For any *fixed finite* run of length $T$, any kernel with $\alpha_0 > 0$ yields $\omega_{0|T} > 0$, so $\mathbf{W}_I^0$ contributes nontrivially to $\mathbf{W}_{\text{WMA}}^T$. Thus, when the student has moved away from initialization ($\mathbf{W}_I^T \neq \mathbf{W}_I^0$), the teacher can remain separated from the final iterate, yielding a nontrivial regularizing gradient at step $T$. Note that under online normalization, the relative weight on any fixed $k$ typically satisfies $\omega_{k|t} \to 0$ as $t \to \infty$; therefore any "non-vanishing" claim must be understood as a finite-horizon / late-training statement. In particular, under infinite-horizon training with online normalization and a convergent student ($\mathbf{W}_I^t \to \mathbf{W}_I^\infty$), one typically has $\mathbf{W}_{\text{WMA}}^t \to \mathbf{W}_I^\infty$, so the teacher–student gap can vanish asymptotically.

*Theorem* C.6 (Finite-Horizon Persistence of the WMA Regularizer). Fix a training horizon $T$ and let the WMA teacher at the end of training be

$$\mathbf{W}_{\text{WMA}}^T = \sum_{k=0}^T \omega_{k|T} \mathbf{W}_I^k, \qquad \omega_{k|T} \geq 0, \ \sum_{k=0}^T \omega_{k|T} = 1,$$

with $\omega_{0|T} > 0$ (true for any kernel with $\alpha_0 > 0$). Assume the student trajectory is *monotone along a single direction* in parameter space: there exist a unit matrix $\mathbf{U}$ with $\|\mathbf{U}\|_F = 1$ and scalars $0 = a_0 \leq a_1 \leq \cdots \leq a_T$ such that

$$\mathbf{W}_I^k = \mathbf{W}_I^0 + a_k \mathbf{U} \qquad \forall k \in \{0, \ldots, T\}.$$

Then, if $\mathbf{W}_I^T \neq \mathbf{W}_I^0$, the teacher remains *strictly behind* the final iterate and

$$\|\mathbf{W}_I^T - \mathbf{W}_{\text{WMA}}^T\|_F \geq \omega_{0|T} \|\mathbf{W}_I^T - \mathbf{W}_I^0\|_F > 0.$$

Moreover, suppose the self-distillation loss is *locally approximately quadratic* in the teacher–student parameter difference (e.g., second-order KL expansion) so that for $W$ near $\mathbf{W}_I^T$,

$$\mathbf{g}_R(W; \mathbf{W}_{\text{WMA}}^T) = \nabla_W \mathcal{L}_{\text{SD}}(\mathbf{W}_{\text{WMA}}^T, W) \approx \mathbf{F}_T (W - \mathbf{W}_{\text{WMA}}^T),$$

and define the terminal linearization residual

$$\mathbf{r}_T := \mathbf{g}_R(\mathbf{W}_I^T; \mathbf{W}_{\text{WMA}}^T) - \mathbf{F}_T(\mathbf{W}_I^T - \mathbf{W}_{\text{WMA}}^T).$$

If the curvature satisfies $\mathbf{F}_T \succeq \mu \mathbf{I}$ on span$\{\mathbf{U}\}$ for some $\mu > 0$, then the regularizing gradient at the end of training admits the *signal-minus-error* lower bound

$$\|\mathbf{g}_R(\mathbf{W}_I^T; \mathbf{W}_{\text{WMA}}^T)\|_F \geq \mu \omega_{0|T} \|\mathbf{W}_I^T - \mathbf{W}_I^0\|_F - \|\mathbf{r}_T\|_F.$$

In particular, if $\|\mathbf{r}_T\|_F < \mu \omega_{0|T} \|\mathbf{W}_I^T - \mathbf{W}_I^0\|_F$, then $\mathbf{g}_R(\mathbf{W}_I^T; \mathbf{W}_{\text{WMA}}^T) \neq \mathbf{0}$.

In contrast, for an EMA teacher with fixed $\rho \in (0, 1)$, if $\mathbf{W}_I^t \to \mathbf{W}_I^\infty$ then $\|\mathbf{W}_{\text{EMA}}^t - \mathbf{W}_I^t\|_F \to 0$, and thus the corresponding regularizing gradient vanishes asymptotically.

*Proof Sketch.* **Step 1: Teacher–student gap under monotone 1D motion.** Under the assumed form $\mathbf{W}_I^k = \mathbf{W}_I^0 + a_k \mathbf{U}$,

$$\mathbf{W}_{\text{WMA}}^T = \sum_{k=0}^T \omega_{k|T}(\mathbf{W}_I^0 + a_k \mathbf{U}) = \mathbf{W}_I^0 + \Big( \sum_{k=0}^T \omega_{k|T} a_k \Big) \mathbf{U}.$$

Hence

$$\mathbf{W}_I^T - \mathbf{W}_{\text{WMA}}^T = \Big( a_T - \sum_{k=0}^T \omega_{k|T} a_k \Big) \mathbf{U} = \sum_{k=0}^T \omega_{k|T}(a_T - a_k) \mathbf{U}.$$

Since $a_T \geq a_k$ and $\omega_{k|T} \geq 0$, all terms are nonnegative multiples of the same direction, so there is no cancellation and

$$\|\mathbf{W}_I^T - \mathbf{W}_{\text{WMA}}^T\|_F = \sum_{k=0}^T \omega_{k|T}(a_T - a_k) \geq \omega_{0|T}(a_T - a_0) = \omega_{0|T} \|\mathbf{W}_I^T - \mathbf{W}_I^0\|_F.$$

**Step 2: Gradient lower bound.** Let $\Delta_T = \mathbf{W}_I^T - \mathbf{W}_{\text{WMA}}^T \in \text{span}\{\mathbf{U}\}$. By definition, $\mathbf{g}_R(\mathbf{W}_I^T; \mathbf{W}_{\text{WMA}}^T) = \mathbf{F}_T \Delta_T + \mathbf{r}_T$, so by the triangle inequality,

$$\|\mathbf{g}_R(\mathbf{W}_I^T; \mathbf{W}_{\text{WMA}}^T)\|_F \geq \|\mathbf{F}_T \Delta_T\|_F - \|\mathbf{r}_T\|_F.$$

Using $\mathbf{F}_T \succeq \mu I$ on $\text{span}\{\mathbf{U}\}$ and Cauchy–Schwarz, $\|\mathbf{F}_T \Delta_T\|_F \geq \mu \|\Delta_T\|_F$, hence

$$\|\mathbf{g}_R(\mathbf{W}_I^T; \mathbf{W}_{\text{WMA}}^T)\|_F \geq \mu \|\Delta_T\|_F - \|\mathbf{r}_T\|_F \geq \mu\, \omega_{0|T} \|\mathbf{W}_I^T - \mathbf{W}_I^0\|_F - \|\mathbf{r}_T\|_F.$$

**Step 3: EMA vanishing.** If $\mathbf{W}_I^t \to \mathbf{W}_I^\infty$, then the EMA recursion is a stable linear filter of a convergent signal, implying $\mathbf{W}_{\text{EMA}}^t \to \mathbf{W}_I^\infty$ and thus $\|\mathbf{W}_{\text{EMA}}^t - \mathbf{W}_I^t\|_F \to 0$. $\qquad\square$

**Conclusion.** This theorem characterizes a *finite-horizon* effect: even when the student's updates become small near the end of training, a WMA teacher can remain separated from the terminal iterate at step $T$ because it retains positive mass on earlier states, yielding a regularizing gradient whose magnitude is lower bounded by a *signal minus approximation error* term. Over an *infinite* horizon with online normalization (where $\omega_{0|t} \to 0$) and a convergent student, the teacher–student gap can vanish, consistent with bias-free convergence results proved later for SD–WMA in the task subspace.

### C.5.3. CONVERGENCE ANALYSIS

We first state the single-step solution and then derive global convergence in the task subspace.

**From static to dynamic SD as a sequence of quadratic problems.** Before stating the single-step solution, it is useful to make explicit what the SD–WMA scheme is doing as an optimization process. At each step $t$, the SD–WMA objective

$$\mathcal{L}_{\text{SD-WMA}}(\mathbf{W}_I) = \frac{1}{2} \|\mathbf{W}_I \mathbf{X}_I - \mathbf{Y}_{\text{FT}}\|_F^2 + \frac{\lambda}{2} \|\mathbf{W}_I \mathbf{X}_I - \mathbf{W}_{\text{Teacher}}^{t-1} \mathbf{X}_I\|_F^2$$

is *still a static quadratic problem in* $\mathbf{W}_I$: it has the exact same algebraic form as the static self-distillation objective analyzed in Theorem C.2, with two simple substitutions relative to the static-SD case:

- the **anchor in the distillation term** is replaced from the fixed pretrained weights $\mathbf{W}_I^0$ to the current WMA teacher $\mathbf{W}_{\text{Teacher}}^{t-1}$;

- the **initialization of gradient descent** is replaced from $\mathbf{W}_I^0$ to the previous student iterate $\mathbf{W}_I^{t-1}$.

These two substitutions touch *different* parts of the closed-form solution given by Lemma C.1:

- changing the *anchor* $\mathbf{W}_I^0 \to \mathbf{W}_{\text{Teacher}}^{t-1}$ modifies the **range component** of the solution (the part lying in $\text{range}(\mathbf{X}_I)$, i.e., the task subspace), since this anchor enters the data term $\mathbf{P}_{SD} = \mathbf{Y}_{\text{FT}} \mathbf{X}_I^\top + \lambda\, \mathbf{W}_{\text{Teacher}}^{t-1} \mathbf{C}_I$;

- changing the *initialization* $\mathbf{W}_I^0 \to \mathbf{W}_I^{t-1}$ modifies the **null-space component** $\Pi_{\mathcal{Q}^\perp}(\cdot)$ which Lemma C.1 preserves from the initialization unchanged; this is what allows the orthogonal pretrained knowledge to be carried forward from one step to the next.

The dynamic-teacher scheme is therefore best understood as a sequence of standard static-SD-style quadratic minimizations, with both the anchor and the initialization updated between rounds in a way that affects geometrically separate subspaces. Proposition C.7 below makes this decomposition concrete in closed form.

*Proposition* C.7 (Single-Step Solution). Let $\mathbf{W}_{\text{FT}}^\star = \mathbf{Y}_{\text{FT}} \mathbf{X}_I^\top (\mathbf{X}_I \mathbf{X}_I^\top)^+$ be the minimum-norm solution for the direct finetuning task, and let $\mathcal{P}_I$ be the orthogonal projector onto $\text{range}(\mathbf{X}_I)$. The SD–WMA update at step $t$ yields

$$\mathbf{W}_I^t = \mathbf{W}_I^{t-1}(\mathbf{I} - \mathcal{P}_I) + \frac{\lambda}{1+\lambda} \mathbf{W}_{\text{Teacher}}^{t-1} \mathcal{P}_I + \frac{1}{1+\lambda} \mathbf{W}_{\text{FT}}^\star. \tag{18}$$

*Proof.* This proposition is immediate from applying Lemma C.1 to the objective in Definition C.3. The objective at step $t$ has the same structure as static self-distillation (analyzed in Theorem C.2), but with the pretrained weights $\mathbf{W}_I^0$ in the regularization term replaced by $\mathbf{W}_{\text{Teacher}}^{t-1}$, and the initialization for gradient descent being $\mathbf{W}_I^{t-1}$. Specifically, we find the minimizer of: $\min_{\mathbf{W}_I} \frac{1}{2} \|\mathbf{W}_I \mathbf{X}_I - \mathbf{Y}_{\text{FT}}\|_F^2 + \frac{\lambda}{2} \|\mathbf{W}_I \mathbf{X}_I - \mathbf{W}_{\text{Teacher}}^{t-1} \mathbf{X}_I\|_F^2$ This corresponds to the self-distillation case in Theorem C.2, where $\mathbf{W}_I^0$ is effectively replaced by $\mathbf{W}_{\text{Teacher}}^{t-1}$ for the purpose of defining the fixed regularization target at this step. The solution form is then directly obtained by substituting $\mathbf{W}_{\text{Teacher}}^{t-1}$ for $\mathbf{W}_I^0$ in the $\mathbf{W}_{SD}$ formula, which yields:

$$\mathbf{W}_I^t = \mathbf{W}_I^{t-1}(\mathbf{I} - \mathcal{P}_I) + \frac{1}{1+\lambda} \mathbf{Y}_{\text{FT}} \mathbf{X}_I^\top (\mathbf{X}_I \mathbf{X}_I^\top)^+ + \frac{\lambda}{1+\lambda} \mathbf{W}_{\text{Teacher}}^{t-1} \mathcal{P}_I.$$

Recognizing $\mathbf{W}_{\text{FT}}^\star = \mathbf{Y}_{\text{FT}} \mathbf{X}_I^\top (\mathbf{X}_I \mathbf{X}_I^\top)^+$, we get the desired result. $\qquad\square$

The key advantage over static SD emerges from the teacher's evolution.

*Theorem* C.8 (Bias-Free Convergence in the Task Subspace). Let $a = \frac{\lambda}{1+\lambda}$ and define the teacher error $\mathbf{E}^t = (\mathbf{W}_{\text{Teacher}}^t - \mathbf{W}_{\text{FT}}^\star)\mathcal{P}_I$. Then for any online weights $\{\omega_t\}$ as in equation 15:

(i) **Teacher contraction.** $\mathbf{E}^t = \left(1 - \frac{\omega_t}{1+\lambda}\right)\mathbf{E}^{t-1}$.

(ii) **Student tracking.** $(\mathbf{W}_I^t - \mathbf{W}_{\text{FT}}^\star)\mathcal{P}_I = a\,\mathbf{E}^{t-1}$.

(iii) **Convergence.** If $\sum_{t \geq 1} \omega_t = \infty$, then $\mathbf{W}_{\text{Teacher}}^t \mathcal{P}_I \to \mathbf{W}_{\text{FT}}^\star$ and $\mathbf{W}_I^t \mathcal{P}_I \to \mathbf{W}_{\text{FT}}^\star$.

*Proof.* Let $\mathbf{W}_{I,\|}^t = \mathbf{W}_I^t \mathcal{P}_I$ and $\mathbf{W}_{\text{Teacher},\|}^t = \mathbf{W}_{\text{Teacher}}^t \mathcal{P}_I$. From Proposition C.7, projecting onto the subspace range$(\mathbf{X}_I)$ gives:

$$\mathbf{W}_{I,\|}^t = \mathbf{W}_I^{t-1}(\mathbf{I} - \mathcal{P}_I)\mathcal{P}_I + \frac{\lambda}{1+\lambda}\,\mathbf{W}_{\text{Teacher}}^{t-1}\mathcal{P}_I + \frac{1}{1+\lambda}\,\mathbf{W}_{\text{FT}}^\star \mathcal{P}_I.$$

Since $(\mathbf{I} - \mathcal{P}_I)\mathcal{P}_I = \mathbf{0}$, and $\mathbf{W}_{\text{FT}}^\star$ is already in the parallel subspace (by definition), we have $\mathbf{W}_{\text{FT}}^\star \mathcal{P}_I = \mathbf{W}_{\text{FT}}^\star$. So,

$$\mathbf{W}_{I,\|}^t = a\,\mathbf{W}_{\text{Teacher},\|}^{t-1} + (1-a)\,\mathbf{W}_{\text{FT}}^\star, \tag{19}$$

where $a = \frac{\lambda}{1+\lambda}$. (ii) Subtracting $\mathbf{W}_{\text{FT}}^\star$ from both sides of equation 19:

$$\mathbf{W}_{I,\|}^t - \mathbf{W}_{\text{FT}}^\star = a\,\mathbf{W}_{\text{Teacher},\|}^{t-1} + (1-a)\,\mathbf{W}_{\text{FT}}^\star - \mathbf{W}_{\text{FT}}^\star = a\,(\mathbf{W}_{\text{Teacher},\|}^{t-1} - \mathbf{W}_{\text{FT}}^\star) = a\,\mathbf{E}^{t-1}.$$

This proves part (ii).

(i) Now consider the teacher recursion (Definition C.4) projected onto $\mathcal{P}_I$:

$$\mathbf{W}_{\text{Teacher},\|}^t = (1 - \omega_t)\,\mathbf{W}_{\text{Teacher},\|}^{t-1} + \omega_t\,\mathbf{W}_{I,\|}^t.$$

Substitute equation 19 into this:

$$\mathbf{W}_{\text{Teacher},\|}^t = (1 - \omega_t)\,\mathbf{W}_{\text{Teacher},\|}^{t-1} + \omega_t\,(a\,\mathbf{W}_{\text{Teacher},\|}^{t-1} + (1-a)\,\mathbf{W}_{\text{FT}}^\star).$$

Rearranging terms to isolate $\mathbf{E}^t = \mathbf{W}_{\text{Teacher},\|}^t - \mathbf{W}_{\text{FT}}^\star$:

$$\begin{aligned}
\mathbf{W}_{\text{Teacher},\|}^t - \mathbf{W}_{\text{FT}}^\star &= (1 - \omega_t)\,\mathbf{W}_{\text{Teacher},\|}^{t-1} + \omega_t a\,\mathbf{W}_{\text{Teacher},\|}^{t-1} + \omega_t(1-a)\,\mathbf{W}_{\text{FT}}^\star - \mathbf{W}_{\text{FT}}^\star \\
&= (1 - \omega_t + \omega_t a)\,\mathbf{W}_{\text{Teacher},\|}^{t-1} - (1 - \omega_t(1-a))\,\mathbf{W}_{\text{FT}}^\star \\
&= (1 - \omega_t(1-a))\,(\mathbf{W}_{\text{Teacher},\|}^{t-1} - \mathbf{W}_{\text{FT}}^\star).
\end{aligned}$$

Since $1 - a = 1 - \frac{\lambda}{1+\lambda} = \frac{1}{1+\lambda}$, we have:

$$\mathbf{E}^t = \left(1 - \frac{\omega_t}{1+\lambda}\right)\mathbf{E}^{t-1}.$$

This proves part (i).

(iii) Iterating the recurrence relation from part (i):

$$\|\mathbf{E}^t\|_F = \left(\prod_{k=1}^t \left(1 - \frac{\omega_k}{1+\lambda}\right)\right) \|\mathbf{E}^0\|_F.$$

For $\mathbf{E}^t$ to converge to 0, we need the product term to converge to 0. This occurs if and only if the sum $\sum_{k=1}^\infty \frac{\omega_k}{1+\lambda}$ diverges to $\infty$. Since $\lambda > 0$, $1 + \lambda$ is a finite constant. Thus, the condition for convergence is $\sum_{k=1}^\infty \omega_k = \infty$. From Definition C.4, $\omega_t = \frac{\alpha_t}{\sum_{j=0}^t \alpha_j}$. If $\kappa(\tau_t)$ is a continuous function on $[0, 1]$ that is non-zero on a set of positive measure, then $\sum_k \alpha_k$ will diverge as $T \to \infty$ (assuming $t$ goes up to $T$), and thus $\sum_k \omega_k$ will diverge. For common kernels like Beta distributions (e.g., arcsine kernel), this condition holds. Since $\mathbf{E}^t \to \mathbf{0}$, we have $\mathbf{W}_{\text{Teacher}}^t \mathcal{P}_I \to \mathbf{W}_{\text{FT}}^\star$. From part (ii), as $\mathbf{E}^{t-1} \to \mathbf{0}$, it follows that $(\mathbf{W}_I^t - \mathbf{W}_{\text{FT}}^\star)\mathcal{P}_I \to \mathbf{0}$, meaning $\mathbf{W}_I^t \mathcal{P}_I \to \mathbf{W}_{\text{FT}}^\star$. $\square$

*Corollary* C.9 (Linear rate under a bounded step weight). If $\omega_t \geq \omega_{\min} > 0$ for all $t \leq T$, then

$$\|(\mathbf{W}_{\text{Teacher}}^t - \mathbf{W}_{\text{FT}}^\star)\mathcal{P}_I\|_F \leq \left(1 - \frac{\omega_{\min}}{1+\lambda}\right)^t \|(\mathbf{W}_{\text{Teacher}}^0 - \mathbf{W}_{\text{FT}}^\star)\mathcal{P}_I\|_F.$$

Hence the training loss in the task subspace decays at least geometrically to the minimum, whereas static SD converges to a biased point for any fixed $\lambda > 0$.

*Proof.* This follows directly from Theorem C.8 part (i). If $\omega_t \geq \omega_{\min}$, then $1 - \frac{\omega_t}{1+\lambda} \leq 1 - \frac{\omega_{\min}}{1+\lambda}$. Since $0 < \omega_{\min} \leq 1$ and $\lambda > 0$, we have $0 < \frac{\omega_{\min}}{1+\lambda} < 1$, so $0 < 1 - \frac{\omega_{\min}}{1+\lambda} < 1$. Thus, the error contracts geometrically. Static SD, as derived in Theorem C.2, converges to a solution that is a convex combination of $\mathbf{W}_I^0 \mathcal{P}_I$ and $\mathbf{W}_{\text{FT}}^\star$. This is a biased point unless $\mathbf{W}_I^0 \mathcal{P}_I = \mathbf{W}_{\text{FT}}^\star$. $\square$

**Geometric Interpretation of Dynamic Self-Distillation.** We decompose the dynamics into orthogonal and parallel components with respect to range$(\mathbf{X}_I)$.

**Orthogonal Preservation.** Applying $(\mathbf{I} - \mathcal{P}_I)$ to Proposition C.7 and using the idempotency of projectors, $\mathcal{P}_I(\mathbf{I} - \mathcal{P}_I) = \mathbf{0}$, we get:

$$\mathbf{W}_I^t(\mathbf{I} - \mathcal{P}_I) = \mathbf{W}_I^{t-1}(\mathbf{I} - \mathcal{P}_I) = \cdots = \mathbf{W}_I^0(\mathbf{I} - \mathcal{P}_I),$$

This demonstrates that SD–WMA preserves pretrained knowledge orthogonal to the finetuning subspace, just like static self-distillation.

**Adaptive Task-Space Evolution.** Within the task subspace, the student update is given by:

$$\mathbf{W}_{I,\|}^t \;=\; \frac{\lambda}{1+\lambda}\,\mathbf{W}_{\text{Teacher},\|}^{t-1} \;+\; \frac{1}{1+\lambda}\,\mathbf{W}_{\text{FT}}^\star.$$

**Early training** ($t$ **small**): The teacher $\mathbf{W}_{\text{Teacher}}^{t-1}$ is still close to $\mathbf{W}_I^0$ (as $\omega_k$ for small $k$ is often high for U-shaped kernels, or simply because few updates have occurred). This means the teacher acts as a strong anchor, mitigating catastrophic forgetting during volatile updates.

**Late training** ($t$ **large**): As $t \to \infty$, Theorem C.8 shows that $\mathbf{W}_{\text{Teacher},\|}^{t-1}$ converges to $\mathbf{W}_{\text{FT}}^\star$. Substituting this into the student update:

$$\lim_{t \to \infty} \mathbf{W}_{I,\|}^t \;=\; \frac{\lambda}{1+\lambda}\,\mathbf{W}_{\text{FT}}^\star \;+\; \frac{1}{1+\lambda}\,\mathbf{W}_{\text{FT}}^\star \;=\; \mathbf{W}_{\text{FT}}^\star.$$

Thus, the dynamic teacher adapts, reducing anchor bias and enabling exact convergence to $\mathbf{W}_{\text{FT}}^\star$ in range($\mathbf{X}_I$).

*Proposition* C.10 (Dominance over Static SD). If $\|\mathbf{W}_{\text{Teacher},\|}^{t-1} - \mathbf{W}_{\text{FT}}^\star\|_F \le \|\mathbf{W}_{I,\|}^0 - \mathbf{W}_{\text{FT}}^\star\|_F$, then for the same $\lambda$ the SD–WMA update attains lower squared error than static SD in the task subspace.

*Proof.* Let $\mathbf{W}_{\text{static SD}}^\star$ be the solution for static SD (from Theorem C.2). The squared error from $\mathbf{W}_{\text{FT}}^\star$ in the task subspace for static SD is proportional to $\|\frac{\lambda}{1+\lambda}\mathbf{W}_I^0 \mathcal{P}_I - \mathbf{W}_{\text{FT}}^\star\|_F^2$. For dynamic SD, the instantaneous target is proportional to $\|\frac{\lambda}{1+\lambda}\mathbf{W}_{\text{Teacher}}^{t-1} \mathcal{P}_I - \mathbf{W}_{\text{FT}}^\star\|_F^2$. If the teacher is closer to $\mathbf{W}_{\text{FT}}^\star$ in the parallel subspace than the initial model $\mathbf{W}_I^0$, i.e., $\|\mathbf{W}_{\text{Teacher},\|}^{t-1} - \mathbf{W}_{\text{FT}}^\star\|_F \le \|\mathbf{W}_{I,\|}^0 - \mathbf{W}_{\text{FT}}^\star\|_F$, then the dynamic SD solution will be closer to $\mathbf{W}_{\text{FT}}^\star$ in that subspace, thus achieving lower error. The convergence result (Theorem C.8) guarantees that the teacher gets arbitrarily close to $\mathbf{W}_{\text{FT}}^\star$, eventually satisfying this condition. □

## C.6. Distillation Loss Definitions in TRACER

TRACER employs a composite self-distillation loss $\mathcal{L}_{\text{SD-WMA}}$ from the WMA teacher, which consists of several complementary terms to transfer different aspects of knowledge. Let $\mathbf{T}$ denote the teacher model and $\mathbf{S}$ denote the student model. $\mathbf{h}_{I_i}^{\mathbf{T}}$ and $\mathbf{h}_{T_i}^{\mathbf{T}}$ are image and text embeddings from the teacher for the $i$-th example, and similarly for the student. $\tau$ denotes the temperature parameter.

**Feature Distillation (FD).** This loss directly minimizes the Mean Squared Error between the student's and teacher's embeddings for each corresponding image-text pair in a mini-batch of size $N$. It helps align the feature spaces.

$$\mathcal{L}_{\text{FD}} = \frac{1}{N} \sum_{i=1}^{N} \left( \left\|\mathbf{h}_{I_i}^{\mathbf{T}} - \mathbf{h}_{I_i}^{\mathbf{S}}\right\|_2^2 + \left\|\mathbf{h}_{T_i}^{\mathbf{T}} - \mathbf{h}_{T_i}^{\mathbf{S}}\right\|_2^2 \right) \tag{20}$$

**Contrastive Relational Distillation (CRD).** CRD aligns the student's contrastive similarity distribution with the teacher's. We first compute the image-to-text ($p$) and text-to-image ($q$) softmax distributions for both student and teacher across the mini-batch:

$$p_i^{\mathbf{T}}[j] = \frac{\exp(\mathbf{h}_{I_i}^{\mathbf{T}\top}\mathbf{h}_{T_j}^{\mathbf{T}}/\tau)}{\sum_{b=1}^{N}\exp(\mathbf{h}_{I_i}^{\mathbf{T}\top}\mathbf{h}_{T_b}^{\mathbf{T}}/\tau)}, \qquad p_i^{\mathbf{S}}[j] = \frac{\exp(\mathbf{h}_{I_i}^{\mathbf{S}\top}\mathbf{h}_{T_j}^{\mathbf{S}}/\tau)}{\sum_{b=1}^{N}\exp(\mathbf{h}_{I_i}^{\mathbf{S}\top}\mathbf{h}_{T_b}^{\mathbf{S}}/\tau)} \tag{21}$$

$$q_i^{\mathbf{T}}[j] = \frac{\exp(\mathbf{h}_{T_i}^{\mathbf{T}\top}\mathbf{h}_{I_j}^{\mathbf{T}}/\tau)}{\sum_{b=1}^{N}\exp(\mathbf{h}_{T_i}^{\mathbf{T}\top}\mathbf{h}_{I_b}^{\mathbf{T}}/\tau)}, \qquad q_i^{\mathbf{S}}[j] = \frac{\exp(\mathbf{h}_{T_i}^{\mathbf{S}\top}\mathbf{h}_{I_j}^{\mathbf{S}}/\tau)}{\sum_{b=1}^{N}\exp(\mathbf{h}_{T_i}^{\mathbf{S}\top}\mathbf{h}_{I_b}^{\mathbf{S}}/\tau)} \tag{22}$$

The distillation loss is the sum of the KL-divergences between these distributions, averaged over the batch.

$$\mathcal{L}_{\text{CRD}} = \frac{1}{N} \sum_{i=1}^{N} \left( D_{KL}(p_i^{\mathbf{T}}\|p_i^{\mathbf{S}}) + D_{KL}(q_i^{\mathbf{T}}\|q_i^{\mathbf{S}}) \right) \tag{23}$$

**Interactive Contrastive Learning (ICL).** ICL forces the student to learn within the teacher's embedding space by performing contrastive learning between the student's anchor embeddings and the teacher's key embeddings. The loss is a symmetric InfoNCE objective computed on these mixed-model pairs.

$$\mathcal{L}_{\text{ICL}} = -\frac{1}{2N} \sum_{i=1}^{N} \left( \log \frac{\exp(\mathbf{h}_{I_i}^{\mathbf{S}\top}\mathbf{h}_{T_i}^{\mathbf{T}}/\tau)}{\sum_{j=1}^{N}\exp(\mathbf{h}_{I_i}^{\mathbf{S}\top}\mathbf{h}_{T_j}^{\mathbf{T}}/\tau)} + \log \frac{\exp(\mathbf{h}_{T_i}^{\mathbf{S}\top}\mathbf{h}_{I_i}^{\mathbf{T}}/\tau)}{\sum_{j=1}^{N}\exp(\mathbf{h}_{T_i}^{\mathbf{S}\top}\mathbf{h}_{I_j}^{\mathbf{T}}/\tau)} \right) \tag{24}$$

**Cross Knowledge Distillation (Cross-KD).** This method acts as a hybrid of CRD and ICL. It aligns the student-to-teacher cross-modal similarity distribution with the teacher's self-modal distribution using KL-divergence. We define the student-to-teacher

cross-modal distributions ($p^{\mathbf{S} \rightarrow \mathbf{T}}, q^{\mathbf{S} \rightarrow \mathbf{T}}$) as:

$$p_i^{\mathbf{S} \rightarrow \mathbf{T}}[j] = \frac{\exp(\mathbf{h}_{I_i}^{\mathbf{S} \top} \mathbf{h}_{T_j}^{\mathbf{T}} / \tau)}{\sum_{b=1}^{N} \exp(\mathbf{h}_{I_i}^{\mathbf{S} \top} \mathbf{h}_{T_b}^{\mathbf{T}} / \tau)} \tag{25}$$

$$q_i^{\mathbf{S} \rightarrow \mathbf{T}}[j] = \frac{\exp(\mathbf{h}_{T_i}^{\mathbf{S} \top} \mathbf{h}_{I_j}^{\mathbf{T}} / \tau)}{\sum_{b=1}^{N} \exp(\mathbf{h}_{T_i}^{\mathbf{S} \top} \mathbf{h}_{I_b}^{\mathbf{T}} / \tau)} \tag{26}$$

The loss then minimizes the divergence from these distributions to the teacher's own relational distributions, $p_i^{\mathbf{T}}$ and $q_i^{\mathbf{T}}$.

$$\mathcal{L}_{\text{CrossKD}} = \frac{1}{2N} \sum_{i=1}^{N} \left( D_{KL}(p_i^{\mathbf{T}} \| p_i^{\mathbf{S} \rightarrow \mathbf{T}}) + D_{KL}(q_i^{\mathbf{T}} \| q_i^{\mathbf{S} \rightarrow \mathbf{T}}) \right) \tag{27}$$

**Geometric bridge to composite distillation.** Our analysis decomposes learning into an orthogonal preservation term and an in-subspace mixing term (Equation 2). The composite distillation terms are chosen to preserve *structure* consistent with this geometry: (i) **FD** anchors pointwise embeddings, biasing updates toward the teacher component within $\text{range}(\mathbf{X}_I)$ while damping drift in orthogonal directions; (ii) **CRD** aligns the teacher's batch-wise similarity *distributions*, preserving inter-example geometry (a probabilistic surrogate for preserving $\mathbf{S} = \mathbf{H}_I^{\top} \mathbf{H}_T$); (iii) **ICL** performs contrastive learning in the teacher's semantic space, encouraging the student to operate on the teacher's subspace and thus to mix along task-relevant directions; and (iv) **CrossKD** aligns cross-modal logits to transmit cross-modal relational structure that vanilla InfoNCE may underweight. Together with the **WMA** teacher, these terms operationalize the geometric principle at feature-, relation-, and cross-modal levels.

## C.7. Connection to Robustness via Inter-Class Feature Sharing

The self-distillation approach, particularly with a dynamic WMA teacher, can be understood through the lens of recent theoretical work on multimodal contrastive learning's robustness mechanisms. Xue et al. (2024) identify *inter-class feature sharing* as a key mechanism behind MMCL's strong robustness to distribution shift, where models learn to leverage information about features appearing across different classes to dissociate spurious correlations.

Building on the insight that self-distillation acts as instance-specific label smoothing (Zhang & Sabuncu, 2020), we argue that the self-distillation method provides a similar robustness benefit by acting as an **informed label smoothing mechanism** that preserves inter-class similarities learned during pretraining. To see this connection, recall the self-distillation solution from Theorem C.2:

$$\mathbf{W}_{SD} = \mathbf{W}_I^0 \left( \mathbf{I} - \frac{1}{1+\lambda} \mathcal{P}_I \right) + \frac{1}{1+\lambda} \mathbf{Y}_{\text{FT}} \mathbf{X}_I^{\top} (\mathbf{X}_I \mathbf{X}_I^{\top})^+ \tag{28}$$

This solution exhibits three key properties that enhance robustness:

**Preservation of Cross-Class Knowledge.** The term $\mathbf{W}_I^0 \left( \mathbf{I} - \frac{1}{1+\lambda} \mathcal{P}_I \right)$ maintains the pretrained model's understanding of feature relationships across classes. Unlike direct finetuning which completely overwrites representations in the finetuning subspace, self-distillation retains a weighted contribution from the original cross-class feature covariances. This is analogous to how Xue et al. (2024) show that MMCL leverages features appearing in multiple contexts to learn their independence from class labels.

**Informed Smoothing via Pretrained Similarities.** By regularizing towards $\mathbf{W}_I^0 \mathbf{X}_I$ rather than arbitrary targets, self-distillation performs label smoothing that is informed by the pretrained model's learned inter-class similarities. This extends the instance-specific label smoothing interpretation of Zhang & Sabuncu (2020) to the finetuning setting, where the smoothing is guided by pretrained knowledge. This regularization preserves the cross-covariance structure that Xue et al. (2024) identify as crucial for robustness, specifically the covariance between features that appear independently across different classes.

**Robustness Through Feature Independence.** Within the finetuning subspace, self-distillation computes a convex combination:

$$\frac{\lambda}{1+\lambda} \left( \mathbf{W}_I^0 \mathcal{P}_I \right) + \frac{1}{1+\lambda} \left( \mathbf{Y}_{\text{FT}} \mathbf{X}_I^{\top} (\mathbf{X}_I \mathbf{X}_I^{\top})^+ \right) \tag{29}$$

This combination maintains the pretrained understanding of feature independence while adapting to the new task. As Xue et al. (2024) demonstrate in their Data Model 2, when features can occur independently across classes (e.g., "trees without green leaves" appearing in non-tree classes), models that preserve these cross-class relationships achieve stronger robustness. The self-distillation mechanism explicitly preserves these relationships through the weighted contribution of $\mathbf{W}_I^0 \mathcal{P}_I$.

The hyperparameter $\lambda$ controls the strength of this inter-class knowledge preservation: larger values of $\lambda$ maintain more of the pretrained model's understanding of how features vary independently across different contexts, potentially enhancing robustness to distribution shift. This suggests that self-distillation's effectiveness stems not merely from preventing catastrophic forgetting, but from actively preserving the rich inter-class feature relationships that contribute to robustness, a mechanism that parallels the theoretical insights of Xue et al. (2024) on why MMCL achieves strong out-of-distribution generalization.

# D. TRACER Algorithm

---

**Algorithm 1** TRACER (Trajectory-Robust Anchoring for Contrastive Encoder Regularization)

---

**Require:** Pretrained CLIP model $\theta_{\text{CLIP}}^0 = \{\mathcal{E}_{\text{Image}}^0, \mathcal{E}_{\text{Text}}^0\}$
**Require:** Finetuning dataset $\mathcal{D}_{\text{FT}} = \{(\mathbf{x}_I, \mathbf{x}_T)\}_{i=1}^N$
**Require:** Learning rate $\eta$, Weight decay $\delta$, Batch size $B$, Number of epochs $E$
**Require:** Distillation coefficient $\lambda_{\text{SD}}$
**Require:** WMA kernel $\kappa(\tau_k)$ (e.g., Beta($\beta_1, \beta_2$)) and total steps $T_{\text{total}}$
**Require:** Temperature $\tau_{\text{NCE}}$ for InfoNCE losses
1: **Initialize Student Model:** $\theta_S \leftarrow \theta_{\text{CLIP}}^0$ (image encoder $\mathcal{E}_{\text{Image},S}$, text encoder $\mathcal{E}_{\text{Text},S}$)
2: **Initialize Teacher Model:** $\theta_T \leftarrow \text{copy}(\theta_S)$
3: **Initialize Optimizer:** Opt $\leftarrow$ AdamW($\theta_S$.parameters(), $\eta, \delta$)
4: **Initialize WMA state:** cumulative_alpha $\leftarrow 0$
5: global_step $\leftarrow 0$
6: **for** epoch $= 1$ to $E$ **do**
7:     **for** batch $= \{(\mathbf{x}_I, \mathbf{x}_T)\}_{i=1}^B$ in $\mathcal{D}_{\text{FT}}$ **do**
8:         global_step $\leftarrow$ global_step $+ 1$
        {— Student Forward Pass —}
9:         $\mathbf{h}_{I,S} \leftarrow \mathcal{E}_{\text{Image},S}(\mathbf{x}_I)$
10:        $\mathbf{h}_{T,S} \leftarrow \mathcal{E}_{\text{Text},S}(\mathbf{x}_T)$
11:        Normalize student embeddings: $\mathbf{h}_{I,S} \leftarrow \text{normalize}(\mathbf{h}_{I,S})$, $\mathbf{h}_{T,S} \leftarrow \text{normalize}(\mathbf{h}_{T,S})$
        {— Compute Multi-Modal Contrastive Loss ($\mathcal{L}_{\text{MMCL}}$) —}
12:        $\text{logits}_{I \leftrightarrow T} \leftarrow \mathbf{h}_{I,S} \cdot \mathbf{h}_{T,S}^\top / \tau_{\text{NCE}}$
13:        $\mathcal{L}_{\text{MMCL}} \leftarrow \text{InfoNCE}(\text{logits}_{I \leftrightarrow T}) + \text{InfoNCE}(\text{logits}_{I \leftrightarrow T}^\top)$ {Symmetric InfoNCE}
        {— Teacher Forward Pass (with no gradient updates) —}
14:        **with** torch.no_grad() :
15:        $\mathbf{h}_{I,T} \leftarrow \mathcal{E}_{\text{Image},T}(\mathbf{x}_I)$
16:        $\mathbf{h}_{T,T} \leftarrow \mathcal{E}_{\text{Text},T}(\mathbf{x}_T)$
17:        Normalize teacher embeddings: $\mathbf{h}_{I,T} \leftarrow \text{normalize}(\mathbf{h}_{I,T})$, $\mathbf{h}_{T,T} \leftarrow \text{normalize}(\mathbf{h}_{T,T})$
        {— Compute Dynamic Self-Distillation Loss ($\mathcal{L}_{\text{SD-WMA}}$) —}
18:        $\mathcal{L}_{\text{FD}} \leftarrow \frac{1}{B} \sum_{i=1}^B (\|\mathbf{h}_{I,T}[i] - \mathbf{h}_{I,S}[i]\|_2^2 + \|\mathbf{h}_{T,T}[i] - \mathbf{h}_{T,S}[i]\|_2^2)$
19:        $\mathcal{L}_{\text{CRD}} \leftarrow \text{KL}(\text{softmax}(\mathbf{h}_{I,T}\mathbf{h}_{T,T}^\top / \tau_{\text{NCE}}) \| \text{softmax}(\mathbf{h}_{I,S}\mathbf{h}_{T,S}^\top / \tau_{\text{NCE}}))$ {+ text-to-image}
20:        $\mathcal{L}_{\text{ICL}} \leftarrow \text{InfoNCE}(\mathbf{h}_{I,S}, \mathbf{h}_{T,T}) + \text{InfoNCE}(\mathbf{h}_{T,S}, \mathbf{h}_{I,T})$
21:        $\mathcal{L}_{\text{CrossKD}} \leftarrow \text{KL}(\text{softmax}(\mathbf{h}_{I,T}\mathbf{h}_{T,T}^\top / \tau_{\text{NCE}}) \| \text{softmax}(\mathbf{h}_{I,S}\mathbf{h}_{T,T}^\top / \tau_{\text{NCE}}))$ {+ text-to-image}
22:        $\mathcal{L}_{\text{SD-WMA}} \leftarrow \mathcal{L}_{\text{FD}} + \mathcal{L}_{\text{CRD}} + \mathcal{L}_{\text{ICL}} + \mathcal{L}_{\text{CrossKD}}$
        {— Total Loss and Optimization —}
23:        $\mathcal{L}_{\text{Total}} \leftarrow \mathcal{L}_{\text{MMCL}} + \lambda_{\text{SD}} \cdot \mathcal{L}_{\text{SD-WMA}}$
24:        Opt.zero_grad()
25:        $\mathcal{L}_{\text{Total}}$.backward()
26:        Opt.step()
        {— Update WMA Teacher —}
27:        $\tau_{\text{current}} \leftarrow (\text{global\_step} + c_1)/(T_{\text{total}} + c_2)$ {Normalized time}
28:        $\alpha_{\text{current}} \leftarrow \kappa(\tau_{\text{current}})$
29:        cumulative_alpha $\leftarrow$ cumulative_alpha $+ \alpha_{\text{current}}$
30:        $\omega_{\text{current}} \leftarrow \alpha_{\text{current}}/\text{cumulative\_alpha}$
31:        **for** parameter $p_S$ in $\theta_S$ and $p_T$ in $\theta_T$ **do**
32:          $p_T \leftarrow (1 - \omega_{\text{current}}) \cdot p_T + \omega_{\text{current}} \cdot p_S$
33:        **end for**
34:     **end for**
35: **end for**
36: **return** $\theta_S$

---

# E. Reproducibility Details

To ensure full reproducibility, we detail our experimental setup, key hyperparameters, and implementation. The full codebase, configuration files, and reproduction scripts are publicly available at <https://github.com/HesamAsad/TRACER>.

## E.1. Computational Environment

- **Operating System:** Linux kernel 5.14.0-427.42.1.el9_4.x86_64.

- **GPU Hardware:** NVIDIA H100 80GB HBM3.

- **NVIDIA Driver Version:** 550.144.03.

- **CUDA Version:** 12.4.

- **Python Version:** 3.10.4.

- **PyTorch Version:** 2.0.1+ (with CUDA support).

## E.2. Implementation and Training Details

Our implementation extends the OpenAI CLIP framework.

- **Model Architectures:** We use pretrained CLIP models (ViT-B/16, ResNet50, ViT-L/14) from OpenAI's official `clip` library.

- **Total Loss:** $\mathcal{L}_{\text{TRACER}} = \mathcal{L}_{\text{MMCL}} + \lambda_{\text{SD}} \, \mathcal{L}_{\text{SD-WMA}}$.
  - $\mathcal{L}_{\text{MMCL}}$: Symmetric InfoNCE loss, directly leveraging OpenAI CLIP's core loss implementation. Optional cross-Frobenius regularizer coefficient was set to 0.05.
  - $\mathcal{L}_{\text{SD-WMA}}$: A composite self-distillation loss. For TRACER, this comprises Feature Distillation (FD), Contrastive Relational Distillation (CRD), Interactive Contrastive Learning (ICL), and Cross Knowledge Distillation (Cross-KD).

- **WMA Teacher:** A custom Weighted Moving Average (WMA) teacher implementation, whose weighting kernel is a Beta distribution with $\beta_1 = \beta_2 = 0.5$.

## E.3. Key Hyperparameters

The following hyperparameters were used for TRACER finetuning on ImageNet-1K:

- **Epochs:** 10.

- **Optimizer:** AdamW.

- **Learning Rate:** $1 \times 10^{-5}$.

- **Weight Decay:** 0.1.

- **Batch Size:** 512 (ViT-B/16, RN50), 224 (ViT-L/14).

- **Warmup Length:** 500 steps (cosine LR schedule).

- **Mixed Precision:** Enabled using `torch.amp.autocast` with `torch.bfloat16`.

- **Distillation Coefficient $\lambda_{\text{SD}}$:** 0.9.

- **WMA Beta Kernel Parameter:** 0.5 (for Beta(0.5,0.5) kernel, i.e., arcsine distribution).

- **Teacher Update Frequency:** 0 or 1 (update every step).

### E.4. Data Processing

Standard OpenAI CLIP image preprocessing was applied. Input images are sourced from ImageNet-1K, and finetuning captions from OpenAI class templates.

### E.5. Code Availability

The full codebase is publicly available at https://github.com/HesamAsad/TRACER to facilitate direct reproduction. The repository includes the TRACER training pipeline, evaluation scripts for all reported ImageNet and OOD benchmarks, configuration files for the three CLIP backbones (RN50, ViT-B/16, ViT-L/14), and the WMA teacher implementation.

# F. Extended Discussion

This appendix expands on three discussion points that we touch on briefly in the main text: how TRACER's gain scales across backbones, how TRACER relates to parameter-efficient and prompt-based adaptation, and how the theory could be extended beyond the linearized setting.

## F.1. Backbone scaling: ViT-B/16 vs. ViT-L/14

A natural question is why TRACER's OOD gain over CaRot is larger on ViT-B/16 ($+1.53$ Avg. shifts in Table 1) than on ViT-L/14 ($+1.19$ Avg. shifts in Table 7). Three factors plausibly contribute. First, ViT-L/14 is a substantially stronger zero-shot starting point (70.93% vs. 58.39% OOD), so there is less absolute headroom for any regularizer to recover; both methods sit closer to a ceiling, compressing differences. Second, even with a smaller absolute gap, TRACER still attains the *best OOD accuracy* among all compared methods on ViT-L/14 (75.32%, vs. 74.13% for CaRot), with consistent gains on the hardest shifts (IN-R $+1.74$, IN-A $+2.19$, ObjectNet $+1.71$), mirroring the ViT-B/16 pattern. Third, TRACER scales *favorably in compute*: its distillation cost is $\mathcal{O}(B^2)$ in the batch and matches standard EMA cost for the teacher update (Table 4), so its overhead does *not* grow with the cubic-in-$d$ spectral term that dominates CaRot at large $d$. We therefore expect the favorable cost–quality trade-off to persist or improve as backbones grow. We frame this as an empirical observation: we do *not* claim a demonstrated scaling law for TRACER across the full vision–language model size spectrum, and we view systematic studies on larger VLMs (e.g., SigLIP variants and CLIP-style backbones at $\geq$ViT-H/14) as natural follow-up work.

## F.2. Relation to PEFT and prompt-based adaptation

TRACER is a regularization mechanism that operates on the standard full-finetuning gradient flow: it modifies *which solution* the optimizer converges to (anchoring it to a trajectory-weighted teacher), but it does *not* restrict *where* parameters can move. Parameter-efficient finetuning (PEFT) methods such as adapters (Houlsby et al., 2019) and LoRA (Hu et al., 2022), as well as prompt-based adaptation (Zhou et al., 2022; Jia et al., 2022), take the orthogonal route: they restrict the hypothesis class so that only a small set of injected parameters or input tokens can change, leaving the pretrained weights untouched by construction. The two strategies address different failure modes of finetuning. PEFT bounds the *capacity* for drift away from the pretrained model; TRACER bounds the *direction* and *magnitude* of drift within whatever capacity the optimizer is given. Because TRACER's WMA teacher and composite distillation losses act on the student's parameters (or, equivalently, its embedding statistics) regardless of how many of those parameters are trainable, TRACER can in principle wrap a LoRA-finetuned or prompt-tuned model directly: the WMA teacher would then average a low-dimensional trajectory in adapter/prompt space rather than full-encoder space. We therefore see TRACER as *complementary* to PEFT and prompt-based adaptation rather than competing with them; a systematic empirical study of TRACER on top of LoRA, adapters, and prompt tuning is a natural follow-up.

## F.3. Future work: NTK and random-feature extensions of the theory

Our theoretical analysis (§3, §C) is developed in the linearized image/text-encoder setting that is standard in the contrastive-learning theory literature (Ji et al., 2023; Tian, 2022; Nakada et al., 2023; Xue et al., 2024). In that setting, the closed-form solutions, the contrastive target matrix, and the bias-free convergence of the SD–WMA teacher all admit clean derivations that we believe capture the essential mechanism underlying TRACER's empirical robustness on nonlinear CLIP backbones (Figures 3–4). A natural next step is to lift these results to the *neural tangent kernel* (NTK) regime (Jacot et al., 2018) and to *random-feature* (RF) models, where the nonlinear encoders are approximated by linear maps in a feature space induced by their gradients or by random nonlinearities. In the NTK regime, the same matrix least-squares reformulation should apply with $\mathbf{X}_I$ replaced by NTK features, allowing both the orthogonal/parallel decomposition and the WMA bias-free convergence theorem to be re-derived for finite-width networks under lazy training. The RF view, in turn, would let us study how the spectrum of the random feature map interacts with the WMA kernel $\kappa(\tau)$ and the regularization strength $\lambda$. Establishing such extensions would tighten the connection between our linearized analysis and large nonlinear vision–language models, and we leave them as a concrete open direction.

## G. AI Usage

Large Language Models were used to improve the manuscript's grammar and readability, and to assist with code formatting and refactoring during implementation. All research design, theoretical analysis, experimental protocols, and interpretation of results were conducted entirely by the authors.

