# OpenReview forum: "TRACER: Persistent Regularization for Robust Multimodal Finetuning"
_ICML.cc/2026/Conference — ICML 2026 regular_

### Official Review · Reviewer_Ctzn · 2026-03-11

**Soundness:** 2
**Presentation:** 3
**Significance:** 3
**Originality:** 3
**Overall Recommendation:** 4
**Confidence:** 3

**Summary:**

In this paper, the authors study the problem of catastrophic forgetting in finetuning CLIP models. They first consider a linear case where the image and text encoders are linear projections, and develop analyses for different finetuning methods (e.g., direct finetuning, static self-distillation, etc). Then they propose a variant of self-distillation method where the teacher model is updated using weighted average. Next, they extend the idea to practical finetuning of CLIP models to propose the TRACER method. Finally, they conduct experiments on two tasks to evaluate the effectiveness of their method.

**Compliance With Llm Reviewing Policy:**

Affirmed.

**Final Justification:**

This papers presents a novel approach to address the problem of catastrophic forgetting in finetuning CLIP models, and the experiment results are promising. My concerns are addressed by the rebuttal, thus I am leaning towards accepting the paper.

**Key Questions For Authors:**

- Key Questions
  1. Line 188 left part: What is the meaning of 'static SD ... can introduce bias because it uses a fixed anchor'?
  2. Theorem 3.4: If the student's projection converges to $W_{FT}^{*}P_{I}$, then how is the student model guaranteed to avoid catastrophic forgetting?
  3. Theorem 3.4: What is the convergence for EMA? Will the student model converge to the same solution as WMA?
  4. Section 5.2: TRACER demonstrates better OOD performance than baselines, but how is this related to catastrophic forgetting?
  5. How are the results of TRACER compared with static self-distillation?
- Other Questions
  1. Figure 2: Conceptually, what is the meaning of $W_{FT}^{*}$?
  2. Definition 3.3: What is $c_{1}, c_{2}$?
  3. Section C.1: Why is minimizing $\frac{1}{2}||W\_{I}X\_{I}-Y\_{FT}||\_{F}^{2}$ equivalent to minimizing $-Tr(Y\_{FT}^{T}W\_{I}X\_{I})$? In my understanding, minimizing the former is equivalent to minimizing $\frac{1}{2}||W\_{I}X\_{I}||\_{F}^{2}- Tr(Y\_{FT}^{T}W\_{I}X\_{I})$.
- Suggestions
  1. Definition 3.1: The definition of $\mathbf{I}$ and $\mathbf{J}$ should be provided.
  2. Theorem 3.4: The term 'optimal direct finetuning solution' can be confused with the direct finetuning solution in Theorem 3.2
  3. line 304 right part: References should be added for baselines such as FLYP and CaRot.

**Limitations:**

yes

**Strengths And Weaknesses:**

- Soundness: The analyses in the linear case are sound, with a few minor issues requiring clarification (please see Key Questions). The experiments are well designed in general, but are missing important baselines such as static self-distillation, which is the most relevant baseline to the proposed method.
- Presentation: The submission is overall clearly written and well structured
- Significance: This papers studies the problem of catastrophic forgetting when finetuning CLIP models, which is a significant problem. The authors leverage linearized analysis to develop a theoretical framework to analyze different fintuning methods. The analyses provide insights on the behavior of different methods, though the scope is limited since an unrealistic setting (i.e., linear model) is considered. The proposed TRACER method provides a practical solution to mitigate catastrophic forgetting.
- Originality: The analyses of different methods in the linearized setting provide new insights. The idea of using weighted moving average to update the teacher model in self-distillation is novel.

---

> ### Author Rebuttal · Authors · 2026-03-31
>
> We thank Reviewer Ctzn for the careful reading and constructive questions. We are encouraged that the reviewer finds the problem significant, the paper generally clear, and the WMA idea novel. We address the main technical points below.
>
> **KQ1.** In the task subspace, static SD converges to
>
> $$
> W_{\mathrm{SD}}\mathcal{P}_I = \frac{\lambda}{1+\lambda} W_I^0\mathcal{P}_I + \frac{1}{1+\lambda} W_{FT}^{\ast}
> $$
>
> Thus, for any fixed $\lambda>0$, the solution remains offset from $W_{FT}^{\ast}$ unless $W_I^0\mathcal{P}_I = W_{FT}^{\ast}$. This is the sense in which static SD introduces **bias**: the fixed anchor keeps pulling the solution toward initialization within the task subspace. Larger $\lambda$ preserves more pretrained structure, but also increases this in-subspace bias.
>
> **KQ2.** The full solution has two parts: a task-subspace component, which should adapt, and an orthogonal component, which should be preserved. Our convergence theorem concerns the first part, while SD-WMA also preserves the orthogonal component exactly:
>
> $$
> W_I^t(I - \mathcal{P}_I) = W_I^0(I - \mathcal{P}_I)
> $$
>
> So the method adapts where the fine-tuning data provides signal while retaining pretrained knowledge outside that subspace. This is the geometric reason it reduces forgetting. Empirically, this is consistent with the toy experiment and the representational similarity results in Figure 4.
>
> **KQ3.** Our current theory does **not** prove that EMA-guided training converges to the same limit as WMA. What we show is the practically important distinction: EMA is strongly recency-weighted, so the teacher can collapse toward the student and the regularization signal weakens late in training, whereas WMA retains contribution from earlier states over the finite training horizon and yields bias-free task-subspace convergence in our analysis. Thus, the key difference is in finite-horizon training dynamics.
>
> **KQ4.** In this literature, degradation of OOD robustness after finetuning is a primary symptom of catastrophic forgetting for pretrained CLIP-like models. Our paper supports this in three ways: (1) Direct FT improves ID accuracy but can reduce OOD average below zero-shot; (2) the toy experiment measures forgetting directly and shows Dynamic Distillation forgetting only 0.1%, versus 37.9% for Direct FT; and (3) Figure 4 shows that TRACER preserves representational similarity to the pretrained model much better than Direct FT. We will make this connection more explicit in §5.2.
>
> **KQ5.** Static SD is already included in Table 2, and we agree we should make this easier to notice. On ViT-B/16, TRACER improves substantially over Static SD in average OOD accuracy, with especially large gains on challenging shifts such as IN-R and IN-A. We will make this comparison more prominent in revision.
>
> **OQ1.** ($W_{FT}^{\ast}$) is the **minimum-norm task solution** for the least-squares reformulation,
>
> $$
> W_{FT}^{\ast} = Y_{FT} X_I^\top(X_I X_I^\top)^+
> $$
>
> It is the task-only solution inside the fine-tuning subspace. It is **not** the full direct-finetuning solution, because the latter also includes the preserved orthogonal component from $W_I^0$. We will clarify this distinction in revision.
>
> **OQ2.** $c_1$ and $c_2$ are small positive constants used to map the discrete training index to normalized time $\tau_k\in(0,1)$, so Beta-type kernels do not evaluate at the singular endpoints 0 and 1. In practice, we use $c_1=0.5$ and $c_2=1$. In the current version, we state only that $c_1,c_2>0$; in the CR version we will make their practical role explicit and specify these implementation values clearly.
>
> **OQ3.** The reviewer is right that this needs more careful wording. Expanding the square gives
>
> $$
> \frac{1}{2}\lVert W_I X_I - Y_{FT} \rVert_F^2 = \frac{1}{2}\lVert W_I X_I \rVert_F^2 - \mathrm{Tr}(Y_{FT}^\top W_I X_I) + \frac{1}{2}\lVert Y_{FT} \rVert_F^2
> $$
>
> So the equivalent optimization is **not** just minimizing the negative trace term alone; the $\frac{1}{2}\lVert W_I X_I \rVert_F^2$ term also matters. Importantly, this does **not** affect our results: all closed-form solutions and subsequent theorems are derived from the least-squares objective itself, not from $-\mathrm{Tr}(\cdot)$ alone. We will revise §C.2 to make this fully precise.
>
> We also appreciate the reviewer’s suggestions and will adopt them in CR version: explicitly define $I_n$ and $J_n$ near Definition 3.1, replace “optimal direct finetuning solution” with “minimum-norm task solution”, and add explicit references for baselines.

---

> > ### Author Rebuttal · Reviewer_Ctzn · 2026-04-04
> >
> > Thank you for your response. My concerns are addressed, and I am raising my score to 4.

---

> > > ### Author Response · Authors · 2026-04-08
> > >
> > > Thank you for your response. We are glad that our rebuttal addressed your concerns, and we sincerely appreciate your updated assessment and support for the paper.

---

### Official Review · Reviewer_RNQz · 2026-03-12

**Soundness:** 3
**Presentation:** 3
**Significance:** 3
**Originality:** 3
**Overall Recommendation:** 4
**Confidence:** 3

**Summary:**

This paper studies the loss of out-of-distribution (OOD) robustness that often occurs when pretrained multimodal models are finetuned. The authors develop a theoretical framework that reformulates contrastive finetuning as a matrix least-squares problem and provides a geometric interpretation of how finetuning modifies pretrained representations. Based on this analysis, the paper proposes a finetuning method that uses a weighted moving average teacher to maintain persistent regularization during training. Experiments on several CLIP backbones and ImageNet distribution shift benchmarks demonstrate improved OOD robustness compared to existing finetuning approaches.

**Compliance With Llm Reviewing Policy:**

Affirmed.

**Final Justification:**

The rebuttal addresses my main concerns, but I would like to keep my rating.

**Key Questions For Authors:**

The single-step update in Proposition C.7 is described as following directly from the static self-distillation result in Theorem C.2. Could the authors clarify this derivation? In particular, it would be helpful to explain why the same solution form still holds when the teacher replaces the pretrained anchor and the initialization changes at each step.

**Limitations:**

Yes.

**Strengths And Weaknesses:**

Strengths

The paper addresses an important and practical problem: the loss of out-of-distribution (OOD) robustness during finetuning of pretrained multimodal models. This submission intends to study the concept of persistent regularization during contrastive finetuning, and the manuscript focuses on a central concept: maintaining a meaningful teacher–student gap to preserve pretrained knowledge during adaptation.

The work proposes a theoretically motivated framework that reformulates contrastive finetuning as a matrix least-squares problem, providing a useful geometric interpretation of how different finetuning strategies affect pretrained representations.

The proposed TRACER method is well motivated by the theoretical analysis, and the design of the weighted moving average teacher is clearly connected to the theoretical insights.

The empirical evaluation is reasonably comprehensive, including multiple CLIP backbones and several standard distribution shift benchmarks, along with ablation studies that analyze different components of the method.

The paper is generally well structured and clearly written, making the main ideas and contributions easy to follow.

Weaknesses

Some parts of the theoretical analysis are presented concisely, and certain derivations, such as the connection between the static self-distillation formulation and the dynamic SD-WMA update, could benefit from additional clarification.

Although several strong finetuning baselines are included, the discussion of alternative CLIP adaptation strategies is relatively limited, which could further strengthen the experimental context.

The proposed framework combines multiple distillation components, which increases methodological complexity and may make it harder to isolate the contribution of individual components.

The novelty mainly lies in the formulation and analysis rather than in introducing entirely new algorithmic components, and the experiments focus primarily on CLIP-style architectures, leaving the broader applicability of the approach less explored.

---

> ### Author Rebuttal · Authors · 2026-03-31
>
> We sincerely thank Reviewer RNQz for the thoughtful review and for recognizing the importance of the problem, the usefulness of the theoretical framework, the connection between the WMA design and the analysis, and the strength of the empirical study. We address the main question and the broader concerns below.
>
> ## KQ1:
>
> This is the key technical bridge from the static analysis to the dynamic method. The short answer is:
>
> **At each step, SD-WMA is still a static quadratic optimization problem with a fixed anchor during that step.**
>
> The objective at step (t) is
>
> $\frac{1}{2}\|W_I X_I - Y_{FT}\|_F^2 + \frac{\lambda}{2}\|W_I X_I - W_{Teacher}^{t-1}X_I\|_F^2$
>
> initialized from $W_I^{t-1}$. During optimization at step (t), the teacher ${W_{\text{Teacher}}^{t-1}}^*$ is fixed, so the problem has the same quadratic form as static self-distillation.
>
> The two substitutions affect different parts of Lemma C.1:
>
> * replacing $W_I^0$ by $W_{\text{Teacher}}^{t-1}$ changes the **range component**, because it changes the anchor in the objective;
> * replacing the initialization $W_I^0$ by $W_I^{t-1}$ changes the **null-space component**, because gradient descent preserves whatever the initialization contributes in directions not acted on by the operator.
>
> This is exactly why Proposition C.7 takes the form
> $$
> W_I^t =
> W_I^{t-1}(I-\mathcal P_I)
> +
> \frac{\lambda}{1+\lambda}W_{\text{Teacher}}^{t-1}\mathcal P_I
> +
> \frac{1}{1+\lambda}W_{FT}^*.
> $$
> We will make this bridge more explicit in the appendix by adding a short paragraph before Proposition C.7.
>
> ## W1: Some derivations could benefit from more clarification.
>
> That’s a good suggestion, the main place where clarification is most needed is precisely the bridge from static SD to dynamic SD-WMA. We will revise that part of the appendix to state the two substitutions explicitly and explain which part of the decomposition each one affects.
>
> ## W2: Discussion of alternative CLIP adaptation strategies is somewhat limited.
>
> We appreciate this point. Our current experiments already compare to a broad set of baselines spanning direct finetuning, L2-style regularization, static self-distillation, two-stage methods, post-hoc averaging, and several contrastive robust finetuning methods. In particular, Table 2 includes Direct FT, L2-SP, Static SD, LP-FT, FLYP, CAR-FT, Lipsum-FT, Model Stock, ARF, and CaRot.
>
> That said, we agree the paper can better situate TRACER relative to **parameter-efficient finetuning** (PEFT) methods such as adapters/LoRA and **prompt-based** adaptation methods, which are complementary rather than directly competing.
>
> We will add this discussion in the camera-ready version.
>
> ## W3: Multiple distillation components increase complexity; their roles are harder to isolate.
>
> We designed Ablation 1 specifically to address this concern. Table 8 evaluates all $2^4$ combinations of the four components, including each single component in isolation and the full combination. This lets us isolate the role of each term rather than only reporting an end-to-end result.
>
> The ablation suggests that (1) FD and CRD are the strongest single components, (2) the components are complementary rather than redundant, (3) combining all four gives the best overall performance across accuracy/calibration trade-offs.
>
> We will make that interpretation more explicit in the main text.
>
> ## W4: The novelty is more in formulation/analysis than in a new algorithmic primitive; experiments focus mainly on CLIP-style models.
>
> We agree that the main novelty is theoretical and mechanistic rather than arising from a completely unfamiliar architectural primitive. But we view that as a strength of the paper. The main contributions are:
>
> 1. a reformulation of linearized multimodal contrastive finetuning through the contrastive target matrix;
> 2. a geometric decomposition explaining where forgetting occurs;
> 3. identification of the late-stage weakness of recency-biased teachers;
> 4. a WMA-based design that follows from this analysis and yields consistent empirical gains.
>
> On scope, we focus on CLIP because it is the standard setting in robust multimodal finetuning, and we evaluate on three backbones covering CNN and Transformer families. We agree that broader evaluation on newer architectures would be valuable future work, and we will emphasize that more clearly.
>
> ---
>
> We thank the reviewer again for the constructive feedback. In revision, we will make the Proposition C.7 bridge more explicit, strengthen the discussion of related adaptation paradigms, and better highlight the component-isolating role of the ablations.

---

> > ### Author Rebuttal · Reviewer_RNQz · 2026-04-04
> >
> > Thank you for the detailed rebuttal. My main concerns, particularly regarding the theoretical connection between static self-distillation and the dynamic WMA update, as well as the positioning relative to existing CLIP adaptation methods, have been addressed. I still believe the paper has some limitations in scope and presentation, but the clarifications are helpful and I appreciate the planned revisions.

---

> > > ### Author Response · Authors · 2026-04-08
> > >
> > > Thank you for your thoughtful follow-up. We are glad that our rebuttal addressed your main concerns. We appreciate your constructive feedback and will incorporate the planned revisions to further strengthen the paper.

---

### Official Review · Reviewer_zSpT · 2026-03-13

**Soundness:** 3
**Presentation:** 2
**Significance:** 3
**Originality:** 3
**Overall Recommendation:** 4
**Confidence:** 3

**Summary:**

The paper proposes TRACER, a method for robust multimodal fine-tuning that aims to mitigate the loss of out-of-distribution (OOD) robustness caused by catastrophic forgetting. The authors develop a theoretical framework for multimodal contrastive fine-tuning by reformulating the objective as a matrix least-squares problem using a contrastive target matrix, which enables closed-form analysis and a geometric interpretation of knowledge preservation. The analysis suggests that commonly used EMA teachers can collapse when the teacher–student gap vanishes, weakening regularization. To address this, the paper proposes a Weighted Moving Average (WMA) teacher that aggregates the full optimization trajectory to maintain persistent regularization. Based on this idea, TRACER combines contrastive learning with WMA-guided multi-perspective distillation. Experiments on CLIP fine-tuning show improvements in OOD accuracy and calibration across several backbone architectures.

**Compliance With Llm Reviewing Policy:**

Affirmed.

**Final Justification:**

The rebuttal has addressed my main concerns.

**Key Questions For Authors:**

1. Subfigures (a) and (b) of Figure 2 appear largely identical. It may be sufficient to keep only one of them.
2. What is the realistic meaning of $Y_{FT}$? Is it a fixed matrix during fine-tuning?
3. The use of multiple colors in Figure 3 seems unnecessary and somewhat distracting. Simplifying the visualization may improve readability.

**Limitations:**

yes

**Strengths And Weaknesses:**

### Strengths

- The topic of fine-tuning pretrained multimodal models while preserving robustness is important and relevant.
- The paper includes extensive experimental results across multiple architectures and settings.

### Weaknesses

- The theoretical analysis is limited to linear image and text encoders. While this may be acceptable as a starting point, more expressive yet still analyzable frameworks such as NTK or random feature models could potentially provide stronger insights.
- The presentation is quite dense, which makes it difficult to follow the main message. For example, Figure 2 contains many symbols that make the figure hard to interpret.
- The paper lacks a clear system setup and preliminaries. After the introduction and related work, the paper directly moves into theoretical analysis without sufficiently introducing the baseline formulation. For instance, the original linearized multimodal contrastive learning (MMCL) objective is not clearly presented before introducing the contrastive target matrix $Y_{FT}$.
- The geometric interpretation is difficult to understand, possibly due to insufficient explanation of the newly defined contrastive target matrix.

---

> ### Author Rebuttal · Authors · 2026-03-31
>
> We sincerely thank the reviewer for the careful reading and for highlighting both the importance of the problem and the breadth of the experiments. We agree that the main weakness of the current draft is presentation and accessibility rather than the core technical contribution. In the revision, we will make four concrete changes: (1) add a clearly labeled “Problem Setting and Preliminaries” subsection at the start of §3; (2) present the original MMCL objective before introducing the reformulation; (3) explain $Y_{FT}$ explicitly as a fixed “contrastive target” induced by the frozen text side; and (4) simplify Figures 2 and 3 with less clutter and stronger annotations.
>
> **W1: Linearity of the theory.** We agree that NTK or random-feature extensions would be valuable future work. Our goal here is not to claim a complete nonlinear theory, but to isolate the mechanism of forgetting in a tractable setting. The linearized framework already enables us to derive the contrastive target matrix, a closed-form comparison of Direct FT / L2-SP / Static SD, and the distinction between EMA and WMA teachers. Just as importantly, the paper does not stop at the linear setting: in §5.1 the toy study uses nonlinear encoders, and the qualitative ordering predicted by the theory is preserved—Direct FT forgets most, L2 helps, Static Distillation helps more, and Dynamic Distillation performs best. On full CLIP backbones, TRACER beats CaRot by 1.5 points OOD on ViT-B/16 and 1.2 on ViT-L/14, with better calibration and 25% less compute overhead. We will add a short paragraph in the revision explicitly positioning NTK/random-feature extensions as an important direction for future work.
>
> **W2: The presentation is dense, especially Figure 2.** Thank you for the suggestion. The intended message of Figure 2 is actually simple: Direct FT preserves the orthogonal component and replaces the parallel component; L2-SP globally blends directions without a clean structural separation; Static SD preserves the orthogonal component and mixes the parallel component by a convex combination. This structure is already reflected in Theorem 3.2, but the current figure is visually denser than necessary. In the revision we will reduce background clutter, add in-figure labels emphasizing “preserve orthogonal / replace parallel / global blend / convex mix,” and make the method vectors more visually distinct. We will also streamline Figure 3 to help  the readers focus on the trade-off, and not be distracted by the color palette.
>
> **W3: Missing setup and preliminaries.** We fully agree, and this is the most actionable point. In the revision, we will begin §3 with the task setup for paired image-text finetuning, the linearized encoder definitions, the original MMCL objective before reformulation, and the key notation $(I_n, J_n, P_I)$. This will let the reader first see the standard objective and then understand $Y_{FT}$ as a natural algebraic reformulation, rather than encountering it too abruptly. The reviewer is right that, in the current draft, Definition 3.1 appears before enough intuition has been established.
>
> **W4 / Q2: Meaning of $Y_{FT}$.** $Y_{FT}$ is the fixed contrastive target implied by the frozen text encoder and the finetuning text batch in our theoretical setup. It encodes, in matrix form, the same “attract the matched text / repel the other texts” structure as the contrastive loss. This is why the linearized MMCL objective can be rewritten as a least-squares problem in $W_I$. We agree that this intuition should appear immediately after Definition 3.1 in the main text, and we will add a short explanation there. A useful analogy is that $Y_{FT}$ plays the role of a target matrix in regression, except that here the target is induced by the frozen text side of the contrastive problem.
>
> **Q1: Figure 2(a) and 2(b) look similar.** While the subfigures look similar, conceptually they serve different purposes: (a) shows a structured decomposition—orthogonal preservation plus full replacement inside the task subspace—whereas (b) shows an unstructured global ridge blend. We have included both, because we believe that contrast is important for motivating self-distillation, but we agree the visual design should make the difference much clearer and we will update this in the CR version.
>
> **Q3: Figure 3 uses too many colors.** We agree and will simplify the styling in the CR version.
>
> In summary, we agree that the current draft can be made substantially easier to follow. The technical content is already present, but the exposition should stage it better. We will revise the paper accordingly by introducing a preliminary subsection, moving the MMCL objective into the main text, adding a direct explanation of $Y_{FT}$, and simplifying the figures. We thank the reviewer for their careful reading, and the improvements they suggest will significantly increase the accessibility of our paper.

---

> > ### Author Rebuttal · Reviewer_zSpT · 2026-04-04
> >
> > Thanks for the response.

---

> > > ### Author Response · Authors · 2026-04-08
> > >
> > > Thank you for your response and acknowledgment. We are pleased that our rebuttal fully addressed your concerns and we sincerely appreciate your time and consideration.

---

### Official Review · Reviewer_mjdd · 2026-03-13

**Soundness:** 3
**Presentation:** 3
**Significance:** 3
**Originality:** 3
**Overall Recommendation:** 6
**Confidence:** 4

**Summary:**

The paper explains how finetuning multimodal models can harm OOD robustness by causing catastrophic forgetting, and shows that commonly used EMA teachers fail to prevent this in late training. It proposes a stronger alternative, TRACER, based on weighted trajectory-aware distillation, and demonstrates consistent robustness and calibration improvements in CLIP finetuning.

**Compliance With Llm Reviewing Policy:**

Affirmed.

**Final Justification:**

The paper explains how finetuning multimodal models can harm OOD robustness, and derives geometric decomposition to effectively separate task-subspace mixing from orthogonal preservation. The rebuttal is detailed and successfully addresses my concerns about the motivation of WMA and its applicability to other weighting kernels. Given the contribution of this paper to the practical robust multimodal finetuning field, I decide to increase my score to further support the paper’s acceptance.

**Key Questions For Authors:**

1. From experiments, WMA is good, which is another contribution from this paper. However, the initial intuition should be explained clearly. I feel the current version does not well-motivate the WMA method. The authors should explain more on this perspective.

2. Are other weighting kernels suitable for WMA?

3. The improved margin over L14 is less than that over B16. Can the authors explain the reason? There is a concern that the improvement might be too small if there are large VLMs. It would be better to have some discussion or analysis here.

**Limitations:**

yes

**Strengths And Weaknesses:**

Pros:

1. This paper addresses an important problem: Most methods are working on fine-tuning but ignore that the finetuning process might cause a larger failure of OOD robustness. If not considering this, after the finetuning, the model might be more sensitive to OOD changes, causing potential security concerns in some scenarios.

2. This paper is clearly written, with strong motivation for both the overall problem and each proposed component, making the algorithm design well justified. The ablation studies further demonstrate the effectiveness of each module in the proposed method.

3. The theoretical contribution of this paper is solid. The authors derive a geometric decomposition that separates task–subspace mixing from orthogonal preservation, explaining when and where forgetting occurs and providing a principled basis for dynamic teachers. This contribution is quite novel in the field of finetuning VLMs.

4. The empirical improvement in performance is clear, highlighting the significance of the proposed method.


Cons:

1. From experiments, WMA is good, which is another contribution from this paper. However, the initial intuition should be explained clearly. I feel the current version does not well-motivate the WMA method. The authors should explain more on this perspective.

2. Figure 1 and Figure 2 should be switched? Section 3.3 is discussing WMA, but with Figure 2, instead of Figure 1.

3. Are other weighting kernels suitable for WMA?

4. It is better to have a problem setting section in the paper. This can make more readers exactly know what the task is.

5. The improved margin over L14 is less than that over B16. Can the authors explain the reason? There is a concern that the improvement might be too small if there are large VLMs. It would be better to have some discussion or analysis here.

---

> ### Author Rebuttal · Authors · 2026-03-31
>
> We sincerely thank Reviewer mjdd for voting for acceptance, recognizing the importance of the problem, the clarity of the writing, the novelty of the geometric decomposition, and the significance of the empirical results and ablation studies. We address each point raised by the reviewer below.
>
> ## Q1: The intuition for WMA
>
> Following the suggestion we will strengthen the intuition in §3.3 before the formal definition.
>
> The motivation for WMA comes from two limitations of existing anchors:
>
> * **Static self-distillation** preserves pretrained knowledge, but its anchor is fixed at initialization. As a result, within the task subspace it converges to a convex combination of the pretrained projection and the task solution, which creates a persistent bias away from the task optimum for any fixed $\lambda > 0$.
> * **EMA teachers** are strongly recency-biased. As training progresses, the teacher tracks the student more and more closely, so the teacher-student gap can shrink and the regularization signal weakens late in training.
>
> The role of WMA is to integrate the advantages of both:
>
> * it remains **dynamic**, so the anchor can move toward the task solution rather than remain fixed;
> * it still remembers **earlier states**, so the regularization signal does not collapse as quickly as with EMA over the finite training horizon.
>
> A concise intuition is: Static SD stays close to the starting point; EMA stays close to the current state; WMA stays close to a trajectory-weighted consensus that remembers the start but adapts over time.
>
> We will add this intuition directly before Definition 3.3.
>
> ## Q2: Should Figure 1 and Figure 2 be switched?
>
> We believe the current ordering is appropriate, but we acknowledge that the presentation could be improved.
>
> * **Figure 1** is the method overview: it shows the student-teacher architecture and how the WMA teacher is updated.
> * **Figure 2** is the geometric interpretation of how different finetuning strategies modify pretrained knowledge.
>
> In §3.3, we refer to both for different purposes: Figure 2 when explaining the limitation of static SD and the geometry of preservation vs adaptation. Figure 1 when introducing the dynamic WMA teacher used in TRACER. So we would keep the current order, but we will make the transition in §3.3 more explicit so the distinction is easier to follow.
>
> ## Q3: Are other weighting kernels suitable for WMA?
>
> Yes. The framework is kernel-agnostic: any nonnegative kernel over normalized time can be used in the WMA construction. We have discussed  this in the definition of the WMA teacher (line 1625) and studied kernel shape explicitly in the appendix ablation.
>
> In Table 11, we evaluate symmetric Beta kernels with different $\beta$ values. The results show that endpoint-emphasizing kernels $(\beta < 1)$ work best for OOD robustness, while mid-trajectory emphasis $(\beta > 1)$ is weaker. Intuitively, this is consistent with our goal: preserve useful early information while still emphasizing late task-adapted states.
>
> We use Beta(0.5,0.5) as the main default in the paper because it is a simple, principled arcsine-like choice and performs strongly, but the main conclusion is that WMA is not tied to one kernel.
>
> ## Q4: It would help to have a clearer problem setting section.
>
> Thank you for the suggestion, in the CR version, we will begin §3 with a short **Problem Setting and Preliminaries** subsection that includes (1) the finetuning task definition; (2) the linearized image/text encoder setup; (3) the original MMCL objective before reformulation; (4) the key notation used later in the theory.
>
> This will make the theoretical section more self-contained and easier to read.
>
> ## Q5: The gain on ViT-L/14 is smaller than on ViT-B/16. Does performance shrink on larger VLMs?
>
> This is a good question. We believe the below three points help explain the result.
>
> First, **ViT-L/14 starts from a stronger baseline**, so there is naturally less room for relative improvement. Even so, TRACER still achieves the best OOD accuracy among the reported methods on that backbone.
>
> Second, the gain over the strongest baseline remains meaningful. In Table 7, TRACER improves over CaRot on ViT-L/14 in average OOD accuracy (75.32 vs 74.13), with similar calibration. The largest gains appear on challenging shifts such as IN-R, IN-A, ObjectNet, and IN-S.
>
> Third, from a complexity perspective, TRACER’s distillation operates on batch similarity structure, whereas CaRot includes a spectral component that scales less favorably with representation dimension. So the computational structure of TRACER is favorable as models grow.
>
> We will add a short discussion clarifying this point and explicitly frame broader scaling as future work rather than as a demonstrated claim.
>
> ---
>
> We thank the reviewer again for the positive evaluation. We will strengthen the WMA intuition, add a problem-setting subsection, and improve the transition between the geometric theory and the method description.

---

> > ### Author Rebuttal · Reviewer_mjdd · 2026-04-02
> >
> > Many thanks for addressing my concerns. Now, they are fully clarified. I appreciate the contributions of this paper, especially the geometric decomposition part. Thus, I decide to increase my score to further support this submission.

---

> > > ### Author Response · Authors · 2026-04-08
> > >
> > > Thank you very much for your thoughtful follow-up. We are glad that our rebuttal helped clarify your concerns, especially regarding the geometric decomposition. We sincerely appreciate your updated assessment and support for the paper.

---

### Decision · Program_Chairs · 2026-04-30

**Decision:**

Accept (regular)

**Comment:**

This paper provides an analysis and a new method for robust fine-tuning of multimodal models. The authors formulate the fine-tuning optimization as a matrix least-squares problem whose closed form solutions shed new mathematical light on three contrastive fine-tuning approaches: direct (unregularized), L2-regularized, and with static self-distillation. This shows that static self-distillation can make the solution too biased towards the fixed-anchor of the fine-tuning initialization. On the other hand, EMA teachers provide regularization that weakens throughout the fine-tuning process and, hence, cannot counteract the out-of-distribution degradation that emerges late in training. This motivates the proposed Weighted Moving Average (WMA) that provides a useful trajectory-weighted regularization that mitigates the main weaknesses of the static self-distillation and EMA. This mathematically motivates the proposed TRACER method, which is shown to perform well on CLIP fine-tuning tasks.

Based on the paper, the reviews, the successful rebuttal by the authors and the following discussions, I find the new mathematical insights and the practical TRACER method as solid contributions to the area of multimodal fine-tuning. My recommendation is therefore to accept this paper.

Upon acceptance, the authors are requested to add the clarifications and explanations, as well as any other promised item, as appeared in the rebuttal and their discussions with the reviewers.